

# Orbifolds of chiral fermionic CFTs and their duality

**Kohki Kawabata and Shinichiro Yahagi**

Department of Physics, Faculty of Science, The University of Tokyo,
Bunkyo-Ku, Tokyo 113-0033, Japan

## Abstract

We consider chiral fermionic conformal field theories (CFTs) constructed from lattices and investigate their orbifolds under reflection and shift $\mathbb{Z}_2$ symmetries. For lattices based on binary error-correcting codes, we show the duality between reflection and shift orbifolds using a triality structure inherited from the binary codes. Additionally, we systematically compute the partition functions of the orbifold theories for both binary and nonbinary codes. Finally, we explore applications of this code-based construction in the search for supersymmetric CFTs and chiral fermionic CFTs without continuous symmetries.

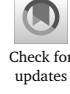

# 1 Introduction

Orbifold is a powerful tool to construct new consistent two-dimensional quantum field theories from a given theory by gauging a global symmetry [1–3]. A notable example for us is the construction of the chiral bosonic CFT with the Monster group symmetry, which elucidates the Monstrous moonshine phenomenon [4,5]. The Monster CFT is constructed by two orbifolds of the chiral bosonic CFT based on a lattice by the reflection $\mathbb{Z}_2$ symmetry $G$ generated by $X \to -X$ and the shift $\mathbb{Z}_2$ symmetry $H$ generated by $X \to X + \pi\delta$, where $X$ is an $n$-dimensional periodic free boson and $\delta$ is a vector specifying the half shift of the periodicity. These procedures rule out continuous symmetry in the original theory and lead to a consistent chiral CFT with the Monster group symmetry.

This paper aims to study the reflection and shift orbifolds of chiral fermionic CFTs based on odd self-dual lattices. The chiral fermionic CFTs are characterized by a half-integer spin operator in the spectrum. When the lattice is odd self-dual, the corresponding set of vertex operators contains half-integer spin operators and gives rise to the Neveu-Schwarz (NS) sector of a chiral fermionic CFT. The Ramond (R) sector is given by the vertex operators based on the shadow of the lattice, a kind of half-shift of the lattice. For a non-anomalous $\mathbb{Z}_2$ symmetry, we find two ways of $\mathbb{Z}_2$ orbifold denoted by $\pm$ that change the NS sector of the original theory $\mathcal{T}$. The shift orbifolds can be interpreted as a modification of the original lattice and the theories $(\mathcal{T}/H)_\pm$ after orbifolding are still a lattice CFT. On the other hand, the reflection orbifold theories $(\mathcal{T}/G)_\pm$ are not a lattice CFT after orbifolding.

One of the main results of this paper is the duality between the reflection and shift orbifolds in chiral fermionic CFTs, when a lattice is constructed from a binary error-correcting code. This result expands the previous one for the bosonic theories constructed from error-correcting codes [6,7]. A binary error-correcting code $C \subset \mathbb{F}_2^n$ is a vector space over a finite field $\mathbb{F}_2 = \{0, 1\}$ and yields an odd self-dual lattice $\Lambda(C) \subset \mathbb{R}^n$ by uplifting codewords to lattice vectors by the so-called Construction A. Finally, this lattice $\Lambda(C)$ gives rise to a chiral fermionic CFT $\mathcal{T}$ of central charge $n$ [8]. In this case, at least one shift $\mathbb{Z}_2$ symmetry $H$ is present and becomes non-anomalous when $n \in 8\mathbb{Z}$. The shift orbifold is a modification of the original lattice $\Lambda(C)$ to another one $\widetilde{\Lambda}(C)$. Since the original theory $\mathcal{T}$ always contains SU(2) current algebras inherited from codewords in the binary code, we can construct the permutation of three generators in SU(2) called the triality. Using the triality, we show the equivalence between the reflection and shift orbifolds:

$$\mathcal{T}/G \cong \mathcal{T}/H \qquad (C: \text{binary code}), \tag{1}$$



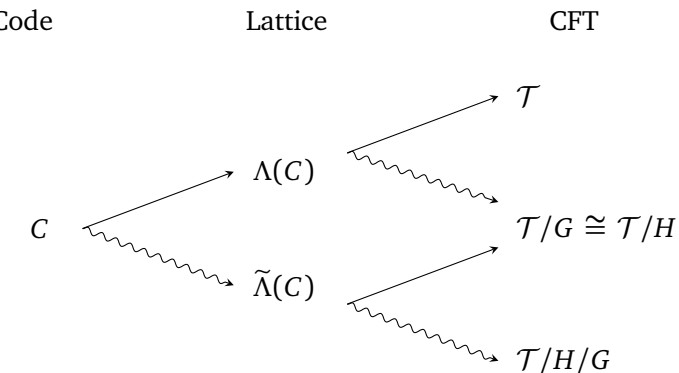

Figure 1: The duality between orbifolds of the reflection $\mathbb{Z}_2$ symmetry $G$ and the shift $\mathbb{Z}_2$ symmetry $H$ in a chiral fermionic CFT from a binary code. For the code-to-lattice arrows, the straight line shows Construction A and the wavy line does its half shift. For the lattice-to-CFT arrows, the straight lines represent lattice CFT construction and the wavy lines do their reflection orbifold. The theory $\mathcal{T}$ is constructed by the previous work [8], and the reflection and shift orbifold together yield three theories from a binary code.

where we denote them by $\mathcal{T}/G$ and $\mathcal{T}/H$ since the two types of orbifold ($\pm$) coincide in the binary construction. We can proceed to the orbifold of the shift orbifold $\mathcal{T}/H$ by the reflection symmetry, which leads to the new theory $\mathcal{T}/H/G$. We show the whole picture of the equivalence between the orbifolds in chiral fermionic CFTs based on binary codes in Fig. 1. We also explicitly present their torus partition functions, which allows us to compute their spectra and identify the profiles of chiral fermionic CFTs.

We extend our analysis to the shift orbifolds of chiral fermionic CFTs constructed from nonbinary $\mathbb{Z}_k$ codes $C \subset \mathbb{Z}_k^n$ where $\mathbb{Z}_k$ is a ring of integers modulo $k \in \mathbb{Z}_{\geq 2}$. We generalize the previous works [8, 9] constructing chiral fermionic CFTs from $\mathbb{F}_p$ codes for a prime number $p$ to $\mathbb{Z}_k$ codes for a positive integer $k \geq 2$. When the code is not binary, the equivalence between reflection and shift orbifolds does not hold in general. However, we can choose a shift $\mathbb{Z}_2$ symmetry independent of the code $C$, which enables a systematic study of the shift orbifolds. We obtain the formula for computing the orbifold partition functions on a torus from the weight enumerators of nonbinary codes.

With the general construction in hand, we provide various chiral fermionic CFTs based on binary and nonbinary codes and their orbifolds. By applying our construction to binary codes of length 16, we reproduce the classification result ( [10–12]) of chiral fermionic CFTs with central charge 16. Up to equivalence, there are only five binary singly-even self-dual codes. We leverage these codes to six odd self-dual lattices and seven chiral fermionic CFTs. This provides the relationship between code, lattice, and CFTs (see Fig. 5 for the results). Also, we find evidence that the reflection and shift orbifolds preserve supersymmetry by checking that some necessary conditions for supersymmetry hold in orbifold theories if the original theory $\mathcal{T}$ satisfies them. Finally, we search for more theories of interest with central charges above 24. At $c = 24$, we construct the "Beauty and the Beast" superconformal field theory (SCFT) [13] and the Baby Monster CFT [14] from binary codes, both of which have sporadic group symmetries. At $c = 32, 40$, we give an example of chiral fermionic CFT with spectral gap $\Delta = 2$. This implies that there do not exist spin-one currents generating continuous symmetry in this theory and only discrete symmetry can exist.

The organization of this paper is as follows. In section 2, we review general aspects of chiral fermionic CFTs with a global $\mathbb{Z}_2$ symmetry. We introduce two types ($\pm$) of orbifolds by

composing the topological operations and present the general prescription for $\mathbb{Z}_2$-orbifolds. Section 3 is devoted to the reflection and shift orbifolds of chiral fermionic CFTs based on binary codes. We give a detailed analysis of the shift orbifolds and their interpretation as the lattice modification. We also show the computation of torus partition functions of the reflection orbifold. In section 4, we construct chiral fermionic CFTs from $\mathbb{Z}_k$ codes through lattices, which is a generalization of the previous works [8,9]. The main purpose of section 5 is to show the equivalence between the reflection and shift orbifolds in chiral fermionic CFTs based on binary codes. We also extend our analysis to nonbinary codes and give the formula for computing the orbifold partition functions in section 6. In section 7, we present various examples of chiral fermionic CFTs and their orbifolds as an application of our general construction. Finally, we conclude in section 8 and discuss future directions.

## 2 Fermionic CFTs and $\mathbb{Z}_2$ orbifolds

The main interest of this paper is a 2d chiral fermionic CFT with a $\mathbb{Z}_2$ symmetry. Since the theory is chiral, its spectrum consists of only the left-moving sector and the right-moving sector is trivial. A fermionic CFT contains an operator with a half-integer spin in its spectrum. Any fermionic CFT has a fermion parity symmetry $\mathbb{Z}_2^f$ generated by $(-1)^F$, which acts on fermionic operators as sign flip and on bosonic operators trivially. We assume the presence of an additional bosonic $\mathbb{Z}_2$ symmetry $\mathsf{G} = \{1, \mathsf{g}\}$. By a bosonic symmetry, we mean a symmetry that does not change a spin structure by the insertion of the corresponding symmetry defect. Namely, we are working on a 2d theory $\mathcal{T}$ with $\mathbb{Z}_2^f \times \mathbb{Z}_2$ symmetry.

Our spacetime is a manifold $M$ that admits a spin structure. Typically we consider a Riemann surface. To define a fermionic theory, we need to specify the choice of a spin structure, labeled by the holonomies around the different cycles of $M$. We denote by $Z_{\mathcal{T}}[\gamma]$ the partition function on $M$ equipped with a spin structure $\gamma$. Furthermore, we can introduce a background $\mathbb{Z}_2$ connection $\alpha \in H^1(M, \mathsf{G})$ associated with the bosonic $\mathbb{Z}_2$ symmetry $\mathsf{G}$. We denote by $Z_{\mathcal{T}}[\gamma; \alpha]$ the partition function on a Riemann surface with spin structure $\gamma$ and $\mathbb{Z}_2$ connection $\alpha$.

In this paper, we mostly set our spacetime as a torus $T^2$. The cylindrical coordinate is $w = x_1 + \mathrm{i} x_0$, identified with $w \sim w + 1 \sim w + \tau$ where $\tau$ is a modulus of the torus. We set $x_1$ as a spatial direction and $x_0$ as a temporal one. The additional structure on a torus (e.g., spin structure and background $\mathbb{Z}_2$ configuration) can be fixed by the periodicities along spatial and temporal cycles. For each cycle, the Neveu-Schwarz (NS) sector sets the anti-periodic boundary condition $\psi \to -\psi$ and the Ramond (R) sector does the periodic boundary condition $\psi \to \psi$ where $\psi$ denotes a fermionic field. We can summarize a spin structure on a torus as $\gamma = (s_0, s_1)$, $s_i \in \{0, 1\}$ by denoting NS $\leftrightarrow 0$ and R $\leftrightarrow 1$. Similarly, the $\mathbb{Z}_2$ connection on a torus can be specified by $\alpha = (a_0, a_1)$, $a_i \in \{0, 1\}$ where $a_i$ represents the periodicity $\phi \to \mathsf{g}^{a_i} \cdot \phi$ for a field $\phi$.

The partition function on a torus with a spin structure $\gamma = (s_0, s_1)$ and a background $\mathbb{Z}_2$ connection $\alpha = (a_0, a_1)$ can be written as

$$Z_{\mathcal{T}}[s_0, s_1; a_0, a_1] = \mathrm{Tr}_{\mathcal{H}_{\rho(s_1), \iota(a_1)}} \left[ \mathsf{g}^{a_0} (-1)^{s_0 F} q^{L_0 - \frac{n}{24}} \right], \tag{2}$$

where the partition function of a chiral CFT depends only on $q = e^{2\pi \mathrm{i} \tau}$. For notation convenience, we define $\rho : \{0, 1\} \to \{\mathrm{NS}, \mathrm{R}\}$ such that $\rho(0) = \mathrm{NS}$, $\rho(1) = \mathrm{R}$ and $\iota(a) = \mathsf{g}^a$ for $a \in \{0, 1\}$. Here, $\mathcal{H}_{\rho(s_1), \iota(a_1)}$ is the Hilbert space quantized on the periodicity twisted by the bosonic $\mathbb{Z}_2$ element $\iota(a_1) \in \mathbb{Z}_2$ under spin structure $\rho(s_1) \in \{\mathrm{NS}, \mathrm{R}\}$. We often omit the second index for the untwisted sector and simply denote $\mathcal{H}_{\mathrm{NS}, 1} = \mathcal{H}_{\mathrm{NS}}$ or $\mathcal{H}_{\mathrm{R}, 1} = \mathcal{H}_{\mathrm{R}}$.

In what follows, we consider the orbifold of a chiral fermionic CFT by a bosonic $\mathbb{Z}_2$ symmetry G. To define the orbifold theory consistently, we need to ensure the vanishing 't Hooft anomaly. In a 2d chiral theory, there occurs a gravitational anomaly characterized by $\nu_{\text{grav}} = -2c$ where $c$ is the central charge.[1] However, as we want to see the effects of orbifolding, we will focus on an 't Hooft anomaly of $\mathbb{Z}_2^f \times G$ symmetry below. Note that we need to care about $\mathbb{Z}_2^f \times G$ rather than only the bosonic symmetry G because the orbifold theory has to possess the fermion parity symmetry $\mathbb{Z}_2^f$.

The 't Hooft anomaly of $\mathbb{Z}_2^f \times G$ symmetry is classified by $\nu \in \mathbb{Z}_8$ [15]. The mod 8 anomaly $\nu$ can be constructed from the three layers ($\nu_1, \nu_2, \nu_3$):

$$\nu = \nu_1 + 2\,\nu_2 + 4\,\nu_4 \quad \text{mod } 8\,, \tag{3}$$

where each layer takes a mod 2 value

$$\nu_1 \in H^1(\mathbb{Z}_2, \mathbb{Z}_2) \cong \mathbb{Z}_2\,, \qquad \nu_2 \in H^2(\mathbb{Z}_2, \mathbb{Z}_2) \cong \mathbb{Z}_2\,, \qquad \nu_3 \in H^3(\mathbb{Z}_2, U(1)) \cong \mathbb{Z}_2\,. \tag{4}$$

The three layers $\nu_1$, $\nu_2$, and $\nu_3$ are called the Majorana layer, the Gu-Wen layer [16], and the bosonic layer, respectively. These layers admit a clear physical and geometric interpretation [17]. Concretely, each layer gives a way to encode rules to place SPT phases with no bosonic symmetries on facets of triangulation by G-symmetry lines. For example, the Majorana layer specifies a way of placing a 2-dimensional Arf theory, which is the non-trivial fermionic SPT phase without bosonic symmetry, on two-dimensional facets. See [17, 18] for more details.

We can diagnose the 't Hooft anomalies of symmetry G in terms of torus partition functions using modular transformation [9, 17, 19]. The mod 8 anomaly $\nu \in \mathbb{Z}_8$ can be read off from the eigenvalue $e^{i\pi\nu/4}$ of the modular transformation $ST^2S^{-1}$ on the relative partition function $Z_{\mathcal{T}}[0,0;1,0]/Z_{\mathcal{T}}[0,0;0,0]$:

$$ST^2S^{-1} : \frac{Z_{\mathcal{T}}[0,0;1,0]}{Z_{\mathcal{T}}[0,0;0,0]} \longrightarrow e^{\frac{i\pi\nu}{4}} \frac{Z_{\mathcal{T}}[0,0;1,0]}{Z_{\mathcal{T}}[0,0;0,0]}\,, \tag{5}$$

where we take the relative to cancel the contribution from the gravitational anomaly of a chiral CFT. Since this procedure uses only the NS-NS partition functions $Z_{\mathcal{T}}[0,0;1,0]$ and $Z_{\mathcal{T}}[0,0;0,0]$, we can compute the anomalies only if we know the action of symmetry in the NS sector. Below we assume the vanishing anomaly $\nu = 0$ mod 8, which allows us to gauge the $\mathbb{Z}_2$ symmetry G.

Let us consider the orbifold of the fermionic CFT by a non-anomalous bosonic $\mathbb{Z}_2$ symmetry G. Orbifolding is one of the topological manipulations since it does not change the local structure such as the chiral algebra and only modifies its global structure like the partition function. By combining topological manipulations, we arrive at various theories associated with a global symmetry G. Here, we focus on the bosonic operations, which map our fermionic theory to another fermionic theory. The bosonic operations on a fermionic theory with $\mathbb{Z}_2^f \times \mathbb{Z}_2$ are known to be generated by the three following operations [18]:

- The shift of the spin structure by the $\mathbb{Z}_2$ gauge field

$$\pi_F : Z_{\mathcal{T}}[s_0, s_1; a_0, a_1] \mapsto Z_{\mathcal{T}}[s_0 + a_0, s_1 + a_1; a_0, a_1]\,. \tag{6}$$

- The stacking of the Arf theory

$$S_F : Z_{\mathcal{T}}[s_0, s_1; a_0, a_1] \mapsto (-1)^{s_0 s_1} Z_{\mathcal{T}}[s_0, s_1; a_0, a_1]\,. \tag{7}$$

---

[1]The gravitational anomaly of a chiral CFT with central charge $c$ always can be canceled by coupling $2c$ Majorana-Weyl fermions to the anti-holomorphic sector.

- The orbifold of the bosonic $\mathbb{Z}_2$ symmetry

$$\mathcal{O} : Z_{\mathcal{T}}[s_0, s_1; a_0, a_1] \mapsto \frac{1}{2} \sum_{c_0, c_1 \in \mathbb{Z}_2} (-1)^{a_0 c_1 - a_1 c_0} Z_{\mathcal{T}}[s_0, s_1; c_0, c_1]. \tag{8}$$

We can combine the three topological manipulations to construct new consistent theories from the original fermionic theory with a non-anomalous $\mathbb{Z}_2$ symmetry G. These operations generate three independent Hilbert spaces up to stacking invertible phases: the original theory $\mathcal{T}$, the orbifold (+) theory $(\mathcal{T}/G)_+$, and the orbifold (−) theory $(\mathcal{T}/G)_-$. These are related by the following topological manipulations:

$$
\begin{aligned}
Z_{(\mathcal{T}/G)_+}[s_0, s_1; a_0, a_1] &= \mathcal{O} \cdot Z_{\mathcal{T}}[s_0, s_1; a_0, a_1], \\
Z_{(\mathcal{T}/G)_-}[s_0, s_1; a_0, a_1] &= (S_F \, \mathcal{O} \, \pi_F \, S_F \, \pi_F) \cdot Z_{(\mathcal{T}/G)_+}[s_0, s_1; a_0, a_1].
\end{aligned}
\tag{9}
$$

In terms of the Hilbert space, these theories are related by swapping the sectors as in table 1. The other theories generated by the topological manipulations are different in the R sector from the three theories $\mathcal{T}$, $(\mathcal{T}/G)_+$ and $(\mathcal{T}/G)_-$. Since the original theory $\mathcal{T}$ has two global symmetries $\mathbb{Z}_2^f$ and G, its Hilbert spaces are extended to twisted sectors with respect to the two symmetries. In total, the extended Hilbert spaces consist of $\mathcal{H}_{\mathrm{NS}}$, $\mathcal{H}_{\mathrm{R}}$, $\mathcal{H}_{\mathrm{NS,g}}$ and $\mathcal{H}_{\mathrm{R,g}}$. Furthermore, each Hilbert space is graded by the charges of the $\mathbb{Z}_2^f \times \mathbb{Z}_2$ symmetry, which finally leads to 16 sectors (A, B, …, P) on the top in table 1.

Orbifolding (+) swaps the original Hilbert space into the middle in table 1. After orbifolding (+), the bosonic $\mathbb{Z}_2$ symmetry becomes trivial because it acts trivially on any local operator. Instead, a dual $\mathbb{Z}_2$ symmetry $\check{G}$ arises in the orbifold (+) theory and the orbifold Hilbert space can be graded by the new $\mathbb{Z}_2$ symmetry [20, 21]. On the other hand, the orbifold (−) theory is still graded by the original $\mathbb{Z}_2$ symmetry G since its NS sector includes the sectors $K, L$, which are odd under the $\mathbb{Z}_2$ symmetry G before gauging. Note that the sectors $O, P$ do not appear in the untwisted sector for both orbifold theories.

## 3 $\mathbb{Z}_2$ orbifolds in lattice CFTs

Up to this point, we have described a general discussion about fermionic CFTs and their orbifolds by $\mathbb{Z}_2$ symmetry. From this section, we restrict our attention to the orbifolds of chiral fermionic CFTs based on lattices (lattice CFTs). In section 3.1, we give a brief review of lattice CFTs. In section 3.2 and 3.3, we analyze their orbifolds by the shift symmetry $H$ and the reflection symmetry $G$ in detail.

### 3.1 Lattice CFTs

Let us recall some definitions associated with lattices following [22] to introduce lattice CFTs. A lattice $\Lambda \subset \mathbb{R}^n$ is a discrete subgroup of $\mathbb{R}^n$, which spans a vector space. For a lattice $\Lambda$ with the standard inner product $x \cdot y = \sum_{i=1}^n x_i y_i$ for $x, y \in \mathbb{R}^n$, the dual lattice is defined by

$$\Lambda^* = \left\{ \lambda' \in \mathbb{R}^n \mid \lambda \cdot \lambda' \in \mathbb{Z}, \text{ for all } \lambda \in \Lambda \right\}. \tag{10}$$

A lattice $\Lambda$ is called integral when $\Lambda \subset \Lambda^*$ and self-dual when $\Lambda = \Lambda^*$. Moreover, an integral lattice is called even when $\lambda \cdot \lambda \in 2\mathbb{Z}$ for all $\lambda \in \Lambda$, and odd otherwise.

Let $\Lambda \subset \mathbb{R}^n$ be an odd self-dual lattice. It can be divided into two disjoint subsets: $\Lambda = \Lambda_0 \sqcup \Lambda_2$ where

$$\Lambda_0 = \left\{ \lambda \in \Lambda \mid \lambda^2 \equiv 0 \mod 2 \right\}, \qquad \Lambda_2 = \left\{ \lambda \in \Lambda \mid \lambda^2 \equiv 1 \mod 2 \right\}. \tag{11}$$

Table 1: The sectors in the original, orbifold (+), and orbifold (−) theory.

(a) Original theory $\mathcal{T}$.

|  |  | untwisted | | twisted | |
| --- | --- | --- | --- | --- | --- |
|  |  | NS | R | NS | R |
| g-even | boson | A | E | I | M |
|  | fermion | B | F | J | N |
| g-odd | boson | C | G | K | O |
|  | fermion | D | H | L | P |

(b) Orbifold theory $(\mathcal{T}/G)_+$.

|  |  | untwisted | | twisted | |
| --- | --- | --- | --- | --- | --- |
|  |  | NS | R | NS | R |
| ǧ-even | boson | A | E | C | G |
|  | fermion | B | F | D | H |
| ǧ-odd | boson | I | M | K | O |
|  | fermion | J | N | L | P |

(c) Orbifold theory $(\mathcal{T}/G)_-$.

|  |  | untwisted | | twisted | |
| --- | --- | --- | --- | --- | --- |
|  |  | NS | R | NS | R |
| g-even | boson | A | M | I | E |
|  | fermion | B | N | J | F |
| g-odd | boson | L | H | D | P |
|  | fermion | K | G | C | O |

The shadow of $\Lambda$ is defined by

$$S(\Lambda) = \Lambda_0^* \setminus \Lambda. \tag{12}$$

It is convenient to introduce a characteristic vector. A lattice vector $\chi \in \Lambda$ is called characteristic if $\lambda \cdot \lambda \equiv \chi \cdot \lambda \mod 2$ for all $\lambda \in \Lambda$. The shadow can be written as $S(\Lambda) = \Lambda + \frac{\chi}{2}$ for any characteristic vector $\chi \in \Lambda$, or equivalently, $S(\Lambda) = \{\frac{\chi}{2} \mid \chi : \text{a characteristic vector of } \Lambda\}$.

For a self-dual lattice $\Lambda$, one can construct a CFT by giving the set of vertex operators from $\Lambda$ (see, for example, [23, 24]). We construct a chiral fermionic CFT $\mathcal{T}$ with central charge $n$ from the lattice $\Lambda \subset \mathbb{R}^n$ by specifying a set of vertex operators in the Neveu-Schwarz (NS) and Ramond (R) sectors as

$$V_\lambda(z) =: e^{i\lambda \cdot X(z)} :, \quad \lambda \in \Lambda \qquad \text{(NS sector)}, \tag{13}$$

$$V_\xi(z) =: e^{i\xi \cdot X(z)} :, \quad \xi \in S(\Lambda) \qquad \text{(R sector)}, \tag{14}$$

where the colon denotes the normal ordering and $X(z)$ is an $n$-dimensional chiral scalar boson whose mode expansion is

$$X^j(z) = q^j - i p^j \ln z + i \sum_{n \neq 0} \frac{\alpha_n^j}{n} z^{-n}. \tag{15}$$

Here, $q$ is a position operator that only appears in $e^{i\lambda \cdot q}$ and acts as $e^{i\lambda \cdot q} |\mu\rangle = |\lambda + \mu\rangle$ where $|\mu\rangle$ denotes a momentum eigenstate ($\mu \in \Lambda$). Since the NS sector consists of local operators, it is required to satisfy the mutual locality

$$V_\lambda(z) V_\mu(w) = \varepsilon \, V_\mu(w) V_\lambda(z), \tag{16}$$

where $\varepsilon = -1$ if both of the operators are fermionic, and $\varepsilon = +1$ otherwise: $\varepsilon = (-1)^{\lambda^2 \mu^2}$. To respect this condition, we need to introduce a "cocycle" factor $\sigma_\lambda$ satisfying

$$\hat{\sigma}_\lambda \hat{\sigma}_\mu = \varepsilon \, (-1)^{\lambda \cdot \mu} \hat{\sigma}_\mu \hat{\sigma}_\lambda, \tag{17}$$

where we conventionally take $\hat{\sigma}_\lambda = \sigma_\lambda \, e^{i\lambda \cdot q}$ [25]. Thus, the precise definition of vertex operators satisfying the mutual locality is

$$V_\lambda(z) =: e^{i\lambda \cdot X(z)} : \sigma_\lambda. \tag{18}$$

The cocycle factor satisfies the following multiplication rule

$$\hat{\sigma}_\lambda \hat{\sigma}_\mu = \epsilon(\lambda, \mu) \hat{\sigma}_{\lambda+\mu}, \tag{19}$$

where $\epsilon(\lambda, \mu) \in \mathbb{C}$. The coefficient $\epsilon$ has to satisfy some conditions. First, the condition (16) related to the mutual locality requires

$$\epsilon(\lambda, \mu) = (-1)^{\lambda \cdot \mu + \lambda^2 \mu^2} \epsilon(\mu, \lambda). \tag{20}$$

Second, since $\lambda = 0$ represents the identity operator, we have the condition

$$\epsilon(\lambda, 0) = \epsilon(0, \lambda) = 1. \tag{21}$$

Finally, the associativity of vertex operators: $(V_\alpha V_\beta) V_\gamma = V_\alpha (V_\beta V_\gamma)$ imposes the condition

$$\epsilon(\alpha, \beta) \epsilon(\alpha + \beta, \gamma) = \epsilon(\alpha, \beta + \gamma) \epsilon(\beta, \gamma), \tag{22}$$

which means that $\epsilon$ is a 2-cocycle of $\Lambda$ as an additive group.

Taking the cocycle factors into account, we consider the operator product expansion (OPE) between vertex operators $V_\lambda(z) V_\mu(w)$ where $\lambda^2 = \mu^2 = 2$ since this type of OPE appears in a later section. The singular terms appear only when $\lambda \cdot \mu = -1, -2$ because $\lambda^2 = \mu^2 = 2$. When $\lambda \cdot \mu = -2$, we have $\mu = -\lambda$ and the OPE becomes

$$V_\lambda(z) V_\mu(w) \sim \frac{\epsilon(\lambda, -\lambda)}{(z-w)^2} \left[ 1 + i(z-w) \lambda \cdot \partial X(w) \right], \tag{23}$$

where we used the multiplication rule (19) of the cocycle factor. When $\lambda \cdot \mu = -1$, the OPE is given by

$$V_\lambda(z) V_\mu(w) \sim \frac{\epsilon(\lambda, \mu)}{z-w} V_{\lambda+\mu}(w). \tag{24}$$

From the state-operator mapping, the Hilbert space of the NS sector $\mathcal{H}_{\text{NS}}(\Lambda)$ is spanned by

$$\prod_{i=1}^{n} \prod_{m=1}^{\infty} (\alpha^i_{-m})^{N_{im}} |\lambda\rangle, \quad \lambda \in \Lambda, \tag{25}$$

where $\alpha^i_m, \, i = \{1, \dots, n\}, \, m \in \mathbb{Z}$ is the oscillator that satisfies $[\alpha^i_m, \alpha^j_k] = m \delta_{m,-k} \delta^{i,j}$ and $N_{im} \in \mathbb{Z}_{\geq 0}$ is the occupation number for each mode. The fermion parity $(-1)^F$ acts on states as $(-1)^{\lambda^2}$, which equals $(-1)^{\chi \cdot \lambda}$ for any characteristic vector $\chi \in \Lambda$.

Similarly, the Hilbert space of the R sector $\mathcal{H}_R(\Lambda)$ is spanned by

$$\prod_{i=1}^{n}\prod_{m=1}^{\infty}(\alpha^i_{-m})^{N_{im}}|\xi\rangle\,,\quad \xi\in S(\Lambda)\,. \tag{26}$$

For the R sector, there is ambiguity in the fermion parity $(-1)^F$ and in this paper we fix a specific characteristic vector $\chi\in\Lambda$ and define $(-1)^F|\xi\rangle=(-1)^{\chi\cdot\lambda}|\xi\rangle$ where $\xi=\lambda+\frac{\chi}{2}$, $\lambda\in\Lambda$.

Let us consider the torus partition functions. For a fermionic CFT, the torus has spin structures specified by spatial and timelike boundary conditions. With the notation introduced in section 2, the partition functions of the CFT $\mathcal{T}$ constructed from the lattice $\Lambda\subset\mathbb{R}^n$ are

$$
\begin{aligned}
Z_\mathcal{T}[0,0] &= \mathrm{Tr}_{\mathcal{H}_{NS}}\left[q^{L_0-\frac{n}{24}}\right] &&= \frac{1}{\eta(\tau)^n}\sum_{\lambda\in\Lambda}q^{\frac{1}{2}\lambda^2}\,,\\
Z_\mathcal{T}[1,0] &= \mathrm{Tr}_{\mathcal{H}_{NS}}\left[(-1)^F\,q^{L_0-\frac{n}{24}}\right] &&= \frac{1}{\eta(\tau)^n}\sum_{\lambda\in\Lambda}(-1)^{\chi\cdot\lambda}q^{\frac{1}{2}\lambda^2}\,,\\
Z_\mathcal{T}[0,1] &= \mathrm{Tr}_{\mathcal{H}_R}\left[q^{L_0-\frac{n}{24}}\right] &&= \frac{1}{\eta(\tau)^n}\sum_{\lambda\in\Lambda}q^{\frac{1}{2}(\lambda+\frac{\chi}{2})^2}\,,\\
Z_\mathcal{T}[1,1] &= \mathrm{Tr}_{\mathcal{H}_R}\left[(-1)^F\,q^{L_0-\frac{n}{24}}\right] &&= \frac{1}{\eta(\tau)^n}\sum_{\lambda\in\Lambda}(-1)^{\chi\cdot\lambda}q^{\frac{1}{2}(\lambda+\frac{\chi}{2})^2}\,,
\end{aligned}
\tag{27}
$$

where $L_0$ is the Virasoro generator, $\eta(\tau)$ is the Dedekind eta function, and $\chi$ is a characteristic vector. Note that there is the ambiguity of the overall sign in $Z_\mathcal{T}[1,1]$, which comes from that of the fermion parity $(-1)^F$. Using the lattice theta function $\Theta_\Lambda(\tau)=\sum_{\lambda\in\Lambda}q^{\lambda^2/2}$, the NS-NS partition function can be written as $Z_\mathcal{T}[0,0]=\Theta_\Lambda(\tau)/\eta(\tau)^n$.

In the rest of this section, we discuss two types of orbifolds for the lattice CFT: the shift orbifold by $h:X\to X+\pi\delta$ and the reflection orbifold by $g:X\to -X$.

## 3.2 Shift orbifold

This section is devoted to the orbifold of a fermionic lattice CFT by a shift $\mathbb{Z}_2$ symmetry $H=\{1,h\}$ generated by a half shift $h:X\to X+\pi\delta$. In section 3.2.1, we describe the Hilbert space extended by the shift $\mathbb{Z}_2$ symmetry $H$. The extended Hilbert space consists of the untwisted and twisted sectors as in the top of table 1. The shift orbifold theory $\mathcal{T}/H$ is given by swapping those sectors. In lattice CFTs, we can interpret the shift orbifold as a modification of lattice, which will be explained in section 3.2.2.

### 3.2.1 $\mathbb{Z}_2$-extended Hilbert space

Let $\Lambda\subset\mathbb{R}^n$ be an odd self-dual lattice and $\delta\in\Lambda$ a vector that is not a characteristic vector and satisfies $\frac{1}{2}\delta\notin\Lambda$. We consider the theory on the orbifold obtained by the shift $h:X\to X+\pi\delta$. Note that if $\delta$ is a characteristic vector, then the orbifold theory can be constructed by simply swapping four sectors (NS/R and boson/fermion) as described in [26], and if $\frac{1}{2}\delta\in\Lambda$, then the shift $h$ is trivial.

Under the shift $\mathbb{Z}_2$ symmetry, the vertex operators are classified into $\mathbb{Z}_2$ even and odd sectors. In terms of the momentum lattice $\Lambda$, the inner product with $\delta$ specifies the $\mathbb{Z}_2$ grading of the corresponding operators. It is convenient to define

$$\Lambda_{\delta\text{-even}}=\{\lambda\in\Lambda\mid\delta\cdot\lambda\in 2\mathbb{Z}\}\,,\qquad \Lambda_{\delta\text{-odd}}=\{\lambda\in\Lambda\mid\delta\cdot\lambda\in 2\mathbb{Z}+1\}\,. \tag{28}$$

The vertex operators $V_\lambda(z)$ are even when $\lambda\in\Lambda_{\delta\text{-even}}$ and odd when $\lambda\in\Lambda_{\delta\text{-odd}}$. Since the oscillator excitations are bosonic, the untwisted NS Hilbert space can be decomposed into

$\mathcal{H}_{\text{NS}}(\Lambda) = \mathcal{H}_{\text{NS}}^{+h} \oplus \mathcal{H}_{\text{NS}}^{-h}$ where

$$\mathcal{H}_{\text{NS}}^{+h} = \left\{ \prod_{i=1}^{n} \prod_{m=1}^{\infty} (\alpha_{-m}^{i})^{N_{im}} |\lambda\rangle \ \middle| \ \lambda \in \Lambda_{\delta\text{-even}} \right\},$$

$$\mathcal{H}_{\text{NS}}^{-h} = \left\{ \prod_{i=1}^{n} \prod_{m=1}^{\infty} (\alpha_{-m}^{i})^{N_{im}} |\lambda\rangle \ \middle| \ \lambda \in \Lambda_{\delta\text{-odd}} \right\}. \tag{29}$$

We define $\mathbb{Z}_2$ grading for the vertex operators in the untwisted R sector by

$$h : V_{\lambda + \frac{\chi}{2}}(z) \to (-1)^{\delta \cdot \lambda} V_{\lambda + \frac{\chi}{2}}(z),$$

and the R sector decomposes into $\mathcal{H}_{\text{R}}(\Lambda) = \mathcal{H}_{\text{R}}^{+h} \oplus \mathcal{H}_{\text{R}}^{-h}$ where

$$\mathcal{H}_{\text{R}}^{+h} = \left\{ \prod_{i=1}^{n} \prod_{m=1}^{\infty} (\alpha_{-m}^{i})^{N_{im}} |\lambda + \tfrac{\chi}{2}\rangle \ \middle| \ \lambda \in \Lambda_{\delta\text{-even}} \right\},$$

$$\mathcal{H}_{\text{R}}^{-h} = \left\{ \prod_{i=1}^{n} \prod_{m=1}^{\infty} (\alpha_{-m}^{i})^{N_{im}} |\lambda + \tfrac{\chi}{2}\rangle \ \middle| \ \lambda \in \Lambda_{\delta\text{-odd}} \right\}. \tag{30}$$

Let us detect its 't Hooft anomaly by using the technique introduced around (5). From direct calculation using the expression

$$Z_{\mathcal{T}}[0, 0; 1, 0] = \frac{1}{\eta(\tau)^n} \sum_{\lambda \in \Lambda} (-1)^{\lambda \cdot \delta} q^{\frac{1}{2}\lambda^2}, \tag{31}$$

the phase from the modular transformation $ST^2S^{-1}$ on $Z_{\mathcal{T}}[0, 0; 1, 0]/Z_{\mathcal{T}}[0, 0; 0, 0]$ is $e^{2\pi i \nu/8}$ with $\nu = 2\delta^2$. Thus, we can conclude that the shift orbifold theory is non-anomalous if and only if $\delta^2 \in 4\mathbb{Z}$. This condition is required to be satisfied for a consistent definition of the orbifold theory.

As mentioned earlier, we take $\delta$ that is not a characteristic vector and satisfies $\frac{1}{2}\delta \notin \Lambda$. Equivalently, $\delta$ satisfies $\frac{\delta}{2} \notin (\Lambda \sqcup S(\Lambda))$. Combining with the condition $\delta^2 \in 4\mathbb{Z}$ for non-anomalous $\delta$, we call the gauging condition ($\frac{\delta}{2} \notin (\Lambda \sqcup S(\Lambda))$ and $\delta^2 \in 4\mathbb{Z}$).

For the shift symmetry, we can construct the twisted sector by shifting the original momentum lattice $\Lambda$ with the half lattice element $\delta/2$. The vertex operators in the twisted NS sector can be expressed as

$$V_{\lambda + \frac{\delta}{2}}(z) =: e^{i(\lambda + \frac{\delta}{2}) \cdot X(z)} : \quad (\lambda \in \Lambda). \tag{32}$$

Note that the cocycle factor is omitted in this section as it does not affect the results. The operators are identified to be in the twisted sector by considering the operator product expansion with $V_{\lambda'}(w)$ in the untwisted NS sector and circling one operator around the other. Then we obtain the phase

$$V_{\lambda + \frac{\delta}{2}}(z) V_{\lambda'}(w) \to (-1)^{\delta \cdot \lambda'} V_{\lambda + \frac{\delta}{2}}(z) V_{\lambda'}(w). \tag{33}$$

This implies that the vertex operators $V_{\lambda + \frac{\delta}{2}}(z)$ are lying on a line operator implementing the shift $\mathbb{Z}_2$ symmetry, from which we can see that they are operators in the twisted sector. Again, since the oscillators are bosonic, the twisted NS sector is spanned by

$$\prod_{i=1}^{n} \prod_{m=1}^{\infty} (\alpha_{-m}^{i})^{N_{im}} |\lambda + \tfrac{\delta}{2}\rangle, \tag{34}$$

Table 2: The action of the shift symmetry $H = \{1, h\}$ and the fermion parity $(-1)^F$ for the vertex operators in each sector.

| | untwisted | | twisted | |
| | NS | R | NS | R |
| | $V_\lambda(z)$ | $V_{\lambda+\frac{\chi}{2}}(z)$ | $V_{\lambda+\frac{\delta}{2}}(z)$ | $V_{\lambda+\frac{\delta}{2}+\frac{\chi}{2}}(z)$ |
|---|---|---|---|---|
| shift symmetry $h$ | $(-1)^{\delta\cdot\lambda}$ | $(-1)^{\delta\cdot\lambda}$ | $(-1)^{\delta\cdot(\lambda+\frac{\delta}{2})}$ | $(-1)^{\delta\cdot(\lambda+\frac{\delta}{2})}$ |
| fermion parity $(-1)^F$ | $(-1)^{\chi\cdot\lambda}$ | $(-1)^{\chi\cdot\lambda}$ | $(-1)^{\chi\cdot(\lambda+\frac{\delta}{2})}$ | $(-1)^{\chi\cdot(\lambda+\frac{\delta}{2})}$ |

where $N_{im}$ is the occupation number for each mode. As in the untwisted case, the twisted R sector consists of vertex operators with momenta shifted by the half characteristic vector $\chi/2$. Thus, the vertex operators in the twisted R sector take the form

$$V_{\lambda+\frac{\delta}{2}+\frac{\chi}{2}}(z) =: e^{i(\lambda+\frac{\delta}{2}+\frac{\chi}{2})\cdot X(z)}: \quad (\lambda \in \Lambda). \tag{35}$$

Here, we can rewrite $\lambda + \frac{\chi}{2}$ by using an element $\xi$ of the shadow $S(\Lambda)$: $\xi = \lambda + \frac{\chi}{2}$. The twisted R sector is spanned by

$$\prod_{i=1}^{n}\prod_{m=1}^{\infty}(\alpha_{-m}^i)^{N_{im}} |\xi + \tfrac{\delta}{2}\rangle\,, \tag{36}$$

where $\xi \in S(\Lambda)$ and $N_{im}$ is the occupation number for each mode.

To define the orbifold, we need to specify the action of the fermion parity and the $\mathbb{Z}_2$ symmetry in the twisted Hilbert space. As in the untwisted sector, we define the $\mathbb{Z}_2$ symmetry on the vertex operators in the twisted sector by

$$h: \quad V_{\lambda+\frac{\delta}{2}}(z) \mapsto (-1)^{\delta\cdot(\lambda+\frac{\delta}{2})} V_{\lambda+\frac{\delta}{2}}(z), \qquad V_{\lambda+\frac{\delta}{2}+\frac{\chi}{2}}(z) \mapsto (-1)^{\delta\cdot(\lambda+\frac{\delta}{2})} V_{\lambda+\frac{\delta}{2}+\frac{\chi}{2}}(z), \tag{37}$$

where $\lambda \in \Lambda$. Note that $(-1)^{\delta\cdot(\lambda+\frac{1}{2}\delta)} = (-1)^{\delta\cdot\lambda}$ from $\delta^2 \in 4\mathbb{Z}$. Additionally, we assume that the fermion parity acts on the twisted Hilbert space as

$$(-1)^F: \quad V_{\lambda+\frac{\delta}{2}}(z) \mapsto (-1)^{\chi\cdot(\lambda+\frac{\delta}{2})} V_{\lambda+\frac{\delta}{2}}(z), \qquad V_{\lambda+\frac{\delta}{2}+\frac{\chi}{2}}(z) \mapsto (-1)^{\chi\cdot(\lambda+\frac{\delta}{2})} V_{\lambda+\frac{\delta}{2}+\frac{\chi}{2}}(z). \tag{38}$$

In table 2, we summarize the action of the shift symmetry and the fermion parity.

To get the spin selection rule for each sector, we take a characteristic vector $\chi \in \Lambda$ of the original momentum lattice $\Lambda$ such that

$$\frac{\chi\cdot\delta}{2} = \frac{\delta^2}{4} \mod 2\,. \tag{39}$$

This is always possible since for a characteristic vector $\chi$, a vector $\chi + 2\lambda_o$ $(\lambda_o \in \Lambda_{\delta\text{-odd}})$ is also characteristic, but it has a different mod 2 value of the inner product with $\frac{\delta}{2}$. Then, we obtain the spin selection rule for each sector extended by the shift symmetry in lattice CFTs in table 3. Note that using the fact that a characteristic vector $\chi$ of a self-dual lattice $\Lambda \subset \mathbb{R}^n$ satisfies $\chi^2 \equiv n \mod 8$ [27, 28], we obtain the spin of the R sector $s \in n/8 + \mathbb{Z}$ except for $s \in \frac{1}{2} + \frac{n}{8} + \mathbb{Z}$ for the $h$-odd states in the twisted R sector.

### 3.2.2 Shift orbifold as lattice CFT

Let us consider the orbifold by the shift symmetry $H = \{1, h\}$.

Table 3: The spin selection rule for the shift symmetry $H = \{1, h\}$ in lattice CFT.

|  |  | untwisted | | twisted | |
|---|---|:---:|:---:|:---:|:---:|
|  |  | NS | R | NS | R |
| $h$-even | boson | $\mathbb{Z}$ | $\frac{n}{8} + \mathbb{Z}$ | $\mathbb{Z}$ | $\frac{n}{8} + \mathbb{Z}$ |
|  | fermion | $\frac{1}{2} + \mathbb{Z}$ | $\frac{n}{8} + \mathbb{Z}$ | $\frac{1}{2} + \mathbb{Z}$ | $\frac{n}{8} + \mathbb{Z}$ |
| $h$-odd | boson | $\mathbb{Z}$ | $\frac{n}{8} + \mathbb{Z}$ | $\frac{1}{2} + \mathbb{Z}$ | $\frac{1}{2} + \frac{n}{8} + \mathbb{Z}$ |
|  | fermion | $\frac{1}{2} + \mathbb{Z}$ | $\frac{n}{8} + \mathbb{Z}$ | $\mathbb{Z}$ | $\frac{1}{2} + \frac{n}{8} + \mathbb{Z}$ |

By following table 1, two orbifold theories $(\mathcal{T}/H)_{\pm}$ can be obtained. We will see that they are interpreted as CFTs constructed from other lattices. The NS sectors of the orbifold theories consist of

$$
\begin{aligned}
(+) &: h\text{-even untwisted NS} + h\text{-even twisted NS}, \\
(-) &: h\text{-even untwisted NS} + h\text{-odd twisted NS},
\end{aligned}
\tag{40}
$$

thus the momentum lattices are

$$
\Lambda_{\delta}^{\text{orb}+} = \Lambda_{\delta\text{-even}} \sqcup \left(\Lambda_{\delta\text{-even}} + \tfrac{1}{2}\delta\right), \qquad \Lambda_{\delta}^{\text{orb}-} = \Lambda_{\delta\text{-even}} \sqcup \left(\Lambda_{\delta\text{-odd}} + \tfrac{1}{2}\delta\right).
\tag{41}
$$

The Hilbert spaces of the NS and R sectors and the torus partition functions with the spin structure can be described with these lattices and their shadows as in section 3.1.

To ensure the consistency of the orbifold theory, we show that $\Lambda_{\delta}^{\text{orb}\pm}$ is an odd self-dual lattice. This guarantees that the orbifold partition functions covariantly transform under the modular transformation.

**Proposition 3.1**
Let $\Lambda \subset \mathbb{R}^n$ be an odd self-dual lattice and $\delta \in \Lambda$ a vector satisfying the gauging condition: $\frac{\delta}{2} \notin (\Lambda \sqcup S(\Lambda))$ and $\delta^2 \in 4\mathbb{Z}$. Then $\Lambda_{\delta}^{\text{orb}\pm}$ defined in (41) is odd self-dual.

*Proof.* The self-orthogonality is obvious from

$$
\begin{aligned}
\lambda_e \cdot (\lambda + \tfrac{1}{2}\delta) &\equiv \tfrac{1}{2}\lambda_e \cdot \delta \equiv 0 \mod 1, \\
(\lambda + \tfrac{1}{2}\delta) \cdot (\lambda' + \tfrac{1}{2}\delta) &\equiv \tfrac{1}{2}(\lambda + \lambda') \cdot \delta + \tfrac{1}{4}\delta^2 \equiv 0 \mod 1,
\end{aligned}
\tag{42}
$$

where $\lambda_e \in \Lambda_{\delta\text{-even}}$ and $\lambda, \lambda'$ are in $\Lambda_{\delta\text{-even}}$ for $\Lambda_{\delta}^{\text{orb}+}$ and in $\Lambda_{\delta\text{-odd}}$ for $\Lambda_{\delta}^{\text{orb}-}$. By combining with the fact that the volume of the fundamental region of $\Lambda_{\delta}^{\text{orb}\pm}$ is equal to that of $\Lambda$ from the construction, i.e., $d(\Lambda_{\delta}^{\text{orb}\pm}) = d(\Lambda) = 1$, the self-orthogonality leads to the self-duality.

Next, we prove that there exists $\lambda \in \Lambda_{\delta\text{-even}}$ s.t. $\lambda^2 \equiv 1 \mod 2$. If any $\lambda_e \in \Lambda_{\delta\text{-even}}$ satisfies $\lambda_e^2 \equiv 0$, then there exists $\lambda' \in \Lambda_{\delta\text{-odd}}$ s.t. $\lambda'^2 \equiv 1$ since the lattice $\Lambda$ is odd. By using this $\lambda'$, any $\lambda_o \in \Lambda_{\delta\text{-odd}}$ can be written as $\lambda_o = \lambda' + \lambda_e$, $\lambda_e \in \Lambda_{\delta\text{-even}}$, thus its norm is odd from $\lambda_o^2 \equiv \lambda'^2 + \lambda_e^2 \equiv 1$. However, this means that $\delta$ is a characteristic vector, which contradicts our assumption $\frac{\delta}{2} \notin S(\Lambda)$.

From $\Lambda_{\delta\text{-even}} \subset \Lambda_{\delta}^{\text{orb}\pm}$, we can conclude that $\Lambda_{\delta}^{\text{orb}\pm}$ is odd self-dual. $\qquad \square$

It is clear from the definition that $\Lambda_{\delta}^{\text{orb}\pm} = \Lambda_{\delta+2\lambda_e}^{\text{orb}\pm}$ (double sign in same order) for any $\lambda_e \in \Lambda_{\delta\text{-even}}$ and thus the orbifold theories by the shift $\delta$ and $\delta + 2\lambda_e$ are equivalent. It can be easily checked that when $\delta$ satisfies the gauging condition, so does $\delta + 2\lambda_e$. Similarly, $\Lambda_{\delta+2\lambda_o}^{\text{orb}\pm} = \Lambda_{\delta}^{\text{orb}\mp}$ for any $\lambda_o \in \Lambda_{\delta\text{-odd}}$ since $\Lambda_{\delta\text{-even}} + \lambda_o = \Lambda_{\delta\text{-odd}}$.

To define the fermion parity as the original theory, the characteristic vector must be fixed. We take $\chi^{\text{orb}+} = \chi$ for $\Lambda_{\delta}^{\text{orb}+}$ and $\chi^{\text{orb}-} = \chi + \delta$ for $\Lambda_{\delta}^{\text{orb}-}$, which is justified by the following proposition.

Table 4: The shift orbifold in terms of lattices. After orbifolding, the shift vector $2\lambda_o$ for $(\mathcal{T}/H)_+$ and $\delta$ for $(\mathcal{T}/H)_-$ is the generator of the dual symmetry. Note that even or odd refers to the action of $h$ ($\check{h}$) on the vertex operator, not the inner product itself.

(a) Original.

| | untwisted | | twisted | |
|---|---|---|---|---|
| | NS | R | NS | R |
| | $\Lambda$ | $S(\Lambda)$ | $\Lambda+\frac{\delta}{2}$ | $S(\Lambda)+\frac{\delta}{2}$ |
| $\delta$-even | $\Lambda_{\delta\text{-even}}$ | $\Lambda_{\delta\text{-even}}+\frac{\chi}{2}$ | $\Lambda_{\delta\text{-even}}+\frac{\delta}{2}$ | $\Lambda_{\delta\text{-even}}+\frac{\delta}{2}+\frac{\chi}{2}$ |
| $\delta$-odd | $\Lambda_{\delta\text{-odd}}$ | $\Lambda_{\delta\text{-odd}}+\frac{\chi}{2}$ | $\Lambda_{\delta\text{-odd}}+\frac{\delta}{2}$ | $\Lambda_{\delta\text{-odd}}+\frac{\delta}{2}+\frac{\chi}{2}$ |

(b) Orbifold (+).

| | untwisted | | twisted | |
|---|---|---|---|---|
| | NS | R | NS | R |
| | $\Lambda_\delta^{\text{orb}+}$ | $S(\Lambda_\delta^{\text{orb}+})$ | $\Lambda_\delta^{\text{orb}+}+\lambda_o$ | $S(\Lambda_\delta^{\text{orb}+})+\lambda_o$ |
| $2\lambda_o$-even | $\Lambda_{\delta\text{-even}}$ | $\Lambda_{\delta\text{-even}}+\frac{\chi}{2}$ | $\Lambda_{\delta\text{-odd}}$ | $\Lambda_{\delta\text{-odd}}+\frac{\chi}{2}$ |
| $2\lambda_o$-odd | $\Lambda_{\delta\text{-even}}+\frac{\delta}{2}$ | $\Lambda_{\delta\text{-even}}+\frac{\delta}{2}+\frac{\chi}{2}$ | $\Lambda_{\delta\text{-odd}}+\frac{\delta}{2}$ | $\Lambda_{\delta\text{-odd}}+\frac{\delta}{2}+\frac{\chi}{2}$ |

(c) Orbifold (−).

| | untwisted | | twisted | |
|---|---|---|---|---|
| | NS | R | NS | R |
| | $\Lambda_\delta^{\text{orb}-}$ | $S(\Lambda_\delta^{\text{orb}-})$ | $\Lambda_\delta^{\text{orb}-}+\frac{\delta}{2}$ | $S(\Lambda_\delta^{\text{orb}-})+\frac{\delta}{2}$ |
| $\delta$-even | $\Lambda_{\delta\text{-even}}$ | $\Lambda_{\delta\text{-even}}+\frac{\delta}{2}+\frac{\chi}{2}$ | $\Lambda_{\delta\text{-even}}+\frac{\delta}{2}$ | $\Lambda_{\delta\text{-even}}+\frac{\chi}{2}$ |
| $\delta$-odd | $\Lambda_{\delta\text{-odd}}+\frac{\delta}{2}$ | $\Lambda_{\delta\text{-odd}}+\frac{\chi}{2}$ | $\Lambda_{\delta\text{-odd}}$ | $\Lambda_{\delta\text{-odd}}+\frac{\delta}{2}+\frac{\chi}{2}$ |

**Proposition 3.2**

We adopt the same conventions as in Proposition 3.1. Given a characteristic vector $\chi$ of $\Lambda$ that satisfies (39), then the vectors

$$\chi^{\text{orb}+} = \chi\,, \qquad \chi^{\text{orb}-} = \chi + \delta\,, \tag{43}$$

are characteristic vectors of $\Lambda_\delta^{\text{orb}+}$ and $\Lambda_\delta^{\text{orb}-}$, respectively.

*Proof.* For $\Lambda_\delta^{\text{orb}+}$, any $\lambda_e \in \Lambda_{\delta\text{-even}}$ satisfies

$$
\begin{aligned}
\lambda_e^2 &\equiv \chi \cdot \lambda_e \quad \text{mod } 2\,, \\
(\lambda_e + \tfrac{1}{2}\delta)^2 &\equiv \lambda_e^2 + \tfrac{1}{4}\delta^2 \equiv \chi \cdot \lambda_e + \tfrac{1}{2}\chi \cdot \delta \equiv \chi \cdot (\lambda_e + \tfrac{1}{2}\delta) \quad \text{mod } 2\,.
\end{aligned}
\tag{44}
$$

For $\Lambda_\delta^{\text{orb}-}$, any $\lambda_e \in \Lambda_{\delta\text{-even}}$, $\lambda_o \in \Lambda_{\delta\text{-odd}}$ satisfies

$$
\begin{aligned}
\lambda_e^2 &\equiv \chi \cdot \lambda_e \equiv (\chi + \delta) \cdot \lambda_e \quad \text{mod } 2\,, \\
(\lambda_o + \tfrac{1}{2}\delta)^2 &\equiv \lambda_o^2 + \tfrac{1}{4}\delta^2 + 1 \equiv \chi \cdot \lambda_o + \tfrac{1}{2}\chi \cdot \delta + 1 \equiv (\chi + \delta) \cdot (\lambda_o + \tfrac{1}{2}\delta) \quad \text{mod } 2\,.
\end{aligned}
\tag{45}
$$

$\square$

In table 1, the fermion parity is flipped for states in $h$-odd sectors of $(\mathcal{T}/H)_-$. In terms of lattices, this comes from $\delta$ in $\chi^{\text{orb}-}$.

It can be easily shown that $(\Lambda_\delta^{\text{orb}+})_{2\lambda_o}^{\text{orb}+} = \Lambda$ for any $\lambda_o \in \Lambda_{\delta\text{-odd}}$ and $(\Lambda_\delta^{\text{orb}-})_\delta^{\text{orb}-} = \Lambda$, thus the shifts by $2\lambda_o$ for $(\mathcal{T}/H)_+$ and $\delta$ for $(\mathcal{T}/H)_-$ can be regarded as the dual operations $\check{h}$. In the orbifold theories, these vectors and the characteristic vectors also satisfy the condition (39) as

$$\frac{\chi^{\text{orb}+} \cdot 2\lambda_o}{2} \equiv \frac{(2\lambda_o)^2}{4} \mod 2, \qquad \frac{\chi^{\text{orb}-} \cdot \delta}{2} \equiv \frac{\delta^2}{4} \mod 2. \tag{46}$$

Therefore, the orbifolded theories have the same spin selection rule as in the original theory.

From (41), (43) and $\delta \in \Lambda_{\delta\text{-even}}$, the shadows can be expressed as

$$S(\Lambda_\delta^{\text{orb}+}) = \Lambda_\delta^{\text{orb}+} + \tfrac{1}{2}\chi^{\text{orb}+} = \left(\Lambda_{\delta\text{-even}} + \tfrac{1}{2}\chi\right) \sqcup \left(\Lambda_{\delta\text{-even}} + \tfrac{1}{2}(\chi + \delta)\right), \tag{47}$$

$$S(\Lambda_\delta^{\text{orb}-}) = \Lambda_\delta^{\text{orb}-} + \tfrac{1}{2}\chi^{\text{orb}-} = \left(\Lambda_{\delta\text{-even}} + \tfrac{1}{2}(\chi + \delta)\right) \sqcup \left(\Lambda_{\delta\text{-odd}} + \tfrac{1}{2}\chi\right). \tag{48}$$

We have seen that the original and orbifold theories are lattice CFTs and consist of sets of vertex operators. The corresponding momentum lattices are summarized in table 4.

For later convenience, we write general expressions of the shadow including the case where the fixed characteristic vector $\chi$ of $\Lambda$ does not satisfy the condition (39):

$$S(\Lambda_\delta^{\text{orb}+}) = \begin{cases} \left(\Lambda_{\delta\text{-even}} + \tfrac{1}{2}\chi\right) \sqcup \left(\Lambda_{\delta\text{-even}} + \tfrac{1}{2}(\chi + \delta)\right) & \left(\tfrac{1}{2}\chi \cdot \delta \equiv \tfrac{1}{4}\delta^2 \mod 2\right), \\ \left(\Lambda_{\delta\text{-odd}} + \tfrac{1}{2}\chi\right) \sqcup \left(\Lambda_{\delta\text{-odd}} + \tfrac{1}{2}(\chi + \delta)\right) & (\text{otherwise}), \end{cases} \tag{49}$$

$$= \begin{cases} S(\Lambda)_{\delta\text{-even}} \sqcup \left(S(\Lambda)_{\delta\text{-even}} + \tfrac{1}{2}\delta\right) & \left(\tfrac{1}{4}\delta^2 \equiv 0 \mod 2\right), \\ S(\Lambda)_{\delta\text{-odd}} \sqcup \left(S(\Lambda)_{\delta\text{-odd}} + \tfrac{1}{2}\delta\right) & (\text{otherwise}), \end{cases} \tag{50}$$

$$S(\Lambda_\delta^{\text{orb}-}) = \begin{cases} \left(\Lambda_{\delta\text{-even}} + \tfrac{1}{2}(\chi + \delta)\right) \sqcup \left(\Lambda_{\delta\text{-odd}} + \tfrac{1}{2}\chi\right) & \left(\tfrac{1}{2}\chi \cdot \delta \equiv \tfrac{1}{4}\delta^2 \mod 2\right), \\ \left(\Lambda_{\delta\text{-even}} + \tfrac{1}{2}\chi\right) \sqcup \left(\Lambda_{\delta\text{-odd}} + \tfrac{1}{2}(\chi + \delta)\right) & (\text{otherwise}), \end{cases} \tag{51}$$

$$= \begin{cases} \left(S(\Lambda)_{\delta\text{-even}} + \tfrac{1}{2}\delta\right) \sqcup S(\Lambda)_{\delta\text{-odd}} & \left(\tfrac{1}{4}\delta^2 \equiv 0 \mod 2\right), \\ S(\Lambda)_{\delta\text{-even}} \sqcup \left(S(\Lambda)_{\delta\text{-odd}} + \tfrac{1}{2}\delta\right) & (\text{otherwise}), \end{cases} \tag{52}$$

where $S(\Lambda)_{\delta\text{-even(odd)}} = \{\xi \in S(\Lambda) \mid \delta \cdot \xi \text{ is even(odd)}\}$.

## 3.3 Reflection orbifold

This section is devoted to the orbifold of lattice CFT by the reflection symmetry $g : X \to -X$. In the bosonic case, the reflection orbifold in lattice CFTs was carefully analyzed in [29]. Our interest is chiral fermionic CFTs constructed from odd self-dual lattices. The reflection symmetry $G = \{1, g\}$ acts as

$$g\,\alpha_n^j\,g^{-1} = -\alpha_n^j, \qquad g\,|\lambda\rangle = |-\lambda\rangle, \tag{53}$$

where $\alpha_n^j$ is a bosonic oscillator and $|\lambda\rangle$ denotes a momentum eigenstate ($\lambda \in \Lambda$). Under this $\mathbb{Z}_2$ symmetry, we can decompose the NS sector as $\mathcal{H}_{\text{NS}}(\Lambda) = \mathcal{H}_{\text{NS}}^{+g} \oplus \mathcal{H}_{\text{NS}}^{-g}$ where the $\mathbb{Z}_2$ even and odd sectors are

$$\mathcal{H}_{\text{NS}}^{+g} = \left\{\alpha_{-n_1}^{j_1} \cdots \alpha_{-n_{2k}}^{j_{2k}}(|\lambda\rangle + |-\lambda\rangle)\right\} \cup \left\{\alpha_{-n_1}^{j_1} \cdots \alpha_{-n_{2k+1}}^{j_{2k+1}}(|\lambda\rangle - |-\lambda\rangle)\right\},$$
$$\mathcal{H}_{\text{NS}}^{-g} = \left\{\alpha_{-n_1}^{j_1} \cdots \alpha_{-n_{2k+1}}^{j_{2k+1}}(|\lambda\rangle + |-\lambda\rangle)\right\} \cup \left\{\alpha_{-n_1}^{j_1} \cdots \alpha_{-n_{2k}}^{j_{2k}}(|\lambda\rangle - |-\lambda\rangle)\right\}. \tag{54}$$

Similarly, the R sector is decomposed into $\mathcal{H}_{\text{R}}(\Lambda) = \mathcal{H}_{\text{R}}^{+g} \oplus \mathcal{H}_{\text{R}}^{-g}$ where

$$\mathcal{H}_{\text{R}}^{+g} = \left\{\alpha_{-n_1}^{j_1} \cdots \alpha_{-n_{2k}}^{j_{2k}}(|\xi\rangle + |-\xi\rangle)\right\} \cup \left\{\alpha_{-n_1}^{j_1} \cdots \alpha_{-n_{2k+1}}^{j_{2k+1}}(|\xi\rangle - |-\xi\rangle)\right\},$$
$$\mathcal{H}_{\text{R}}^{-g} = \left\{\alpha_{-n_1}^{j_1} \cdots \alpha_{-n_{2k+1}}^{j_{2k+1}}(|\xi\rangle + |-\xi\rangle)\right\} \cup \left\{\alpha_{-n_1}^{j_1} \cdots \alpha_{-n_{2k}}^{j_{2k}}(|\xi\rangle - |-\xi\rangle)\right\}, \tag{55}$$

for $\xi \in S(\Lambda)$.

Before proceeding to the gauging of this $\mathbb{Z}_2$ symmetry, we need to diagnose whether this symmetry is anomalous or not. We consider the $g$-graded partition function

$$Z_{\mathcal{T}}[0,0;1,0] = \text{Tr}_{\mathcal{H}_{\text{NS}}(\Lambda)}\left[g\, q^{L_0 - \frac{n}{24}}\right] = \frac{q^{-\frac{n}{24}}}{\prod_{m=1}^{\infty}(1+q^m)^n}. \tag{56}$$

To compute its modular transformation, it is useful to rewrite it as

$$Z_{\mathcal{T}}[0,0;1,0] = \frac{\eta(\tau)^n}{\eta(2\tau)^n} = \left(\frac{2\eta(\tau)}{\theta_2(\tau)}\right)^{\frac{n}{2}}, \tag{57}$$

where $\theta_i(\tau)$ ($i = 2, 3, 4$) are the Jacobi theta functions. The modular transformation $ST^2S^{-1}$ acts on $Z_{\mathcal{T}}[0,0;1,0]/Z_{\mathcal{T}}[0,0;0,0]$ with the eigenvalue $e^{2\pi i n/8}$. Thus, the non-anomalous condition is $n \in 8\mathbb{Z}$. This condition is required to be satisfied for a consistent definition of the orbifold theory.

The theory can be quantized under the periodicity twisted by the $\mathbb{Z}_2$ symmetry $g$. Then, a chiral bosonic field $R^j(z)$ under the twisted periodicity admits the mode expansion

$$R^j(z) = \mathrm{i} \sum_{r \in \mathbb{Z}+1/2} \frac{c_r^j}{r} z^{-r}. \tag{58}$$

The oscillators with half-integer modes satisfy the commutation relation

$$[c_r^i, c_s^j] = r\,\delta^{i,j}\delta_{r+s,0}. \tag{59}$$

As in the straight lattice construction, we have vertex operators defined by

$$V_\lambda^T(z) =: e^{\mathrm{i}\lambda \cdot R(z)} : \gamma_\lambda, \tag{60}$$

where $\gamma_\lambda$ is a cocycle factor to respect the mutual locality.

Since a vertex operator $V_\lambda^T(z)$ acts on a twisted ground state $|a\rangle$ as $\gamma_\lambda |a\rangle$, the twisted ground states form a representation $\Upsilon(\Lambda)$ of gamma matrices $\gamma_\lambda$ ($\lambda \in \Lambda$) satisfying the algebra

$$\gamma_\lambda \gamma_\mu = \varepsilon\, (-1)^{\lambda \cdot \mu} \gamma_\mu \gamma_\lambda, \tag{61}$$

where $\varepsilon = -1$ if both $\lambda, \mu$ have an odd norm, and $\varepsilon = +1$ otherwise, so it can be written as $\varepsilon = (-1)^{\lambda^2 \mu^2}$. While this algebra is infinite-dimensional, the non-trivial part is given by $\Lambda/2\Lambda$ since $\gamma_\lambda$ for $\lambda \in 2\Lambda$ commutes with any element. Since the central elements of the algebra are only $\gamma_\lambda$ ($\lambda \in 2\Lambda$), following [29, Appendix C], we can show that the gamma matrices can be represented from the Dirac gamma matrices and thus it has a unique irreducible representation of dimension $2^{\frac{n}{2}}$. We denote its basis as $|a\rangle = |\pm \pm \cdots \pm\rangle$.

By the oscillator excitation, the twisted NS Hilbert space $\mathcal{H}_{\text{NS},g}$ is spanned by

$$\prod_{i=1}^{n} \prod_{r=1/2}^{\infty} (c_{-r}^i)^{N_{ir}} |a\rangle, \quad |a\rangle \in \Upsilon(\Lambda), \tag{62}$$

where $N_{ir}$ is occupation number for each mode.

Under the anti-periodic boundary condition, the ground states acquire the vacuum energy $E_0 = n/48$ on the cylindrical coordinate. Correspondingly, the Virasoro generators are

$$L_m = \frac{1}{2} \sum_{r=1/2}^{\infty} c_{m-r} \cdot c_r + \frac{n}{16}\delta_{m,0}. \tag{63}$$

This implies that the twisted ground states have the conformal weight $h = n/16$. Therefore, a general state (62) has

$$h = \sum_{i=1}^{n} \sum_{r=1/2}^{\infty} r N_{ir} + \frac{n}{16}. \tag{64}$$

The partition function for the twisted NS sector is given by

$$Z_{\mathcal{T}}[0,0;0,1] = \text{Tr}_{\mathcal{H}_{\text{NS},g}}\left[q^{L_0 - \frac{n}{24}}\right] = \frac{2^{\frac{n}{2}} q^{\frac{n}{48}}}{\prod_{m=1}^{\infty}(1 - q^{m-\frac{1}{2}})^n} = \left(\frac{2\eta}{\theta_4}\right)^{\frac{n}{2}}. \tag{65}$$

This partition function can be reproduced from (56) by applying the modular $S$ transformation.

We define the $\mathbb{Z}_2$ grading and the fermion parity in the twisted NS sector under the non-anomalous condition $n \in 8\mathbb{Z}$ for the reflection symmetry. Concretely, we propose to define the $\mathbb{Z}_2$ action $g$ on the twisted NS sector by

$$g\, c_r^i\, g^{-1} = -c_r^i, \qquad g\,|a\rangle = \begin{cases} +(-1)^{\frac{n}{8}}\,|a\rangle & (a : + \pm \cdots \pm), \\ -(-1)^{\frac{n}{8}}\,|a\rangle & (a : - \pm \cdots \pm). \end{cases} \tag{66}$$

Half of the ground states are even and the others are odd. Combining the oscillator excitation, we obtain the $\mathbb{Z}_2$ even and odd sectors in the twisted NS sector. For example, in the case of $n \in 16\mathbb{Z}$, each Hilbert space is given by

$$\mathcal{H}_{\text{NS},g}^{+_g} = \left\{c_{-r_1}^{j_1} \cdots c_{-r_{2k}}^{j_{2k}} |+\pm\cdots\pm\rangle\right\} \cup \left\{c_{-r_1}^{j_1} \cdots c_{-r_{2k+1}}^{j_{2k+1}} |-\pm\cdots\pm\rangle\right\},$$
$$\mathcal{H}_{\text{NS},g}^{-_g} = \left\{c_{-r_1}^{j_1} \cdots c_{-r_{2k+1}}^{j_{2k+1}} |+\pm\cdots\pm\rangle\right\} \cup \left\{c_{-r_1}^{j_1} \cdots c_{-r_{2k}}^{j_{2k}} |-\pm\cdots\pm\rangle\right\}. \tag{67}$$

Since the even and odd sectors have the same energy spectrum, the partition functions for the $g$-even and odd states are

$$\text{Tr}_{\mathcal{H}_{\text{NS},g}^{\pm_g}}\left[q^{L_0 - \frac{n}{24}}\right] = \frac{1}{2}\left(\frac{2\eta}{\theta_4}\right)^{\frac{n}{2}}. \tag{68}$$

In other words, $Z_{\mathcal{T}}[0,0;1,1] = 0$. Also, we need to introduce the fermion parity in the twisted NS sector. By using the modular transformation depicted in Fig. 2, we can see that $Z_{\mathcal{T}}[1,0;0,1]$ is vanishing and $Z_{\mathcal{T}}[1,0;1,1] \propto (2\eta/\theta_3)^{\frac{n}{2}}$ up to a phase factor. The results suggest that, under the fermion parity, half of the twisted ground states are odd, and the others are even. Furthermore, the diagonal action $(-1)^F g$ acts as a constant on the twisted ground states. Thus, the fermion parity on the twisted NS sector can be defined by

$$(-1)^F |a\rangle = \begin{cases} +|a\rangle & (a : + \pm \cdots \pm), \\ -|a\rangle & (a : - \pm \cdots \pm), \end{cases} \tag{69}$$

and the bosonic oscillators are invariant under the action of the fermion parity.

Now we move onto the $\mathbb{Z}_2$ grading and the fermion parity on the twisted R sector. Since we have the partition function $Z_{\mathcal{T}}[0,1;0,1]$ through the modular transformation from (56)

$$Z_{\mathcal{T}}[0,1;0,1] = \left(\frac{2\eta(\tau)}{\theta_4(\tau)}\right)^{\frac{n}{2}} = q^{\frac{n}{48}}\left(2^{\frac{n}{2}} + 2^{\frac{n}{2}} n \sqrt{q} + \dots\right). \tag{70}$$

We see that there are $2^{\frac{n}{2}}$ twisted R ground states $|\Omega_j\rangle$ ($j = 1, 2, \dots, 2^{\frac{n}{2}}$) with conformal weight $h = \frac{n}{16}$. The twisted R sector consists of the excited states by the bosonic oscillators

$$\prod_{i=1}^{n} \prod_{r=1/2}^{\infty} (c_{-r}^i)^{N_{ir}} |\Omega_j\rangle, \tag{71}$$

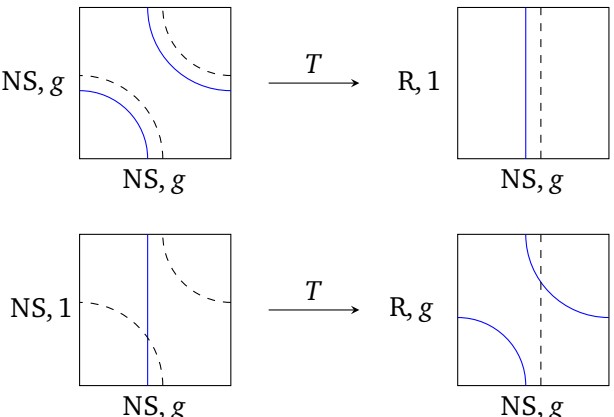

Figure 2: The schematic illustration for the modular transformation of the partition functions. The above and below show the modular $T$ transformation of $Z_{\mathcal{T}}[0,0;1,1]$ and $Z_{\mathcal{T}}[0,0;0,1]$, respectively. The blue (black dashed) line represents the insertion of the $\mathbb{Z}_2$ action $g$ (the fermion parity) on the torus.

where $j = 1, 2, \ldots, 2^{\frac{n}{2}}$. From $Z_{\mathcal{T}}[0,1;1,1] \propto (2\eta/\theta_3)^{\frac{n}{2}}$ up to a phase factor, we propose to define the $\mathbb{Z}_2$ symmetry $g$ acts as

$$g\, |\Omega_j\rangle = (-1)^{\frac{n}{8}} |\Omega_j\rangle \qquad (j = 1, 2, \ldots, 2^{\frac{n}{2}}). \tag{72}$$

On the other hand, the fermion parity acts as $+1$ on half of the twisted R ground states and as $-1$ on the other half. Otherwise, the partition function $Z_{\mathcal{T}}[1,1;1,1]$ is no longer invariant under the modular transformation $T$ and does not have a proper modular transformation rule. Thus, we can always set

$$(-1)^F |\Omega_j\rangle = \begin{cases} +|\Omega_j\rangle & (j \le 2^{\frac{n}{2}-1}), \\ -|\Omega_j\rangle & (j > 2^{\frac{n}{2}-1}). \end{cases} \tag{73}$$

Up to this point, we have defined the states in each sector and the action of the reflection $g$ and the fermion parity $(-1)^F$ on them. The partition functions can be summarized as follows:

$$\begin{aligned} Z_{\mathcal{T}}[s_0, s_1; 0, 0] &= Z_{\mathcal{T}}[s_0, s_1], \\ Z_{\mathcal{T}}[0,0;1,0] &= Z_{\mathcal{T}}[1,0,1,0] = (2\eta/\theta_2)^{\frac{n}{2}}, \\ Z_{\mathcal{T}}[0,0;0,1] &= Z_{\mathcal{T}}[0,1;0,1] = (2\eta/\theta_4)^{\frac{n}{2}}, \\ Z_{\mathcal{T}}[1,0;1,1] &= Z_{\mathcal{T}}[0,1;1,1] = (-1)^{\frac{n}{8}}(2\eta/\theta_3)^{\frac{n}{2}}, \\ Z_{\mathcal{T}}[0,1;1,0] &= Z_{\mathcal{T}}[1,1;1,0] = Z_{\mathcal{T}}[1,0;0,1] \\ &= Z_{\mathcal{T}}[0,0;1,1] = Z_{\mathcal{T}}[1,1;0,1] = Z_{\mathcal{T}}[1,1;1,1] = 0, \end{aligned} \tag{74}$$

where $Z_{\mathcal{T}}[s_0, s_1]$ is given by (27).

We can obtain the spin selection rule as summarized in table 5. The spin selection rule in the untwisted sector is ordinary: The NS sector consists of local operators satisfying the spin-statistics theorem. The spin of the R sector is at least a multiple of $1/16$. For a generic central charge $n$, the twisted sector contains operators with a fractional spin $s \notin \mathbb{Z}/2$. The non-anomalous condition $n \in 8\mathbb{Z}$ ensures that the twisted NS sector consists of $s \in \mathbb{Z}$ (boson) and $s \in \mathbb{Z} + \frac{1}{2}$ (fermion). After orbifolding, those operators join in the spectrum of the theory. The selection rule can be obtained through the modular $T$ transformation. For example, to

Table 5: Spin selection rule for the reflection symmetry $g$ in lattice CFT. We assume the non-anomalous condition ($n \in 8\mathbb{Z}$). Including the twisted sector, the theory only contains operators with an integral and half-integral spin.

|  |  | untwisted | | twisted | |
| --- | --- | :---: | :---: | :---: | :---: |
|  |  | NS | R | NS | R |
| $g$-even | boson | $\mathbb{Z}$ | $\mathbb{Z}$ | $\mathbb{Z}$ | $\mathbb{Z}$ |
|  | fermion | $\frac{1}{2}+\mathbb{Z}$ | $\mathbb{Z}$ | $\frac{1}{2}+\mathbb{Z}$ | $\mathbb{Z}$ |
| $g$-odd | boson | $\mathbb{Z}$ | $\mathbb{Z}$ | $\frac{1}{2}+\mathbb{Z}$ | $\frac{1}{2}+\mathbb{Z}$ |
|  | fermion | $\frac{1}{2}+\mathbb{Z}$ | $\mathbb{Z}$ | $\mathbb{Z}$ | $\frac{1}{2}+\mathbb{Z}$ |

give the spin selection rule for $g$-even and bosonic states in the twisted NS sector, we consider

$$\mathrm{Tr}_{\mathcal{H}_{\mathrm{NS},g}}\left[\frac{1+(-1)^F}{2}\frac{1+g}{2}q^{L_0-\frac{n}{24}}\right] = \frac{1}{4}(Z_{\mathcal{T}}[0,0;0,1]+Z_{\mathcal{T}}[0,0;1,1]+Z_{\mathcal{T}}[1,0;0,1]+Z_{\mathcal{T}}[1,0;1,1])$$

$$= \frac{1}{4}\left(\frac{2\eta}{\theta_4}\right)^{\frac{n}{2}}+\frac{1}{4}(-1)^{\frac{n}{8}}\left(\frac{2\eta}{\theta_3}\right)^{\frac{n}{2}}. \tag{75}$$

The modular $T$ transformation gives an eigenvalue $e^{\pi i n/24}$ when $n \in 16\mathbb{Z}$ and $-e^{\pi i n/24}$ when $n \in 8+16\mathbb{Z}$. Since the $T$ transformation acts on a state with spin $s$ by an eigenvalue $e^{2\pi i(s-n/24)}$, we obtain the spin selection rule $s \in \mathbb{Z}$.

As another example, we can compute the spin selection rule for the $g$-even/odd and bosonic states in the twisted R sector from

$$\mathrm{Tr}_{\mathcal{H}_{\mathrm{R},g}}\left[\frac{1+(-1)^F}{2}\frac{1\pm g}{2}q^{L_0-\frac{n}{24}}\right] = \frac{1}{4}\left(\frac{2\eta}{\theta_4}\right)^{\frac{n}{2}}\pm\frac{1}{4}(-1)^{\frac{n}{8}}\left(\frac{2\eta}{\theta_3}\right)^{\frac{n}{2}}, \tag{76}$$

which gives the phase $\pm(-1)^{\frac{n}{8}}e^{\frac{\pi i n}{24}}$ by the $T$ transformation. Therefore, the $g$-even sector consists of operators with $s \in \mathbb{Z}$, while the odd sector contains only operators with $s \in 1/2+\mathbb{Z}$. This means that a consistent orbifold theory cannot include the $g$-odd states in the twisted R sector of the original theory.

As in the case of the shift symmetry, two orbifold theories can be constructed for the reflection symmetry. In this case, it is clear from the construction that their spectra are identical. For example, the NS sectors of the orbifold theories are

$$\mathcal{H}_{\mathrm{NS}}^{\mathrm{orb}+}(\Lambda) = \mathcal{H}_{\mathrm{NS}}^{+g}\oplus\mathcal{H}_{\mathrm{NS},g}^{+g}, \qquad \mathcal{H}_{\mathrm{NS}}^{\mathrm{orb}-}(\Lambda) = \mathcal{H}_{\mathrm{NS}}^{+g}\oplus\mathcal{H}_{\mathrm{NS},g}^{-g}, \tag{77}$$

and the difference lies only in the choice of the ground states $|a\rangle$ in $\mathcal{H}_{\mathrm{NS},g}$ as shown in (67). Indeed, from table 1 and (74), the NS-NS and R-R partition functions of $(\mathcal{T}/G)_+$ are

$$Z_{(\mathcal{T}/G)_+}[0,0] = \frac{1}{2}(Z_{\mathcal{T}}[0,0;0,0]+Z_{\mathcal{T}}[0,0;1,0])+\frac{1}{2}(Z_{\mathcal{T}}[0,0;0,1]+Z_{\mathcal{T}}[0,0;1,1])$$

$$= \frac{\Theta_\Lambda(\tau)+(\theta_3\theta_4)^{\frac{n}{2}}+(\theta_2\theta_3)^{\frac{n}{2}}}{2\eta(\tau)^n}, \tag{78}$$

$$Z_{(\mathcal{T}/G)_+}[1,1] = \frac{1}{2}(Z_{\mathcal{T}}[1,1;0,0]+Z_{\mathcal{T}}[1,1;1,0])+\frac{1}{2}(Z_{\mathcal{T}}[1,1;0,1]+Z_{\mathcal{T}}[1,1;1,1])$$

$$= \frac{1}{2}Z_{\mathcal{T}}[1,1], \tag{79}$$

and the same functions are obtained for $(\mathcal{T}/G)_-$.

Let $V_{1/2}(\mathcal{T})$ be the space of the operators with the weight $h = 1/2$ in the NS sector of $\mathcal{T}$. From (78), the dimension of the space in the reflection orbifold theory becomes

$$|V_{1/2}((\mathcal{T}/G)_\pm)| = \frac{1}{2}|V_{1/2}(\mathcal{T})| + 8\,\delta_{n,8}\,. \tag{80}$$

It is known that a chiral CFT can be decomposed into Majorana-Weyl fermions $\psi$ and a sector without $h = 1/2$ operators [30], thus this means that the number of Majorana-Weyl fermions is halved except for $n = 8$. In the case of $n = 8$, the only chiral fermionic theory is $16\psi$ with $|V_{1/2}(16\psi)| = 16$ and then $\mathcal{T} \cong (\mathcal{T}/G)_\pm \cong 16\psi$, which is consistent with (80).

## 4  Chiral fermionic CFTs from $\mathbb{Z}_k$ codes

In this section, we introduce the construction of chiral fermionic CFTs from classical $\mathbb{Z}_k$ codes where $\mathbb{Z}_k$ is the ring of integers modulo $k$ with $k \geq 2$. Starting with $\mathbb{Z}_k$ codes, we give lattice CFTs based on odd self-dual lattices. This is a generalization of the construction from ternary codes ($k = 3$) [9] and $p$-ary codes ($k = p$ : a prime number) [8].[2]

A $\mathbb{Z}_k$ code $C$ of length $n$ is an additive Abelian subgroup of $\mathbb{Z}_k^n$. For a code $C \subset \mathbb{Z}_k^n$ with the standard inner product $c \cdot c' = \sum_{i=1}^n c_i c_i' \in \mathbb{Z}_k$ for $c, c' \in \mathbb{Z}_k^n$, the dual code is defined by

$$C^\perp = \left\{ c' \in \mathbb{Z}_k^n \mid c \cdot c' = 0,\ \text{for all } c \in C \right\}\,. \tag{81}$$

A code $C$ is called self-dual when $C = C^\perp$. In particular, when $k$ is even, a self-dual code is called Type II when $c^2 \in 2k\mathbb{Z}$ for all $c \in C$ and Type I when it is not Type II. In the case of binary codes, Type I codes are called singly-even.

For $c \in \mathbb{Z}_k^n$ and $a \in \mathbb{Z}_k$, $\mathrm{wt}_a(c) = |\{i \in \{1, \ldots, n\} \mid c_i = a\}|$ is called the weight and satisfies $\sum_{a \in \mathbb{Z}_k} \mathrm{wt}_a(c) = n$. In the binary case $k = 2$, we simply denote $\mathrm{wt}_1(c)$ by $\mathrm{wt}(c)$ and the error correction ability of a code $C \subset \mathbb{Z}_k^n$ is represented by the minimum weight $d(C) = \min\{\mathrm{wt}(c) \mid c \in C, c \neq 0\}$.

The complete weight enumerator of a subset $D \subset \mathbb{Z}_k^n$ is

$$W_D(\{x_a\}) = \sum_{c \in D} \prod_{a \in \mathbb{Z}_k} x_a^{\mathrm{wt}_a(c)}\,. \tag{82}$$

The simplest way to construct a lattice from a code $C \subset \mathbb{Z}_k^n$ is the method called Construction A:

$$\Lambda(C) = \left\{ \frac{c + k m}{\sqrt{k}} \in \mathbb{R}^n \ \middle|\ c \in C,\, m \in \mathbb{Z}^n \right\}\,. \tag{83}$$

It is known that $\Lambda(C)$ is odd self-dual if and only if $C$ is Type I when $k \in 2\mathbb{Z}$ and self-dual when $k \in 2\mathbb{Z} + 1$ [22]. Once an odd self-dual lattice is obtained, we can construct a fermionic CFT according to the method in section 3.1. In the following, we show the expression of the partition functions using the weight enumerator for even/odd $k$.

**$k$: Even**  A Type I code $C \subset \mathbb{Z}_k^n$ for even $k$ can be divided into two disjoint subsets $C = C_0 \sqcup C_2$ where

$$C_0 = \{c \in C \mid c^2 \in 2k\mathbb{Z}\}\,, \qquad C_2 = \{c \in C \mid c^2 \in 2k\mathbb{Z} + k\}\,. \tag{84}$$

The shadow of $C$ is defined by

$$S(C) = C_0^\perp \setminus C\,. \tag{85}$$

---

[2]Recently, the construction of CFTs from classical and quantum codes through lattices has been developed (see [31–50] for recent progress in this direction).

We choose a specific element $s \in S(C)$ and define

$$C_1 = C_0 + s, \qquad C_3 = C_2 + s, \tag{86}$$

which satisfy $S(C) = C_1 \sqcup C_3$. Note that $C_1$ and $C_3$ can be interchanged depending on the choice of $s \in S(C)$. For later convenience, we define

$$\widetilde{W}_C(\{x_a\}) = W_{C_1}(\{x_a\}) - W_{C_3}(\{x_a\}). \tag{87}$$

A characteristic vector of $\Lambda(C)$ can be written as

$$\chi = \frac{2}{\sqrt{k}}(s + km), \quad s \in S(C), \quad m \in \mathbb{Z}^n. \tag{88}$$

Using the theta function

$$\Theta_{a,l}(\tau) = \sum_{m=-\infty}^{\infty} q^{l\left(m + \frac{a}{2l}\right)^2}, \tag{89}$$

the partition functions of the CFT $\mathcal{T}$ constructed from $\Lambda(C)$ can be expressed as

$$Z_{\mathcal{T}}[0,0] = \frac{1}{\eta(\tau)^n}\left(W_{C_0}(\{\Theta_{a,\frac{k}{2}}\}) + W_{C_2}(\{\Theta_{a,\frac{k}{2}}\})\right) = \frac{1}{\eta(\tau)^n}W_C(\{\Theta_{a,\frac{k}{2}}\}), \tag{90a}$$

$$Z_{\mathcal{T}}[1,0] = \frac{1}{\eta(\tau)^n}\left(W_{C_0}(\{\Theta_{a,\frac{k}{2}}\}) - W_{C_2}(\{\Theta_{a,\frac{k}{2}}\})\right), \tag{90b}$$

$$Z_{\mathcal{T}}[0,1] = \frac{1}{\eta(\tau)^n}\left(W_{C_1}(\{\Theta_{a,\frac{k}{2}}\}) + W_{C_3}(\{\Theta_{a,\frac{k}{2}}\})\right) = \frac{1}{\eta(\tau)^n}W_{S(C)}(\{\Theta_{a,\frac{k}{2}}\}), \tag{90c}$$

$$Z_{\mathcal{T}}[1,1] = \frac{1}{\eta(\tau)^n}\left(W_{C_1}(\{\Theta_{a,\frac{k}{2}}\}) - W_{C_3}(\{\Theta_{a,\frac{k}{2}}\})\right) = \frac{1}{\eta(\tau)^n}\widetilde{W}_C(\{\Theta_{a,\frac{k}{2}}\}), \tag{90d}$$

where $C_1$ and $C_3$ are defined by $s \in S(C)$ when the fermion parity is determined by the characteristic vector $\chi = \frac{2}{\sqrt{k}}(s + km), m \in \mathbb{Z}^n$. In the binary case, $\Theta_{0,1}(\tau) = \theta_3(2\tau)$ and $\Theta_{1,1}(\tau) = \theta_2(2\tau)$.

**$k$: Odd**   For a self-dual code $C \subset \mathbb{Z}_k^n$ for odd $k$, the vector $\chi = \sqrt{k}(1, \ldots, 1) \in \Lambda(C)$ is always the characteristic vector of $\Lambda(C)$ and we choose this $\chi$ to determine the fermion parity in R sector.

By using a slightly modified version of [8] (3.36):

$$f_{a,k}^{\alpha,\beta}(\tau) = \sum_{m \in \mathbb{Z}} (-1)^{\alpha(km+a)} q^{\frac{k}{2}\left(m + \beta\frac{1}{2} + \frac{a}{k}\right)^2}, \tag{91}$$

the partition functions are

$$Z_{\mathcal{T}}[\alpha, \beta] = \frac{1}{\eta(\tau)^n} W_C\left(f_{0,k}^{\alpha,\beta}(\tau), f_{1,k}^{\alpha,\beta}(\tau), \ldots, f_{k-1,k}^{\alpha,\beta}(\tau)\right), \tag{92}$$

where $\alpha, \beta \in \{0, 1\}$. In concrete calculations, relations such as $f_{0,k}^{1,1}(\tau) = 0$ and $f_{a,k}^{\alpha,\beta}(\tau) = (-1)^{\alpha\beta} f_{-a,k}^{\alpha,\beta}(\tau)$ are useful.

## 5   Triality structure from binary codes

In this section, we discuss the equivalence between the reflection orbifold and the shift orbifold for the binary codes. We first compute the partition functions of the lattice CFT from binary codes and then show the equivalence between the reflection and shift orbifolds.

## 5.1 Shift orbifold for binary codes

As discussed in section 3.2, we can define the shift orbifold of the lattice CFT based on an odd self-dual lattice $\Lambda$ and the shift vector $\delta$ that is not a characteristic vector and $\delta/2 \notin \Lambda$. The non-anomalous condition of the shift symmetry $H$ is $\delta^2 \in 4\mathbb{Z}$.

In the rest of this subsection, we focus on the shift orbifold of the lattice CFT constructed from a binary code $C \subset \mathbb{Z}_2^n$ and compute their explicit partition functions. For a singly-even self-dual code $C \subset \mathbb{Z}_2^n$, we can construct an odd self-dual lattice $\Lambda(C)$ by the Construction A (83). Correspondingly, we can construct a chiral fermionic CFT $\mathcal{T}$ based on the lattice $\Lambda(C)$. By the shift symmetry $H$, the original theory $\mathcal{T}$ is graded into two parts. In terms of the momentum lattice $\Lambda(C)$, this grading is given by $\Lambda(C) = \Lambda_{\delta\text{-even}} \sqcup \Lambda_{\delta\text{-odd}}$ where

$$
\begin{aligned}
\Lambda_{\delta\text{-even}} &= \{\lambda \in \Lambda \mid \lambda \cdot \delta \in 2\mathbb{Z}\}\,, \\
\Lambda_{\delta\text{-odd}} &= \{\lambda \in \Lambda \mid \lambda \cdot \delta \in 2\mathbb{Z}+1\}\,.
\end{aligned}
\tag{93}
$$

Following the general prescription (41), the momentum lattices of the orbifold theories $(\mathcal{T}/H)_{\pm}$ are given by

$$
\begin{aligned}
\Lambda_{\delta}^{\text{orb}+} &= \Lambda_{\delta\text{-even}} \sqcup \left(\Lambda_{\delta\text{-even}} + \tfrac{1}{2}\delta\right)\,, \\
\Lambda_{\delta}^{\text{orb}-} &= \Lambda_{\delta\text{-even}} \sqcup \left(\Lambda_{\delta\text{-odd}} + \tfrac{1}{2}\delta\right)\,.
\end{aligned}
\tag{94}
$$

In the binary case, we set the shift symmetry $H$ generated by the shift vector

$$
\delta = \frac{1}{\sqrt{2}}(1,1,\ldots,1)\,.
\tag{95}
$$

The shift vector is not a characteristic vector and $\delta/2 \notin \Lambda(C)$. The non-anomalous condition leads to $n \in 8\mathbb{Z}$.

For the shift vector $\delta$, we can write down

$$
\begin{aligned}
\Lambda_{\delta\text{-even}} &= \left(\frac{C_0 + 2\mathbb{Z}_+^n}{\sqrt{2}}\right) \sqcup \left(\frac{C_2 + 2\mathbb{Z}_-^n}{\sqrt{2}}\right)\,, \\
\Lambda_{\delta\text{-odd}} &= \left(\frac{C_0 + 2\mathbb{Z}_-^n}{\sqrt{2}}\right) \sqcup \left(\frac{C_2 + 2\mathbb{Z}_+^n}{\sqrt{2}}\right)\,,
\end{aligned}
\tag{96}
$$

where $C_0$ and $C_2$ are the set of doubly-even and singly-even codewords, respectively ($C = C_0 \sqcup C_2$). Here, we use the notation

$$
\begin{aligned}
\mathbb{Z}_+^n &= \left\{m \in \mathbb{Z}^n \mid m^2 \in 2\mathbb{Z}\right\}\,, \\
\mathbb{Z}_-^n &= \left\{m \in \mathbb{Z}^n \mid m^2 \in 2\mathbb{Z}+1\right\}\,.
\end{aligned}
\tag{97}
$$

On the other hand, the two parts of the shifted lattice $\Lambda + \delta/2$ are given by

$$
\begin{aligned}
(\Lambda + \tfrac{\delta}{2})_{\delta\text{-even}} &= \left(\frac{C_0 + 2\mathbb{Z}_+^n}{\sqrt{2}} + \frac{\delta}{2}\right) \sqcup \left(\frac{C_2 + 2\mathbb{Z}_-^n}{\sqrt{2}} + \frac{\delta}{2}\right)\,, \\
(\Lambda + \tfrac{\delta}{2})_{\delta\text{-odd}} &= \left(\frac{C_2 + 2\mathbb{Z}_+^n}{\sqrt{2}} + \frac{\delta}{2}\right) \sqcup \left(\frac{C_0 + 2\mathbb{Z}_-^n}{\sqrt{2}} + \frac{\delta}{2}\right)\,.
\end{aligned}
\tag{98}
$$

The twisted ground states with conformal weight $h = n/16$ are in the first terms in the above two equations. The total number of the twisted ground states is $2^{n/2}$. We can see that half of them are even, and the other half are odd under the $\mathbb{Z}_2$ symmetry given by $\delta$.

In what follows, we compute the NS-NS partition function of the shift orbifold theory $(\mathcal{T}/H)_\pm$. Since the theory after orbifold is still a lattice CFT, the NS-NS partition function is given by

$$Z_{(\mathcal{T}/H)_\pm}[0,0] = \frac{1}{\eta(\tau)^n} \Theta_{\Lambda_\delta^{\text{orb}\pm}}(q),$$ (99)

where $\Theta_{\Lambda_\delta^{\text{orb}\pm}}(q)$ is the theta function of the shifted lattice $\Lambda_\delta^{\text{orb}\pm}$:

$$\Theta_{\Lambda_\delta^{\text{orb}\pm}}(q) = \sum_{\lambda \in \Lambda_\delta^{\text{orb}\pm}} q^{\lambda^2/2}.$$ (100)

From (94), we are required to have the theta function of $\Lambda_{\delta\text{-even}}$, $\Lambda_{\delta\text{-even}} + \delta/2$ and $\Lambda_{\delta\text{-odd}} + \delta/2$. To this purpose, we show the following propositions:

**Proposition 5.1**

For a singly-even self-dual code $C \subset \mathbb{Z}_2^n$, the theta functions of $\Lambda_{\delta\text{-even}}$ and $\Lambda_{\delta\text{-odd}}$ are

$$\begin{aligned}
\Theta_{\Lambda_{\delta\text{-even}}}(q) &= \frac{\Theta_{\Lambda(C)}(q) + (\theta_3(q)\theta_4(q))^{n/2}}{2}, \\
\Theta_{\Lambda_{\delta\text{-odd}}}(q) &= \frac{\Theta_{\Lambda(C)}(q) - (\theta_3(q)\theta_4(q))^{n/2}}{2},
\end{aligned}$$ (101)

where $\theta_i(q)$ are the Jacobi theta functions.

*Proof.* The essential ingredient of the proof is the equality

$$\sum_{c \in K} \sum_{m \in \mathbb{Z}_+^n} q^{(m+\frac{c}{2})^2} = \sum_{c \in K} \sum_{m \in \mathbb{Z}_-^n} q^{(m+\frac{c}{2})^2}.$$ (102)

This holds for any subset $K \subset \mathbb{Z}_2^n$ that does not contain the all-zeros vector: $0^n \notin K$ since if $c_i = 1$, we can transform into $m_i \to -m_i - 1$, which only exchanges $\mathbb{Z}_+^n$ and $\mathbb{Z}_-^n$ of the region where $m$ runs. Now we apply this equality for proof.

Let us consider

$$\Theta_{\Lambda(C)}(q) = \Theta_{\Lambda_{\delta\text{-even}}}(q) + \Theta_{\Lambda_{\delta\text{-odd}}}(q),$$ (103)

where

$$\Theta_{\Lambda_{\delta\text{-even}}}(q) = \sum_{c \in C_0} \sum_{m \in \mathbb{Z}_+^n} q^{(m+\frac{c}{2})^2} + \sum_{c \in C_2} \sum_{m \in \mathbb{Z}_-^n} q^{(m+\frac{c}{2})^2},$$ (104)

$$\Theta_{\Lambda_{\delta\text{-odd}}}(q) = \sum_{c \in C_0} \sum_{m \in \mathbb{Z}_-^n} q^{(m+\frac{c}{2})^2} + \sum_{c \in C_2} \sum_{m \in \mathbb{Z}_+^n} q^{(m+\frac{c}{2})^2}.$$ (105)

Since the equality (102), the last terms in (104) and (105) are the same. On the other hand, the first terms are not due to the all-zeros vector $c = 0^n$, which contributes to $\Theta_{\Lambda(C)}(q)$ as

$$\sum_{m \in \mathbb{Z}^n} q^{m^2} = \theta_3(q^2)^n.$$ (106)

Therefore, after using (102), we can write

$$\begin{aligned}
\Theta_{\Lambda(C)}(q) &= \theta_3(q^2)^n + 2 \sum_{c \in C_0 - \{0^n\}} \sum_{m \in \mathbb{Z}_+^n} q^{(m+\frac{c}{2})^2} + 2 \sum_{c \in C_2} \sum_{m \in \mathbb{Z}_-^n} q^{(m+\frac{c}{2})^2} \\
&= -\theta_4(q^2)^n + 2 \Theta_{\Lambda_{\delta\text{-even}}}(q),
\end{aligned}$$ (107)

where we have used $2 \sum_{m \in \mathbb{Z}_+^n} q^{m^2} = \theta_3(q^2)^n + \theta_4(q^2)^n$. Lastly, we can use the identity $\theta_4(q^2) = \sqrt{\theta_3(q)\theta_4(q)}$. □

**Proposition 5.2**

For a singly-even self-dual code $C \subset \mathbb{Z}_2^n$, the theta functions of $\Lambda_{\delta\text{-even}} + \frac{\delta}{2}$ and $\Lambda_{\delta\text{-odd}} + \frac{\delta}{2}$ are

$$\Theta_{\Lambda_{\delta\text{-even}}+\delta/2}(q) = \Theta_{\Lambda_{\delta\text{-odd}}+\delta/2}(q) = \frac{(\theta_2(q)\theta_3(q))^{n/2}}{2}. \tag{108}$$

*Proof.* Let us consider

$$\Theta_{\Lambda_{\delta\text{-even}}+\delta/2}(q) = \sum_{c \in C_0} \sum_{m \in \mathbb{Z}_+^n} q^{(m+\frac{c}{2}+\frac{1}{4})^2} + \sum_{c \in C_2} \sum_{m \in \mathbb{Z}_-^n} q^{(m+\frac{c}{2}+\frac{1}{4})^2}. \tag{109}$$

To evaluate the above equation, we utilize the equality

$$\sum_{m \in \mathbb{Z}_\pm^n} q^{(m+\frac{c}{2}+\frac{1}{4})^2} = \sum_{m \in \mathbb{Z}_\pm^n} q^{(m+\frac{1}{4})^2}, \tag{110}$$

where $\mathrm{wt}(c) \in 2\mathbb{Z}$. To see this, suppose that the first $l$ components of $c$ are 1 and the others are 0 where $l$ is even due to $\mathrm{wt}(c) \in 2\mathbb{Z}$. Then, we change the variable $m$ by $m_i \to -m_i - 1$ $(i = 1, 2, \ldots, l)$ and $m_i \to m_i$ $(i = l+1, \ldots, n)$. Since $l \in 2\mathbb{Z}$, this change of variables does not affect the region where $m$ runs and

$$\left(m_1 + \tfrac{1}{2} + \tfrac{1}{4}, \ldots, m_l + \tfrac{1}{2} + \tfrac{1}{4}, m_{l+1} + \tfrac{1}{4}, \ldots, m_n + \tfrac{1}{4}\right)$$
$$\to \left(-m_1 - \tfrac{1}{4}, \ldots, -m_l - \tfrac{1}{4}, m_{l+1} + \tfrac{1}{4}, \ldots, m_n + \tfrac{1}{4}\right). \tag{111}$$

Thus, we obtain (110). If one takes the sum over the subset $K \subset \mathbb{Z}_2^n$ satisfying $\mathrm{wt}(c) \in 2\mathbb{Z}$ for any $c \in K$, we obtain the equality

$$\sum_{c \in K} \sum_{m \in \mathbb{Z}_\pm^n} q^{(m+\frac{c}{2}+\frac{1}{4})^2} = |K| \sum_{m \in \mathbb{Z}_\pm^n} q^{(m+\frac{1}{4})^2}, \tag{112}$$

where $|K|$ is the number of elements in $K \subset \mathbb{Z}_2^n$.

By using (112), we obtain

$$\begin{aligned}
\Theta_{\Lambda_{\delta\text{-even}}+\delta/2}(q) &= |C_0| \sum_{m \in \mathbb{Z}_+^n} q^{(m+\frac{1}{4})^2} + |C_2| \sum_{m \in \mathbb{Z}_-^n} q^{(m+\frac{1}{4})^2} \\
&= \frac{|C|}{2} \sum_{m \in \mathbb{Z}^n} q^{(m+\frac{1}{4})^2} = \frac{(\theta_2(q)\theta_3(q))^{n/2}}{2},
\end{aligned} \tag{113}$$

where we used $|C_0| = |C_2| = |C|/2$ and the identity $2\rho_0(q^2)^2 = \theta_2(q)\theta_3(q)$ [51] where $\rho_0(q)$ is the theta function of lattice $\mathbb{Z} + \frac{1}{4}$: $\rho_0(q) = \sum_{m \in \mathbb{Z}} q^{\frac{1}{2}(m+\frac{1}{4})^2}$. For the other $\Lambda_{\delta\text{-odd}} + \delta/2$, the same argument can be made. $\square$

Combining the above two propositions, we obtain the theta functions of the momentum lattices of the orbifold theory $(\mathcal{T}/H)_\pm$:

**Proposition 5.3**

For a singly-even self-dual code $C \subset \mathbb{Z}_2^n$, the theta functions of the shifted lattices are

$$\Theta_{\Lambda_\delta^{\text{orb}\pm}}(q) = \frac{\Theta_{\Lambda(C)}(q) + (\theta_2\theta_3)^{n/2} + (\theta_3\theta_4)^{n/2}}{2}. \tag{114}$$

Under the modular transformations, the theta functions of the shifted lattice transform as ($n \in 8\mathbb{Z}$ due to the non-anomalous condition)

$$S: \quad \Theta_{\Lambda_\delta^{\mathrm{orb}\pm}}(q) \to (-\mathrm{i}\tau)^{\frac{n}{2}} \, \Theta_{\Lambda_\delta^{\mathrm{orb}\pm}}(q)\,, \tag{115}$$

$$T^2: \quad \Theta_{\Lambda_\delta^{\mathrm{orb}\pm}}(q) \to \Theta_{\Lambda_\delta^{\mathrm{orb}\pm}}(q)\,. \tag{116}$$

This implies that the theta functions of the shifted lattices are a modular form of weight $n/2$ for the subgroup $\Gamma \subset \mathrm{SL}(2,\mathbb{Z})$ generated by $S$ and $T^2$. Therefore, from (99), the NS-NS partition function of the orbifold theory

$$Z_{(\mathcal{T}/H)_\pm}[0,0] = \frac{\Theta_{\Lambda(C)}(q) + (\theta_2\theta_3)^{n/2} + (\theta_3\theta_4)^{n/2}}{2\eta(\tau)^n}\,, \tag{117}$$

properly transforms under modular transformation.

The above proposition states that the two orbifold theories $(\mathcal{T}/H)_\pm$ have the same NS-NS partition function. The following proposition ensures that the two orbifold theories are equivalent since their momentum lattices are equivalent up to a reflection transformation. Thus, in the rest of this paper, we denote the shift orbifold theory from binary codes by $\mathcal{T}/H$ and its momentum lattice by $\Lambda_\delta^{\mathrm{orb}}$, omitting types of orbifold ($\pm$).

**Proposition 5.4**
In the binary construction, the two shifted lattices are equivalent

$$\Lambda_\delta^{\mathrm{orb}+} \cong \Lambda_\delta^{\mathrm{orb}-}\,. \tag{118}$$

*Proof.* Let us construct an isomorphism between $\Lambda_{\delta\text{-even}} + \frac{\delta}{2}$ and $\Lambda_{\delta\text{-odd}} + \frac{\delta}{2}$ by reflection. Any element in $\Lambda_{\delta\text{-even}} + \frac{\delta}{2}$ can be written as $\lambda = \sqrt{2}(m + \frac{c}{2} + \frac{1}{4})$ where $(m \in \mathbb{Z}_+^n$ and $c \in C_0) \sqcup (m \in \mathbb{Z}_-^n$ and $c \in C_2)$. Given $t \in C_2$, $C_2 = C_0 + t$. This can be used to write an element of $\Lambda_{\delta\text{-even}} + \frac{\delta}{2}$ as

$$\lambda = \sqrt{2}(m + \tfrac{c+t}{2} + \tfrac{1}{4})\,, \tag{119}$$

where $(m \in \mathbb{Z}_+^n$ and $c \in C_2) \sqcup (m \in \mathbb{Z}_-^n$ and $c \in C_0)$. For simplicity, suppose $t = 1^l 0^{n-l} \in C_2$ where $l \in 2\mathbb{Z}$ due to the self-orthogonality of $C$. We can rewrite (119) by $m_i \to -m_i - c_i - 1$ $(i = 1, 2, \ldots, l)$ and $m_i \to m_i$ $(i = l+1, \ldots, n)$. Then, we obtain

$$\lambda = (-m_1 - \tfrac{c_1}{2} - \tfrac{1}{4}, \ldots, -m_l - \tfrac{c_l}{2} - \tfrac{1}{4}, m_{l+1} + \tfrac{c_{l+1}}{2} + \tfrac{1}{4}, \ldots, m_n + \tfrac{c_n}{2} + \tfrac{1}{4})\,. \tag{120}$$

Note that this change of variables does not affect the region where $m$ runs because $\sum_{i=1}^l (-m_i - c_i - 1) + \sum_{i=l+1}^n m_i = \sum_{i=1}^n m_i + c \cdot t = \sum_{i=1}^l m_i \bmod 2$. Here, we used $l \in 2\mathbb{Z}$ and $c \cdot t \in 2\mathbb{Z}$ by self-orthogonality. The above procedure just changes the representation of an element in $\Lambda_{\delta\text{-even}} + \frac{\delta}{2}$.

Now we give a map between $\Lambda_{\delta\text{-even}} + \frac{\delta}{2}$ and $\Lambda_{\delta\text{-odd}} + \frac{\delta}{2}$ by the reflection of the first $l$ components. The reflection maps (120) to an element $\lambda = \sqrt{2}(m + \frac{c}{2} + \frac{1}{4})$ where $(m \in \mathbb{Z}_+^n$ and $c \in C_2) \sqcup (m \in \mathbb{Z}_-^n$ and $c \in C_0)$, which is the definition of $\Lambda_{\delta\text{-odd}} + \frac{\delta}{2}$. We can easily check that the reflection preserves $\Lambda_{\delta\text{-even}}$, so we conclude the equivalence between the two shifted lattices. $\qquad\square$

## 5.2 Triality structure in chiral fermionic CFTs

Now we move on to the equivalence between the reflection orbifold and the shift orbifold for the binary codes. For bosonic CFTs from binary codes, the equivalence between the reflection and shift orbifolds was shown in [6,7]. We extend this result to chiral fermionic CFTs constructed from binary codes.

Our goal is the following proposition:

**Proposition 5.5**
Let $\mathcal{T}$ be a chiral fermionic CFT constructed from a singly-even self-dual code $C$. The reflection orbifold $(\mathcal{T}/G)_{\pm}$ and the shift orbifold $\mathcal{T}/H$ are isomorphic: $(\mathcal{T}/G)_{\pm} \cong \mathcal{T}/H$.

From the above proposition, we can deduce the equivalence $(\mathcal{T}/G)_{+} \cong (\mathcal{T}/G)_{-}$ between two types of orbifold in the binary construction. We can simply denote them by $\mathcal{T}/G$.

This equivalence is expected because the NS-NS partition function of the shift orbifold (117) is the same as that of the reflection orbifold (78). By the equivalence between the two orbifolds, we obtain the pyramid structure of the chiral fermionic CFTs from binary codes as shown in Fig. 1.

To hold the equivalence, an SU(2) symmetry from binary codes is crucial. Following [13, 52], the equivalence between the reflection and shift orbifolds can be understood intuitively. The original theory $\mathcal{T}$ has an SU(2) symmetry for each direction, generated by

$$
\begin{aligned}
J_1^i(z) &= \frac{i}{2}\left(V_{\sqrt{2}e_i}(z) + V_{-\sqrt{2}e_i}(z)\right), \\
J_2^i(z) &= \frac{1}{2}\left(V_{\sqrt{2}e_i}(z) - V_{-\sqrt{2}e_i}(z)\right), \\
J_3^i(z) &= \frac{i}{\sqrt{2}}\,\partial X^i(z),
\end{aligned}
\tag{121}
$$

where our notation is $X^i(z)X^j(w) = -\delta^{ij}\log(z-w)$. After the mode expansion $J_a^i(z) = \sum_{n\in\mathbb{Z}} J_n^{ai}/z^{n+1}$, these modes realize the commutation relations of $su(2)_1^n$ algebra

$$
[J_m^{ai}, J_n^{bj}] = i\,\epsilon_{abc}\,\delta^{ij}\,J_{m+n}^{ic} + \frac{m}{2}\,\delta^{ij}\,\delta^{ab}\,\delta_{m+n,0},
\tag{122}
$$

where $\epsilon_{abc}$ denotes the completely antisymmetric tensor ($\epsilon_{123} = 1$). Also, to compute the OPE (23), we assumed the gauge $\epsilon(\sqrt{2}e_i, -\sqrt{2}e_i) = \epsilon(-\sqrt{2}e_i, \sqrt{2}e_i) = -1$, which is compatible with (20). The reflection $g \in G$, the shift $h \in H$, and their product $gh$ act on the SU(2) symmetry as

$$
\begin{aligned}
g:&\quad J_1^i(z) \to +J_1^i(z), &\quad J_2^i(z) \to -J_2^i(z), &\quad J_3^i(z) \to -J_3^i(z), \\
h:&\quad J_1^i(z) \to -J_1^i(z), &\quad J_2^i(z) \to -J_2^i(z), &\quad J_3^i(z) \to +J_3^i(z), \\
gh:&\quad J_1^i(z) \to -J_1^i(z), &\quad J_2^i(z) \to +J_2^i(z), &\quad J_3^i(z) \to -J_3^i(z).
\end{aligned}
\tag{123}
$$

These actions are equivalent under the permutation $S_3$ group that interchanges the SU(2) symmetry generators. This triality symmetry suggests the equivalence between the $\mathbb{Z}_2$ orbifold theories by $G$ and $H$. For a compact boson theory with $c = \bar{c} = 1$, starting from the radius with SU(2) symmetry, the triality gives the intersection point of the torus branch and orbifold branch by the shift and reflection orbifold [53].

To relate the $\mathbb{Z}_2$ actions $g$ and $h$, we need to consider an operator swapping the SU(2) generators $J_1^i(z)$ and $J_3^i(z)$. The triality operator is given by

$$
\sigma^i = \exp\left\{\frac{i\pi}{\sqrt{2}}(J_0^{1i} + J_0^{3i})\right\}.
\tag{124}
$$

By taking the product $\sigma = \prod_{i=1}^{n} \sigma^i$, we can construct the operator that swaps the SU(2) generators $J_1^i(z)$ and $J_3^i(z)$ for all directions:

$$
\sigma J_m^{1i} \sigma^{-1} = J_m^{3i}, \qquad \sigma J_m^{2i} \sigma^{-1} = -J_m^{2i}, \qquad \sigma J_m^{3i} \sigma^{-1} = J_m^{1i}.
\tag{125}
$$

Let $V_{\delta\text{-even}}^{+} \subset \mathcal{H}_{\text{NS}}$ be the space of states fixed by the action of $g$ and $h$ in the NS sector of the theory $\mathcal{T}$: for a state $|\psi\rangle \in V_{\delta\text{-even}}^{+}$, $|\psi\rangle = g|\psi\rangle = h|\psi\rangle$. For a codeword $c \in C$, we define

operators that map $V^+_{\delta\text{-even}}$ to itself:

$$V^\pm_{ij}(z) = \frac{1}{\sqrt{2}} \left( : e^{i\sqrt{2}(e_i \pm e_j)\cdot X(z)} : + : e^{-i\sqrt{2}(e_i \pm e_j)\cdot X(z)} : \right), \tag{126}$$

$$V_{\zeta_c}(z) = \frac{1}{\sqrt{2}} \left( : e^{i\frac{c}{\sqrt{2}}\cdot X(z)} : + : e^{-i\frac{c}{\sqrt{2}}\cdot X(z)} : \right), \tag{127}$$

where $e_i$ is the unit vector in direction $i = 1, 2, \ldots, n$. From (96), any state in $V^+_{\delta\text{-even}}$ is generated by acting $: \partial X^i(z) \partial X^j(z) :$, $V^\pm_{ij}(z)$ and $V_{\zeta_c}(z)$ ($c \in C_0$: doubly-even codewords) on $|0\rangle$ and $|\lambda\rangle_+ = |\lambda\rangle + |-\lambda\rangle$ where $\lambda \in (C_2 + 2\mathbb{Z}^n)/\sqrt{2}$.

Now we show the following proposition:

**Proposition 5.6** (Generalization of Proposition 7.2 in [7])
The triality operator $\sigma$ maps $V^+_{\delta\text{-even}}$ to $V^+_{\delta\text{-even}}$.

*Proof.* The space $V^+_{\delta\text{-even}}$ is divided into two parts $V^+_{\delta\text{-even}} = V^{+,\text{bos}}_{\delta\text{-even}} \oplus V^{+,\text{ferm}}_{\delta\text{-even}}$. The bosonic sector $V^{+,\text{bos}}_{\delta\text{-even}}$ is generated by acting $: \partial X^i(z) \partial X^j(z) :$, $V^\pm_{ij}(z)$ and $V_{\zeta_c}(z)$ ($c \in C_0$) on the vacuum $|0\rangle$. The fermionic sector $V^{+,\text{ferm}}_{\delta\text{-even}}$ is generated by acting the same operators on a state $|\lambda\rangle_+$ where $\lambda \in (C_2 + 2\mathbb{Z}^n_-)/\sqrt{2}$. For the bosonic sector, the proposition has already been shown in [7]. The only difference with our case is the presence of the fermionic sector $V^{+,\text{ferm}}_{\delta\text{-even}}$ in $V^+_{\delta\text{-even}}$.

The strategy of proof is to show that the operators $\sigma : \partial X^i(z) \partial X^j(z) : \sigma^{-1}$, $\sigma V^\pm_{ij}(z) \sigma^{-1}$ and $\sigma V_{\zeta_c}(z) \sigma^{-1}$ map $V^+_{\delta\text{-even}}$ to itself and $\sigma |\lambda\rangle_+ \in V^+_{\delta\text{-even}}$. Once these are true, using $\sigma |0\rangle = |0\rangle$, we can conclude that $\sigma$ maps $V^+_{\delta\text{-even}}$ to itself. From Proposition 7.2 in [7], we know that $\sigma : \partial X^i(z) \partial X^j(z) : \sigma^{-1}$, $\sigma V^\pm_{ij}(z) \sigma^{-1}$ and $\sigma V_{\zeta_c}(z) \sigma^{-1}$ are operators that map $V^+_{\delta\text{-even}}$ to itself. Thus, we only need to show that $\sigma |\lambda\rangle_+ \in V^+_{\delta\text{-even}}$ for our proof.

Below, we show that $\sigma |\lambda\rangle_+ \in V^+_{\delta\text{-even}}$. This statement is equivalent to that the operator $\sigma (V_\lambda(z) + V_{-\lambda}(z)) \sigma^{-1}$ maps $V^+_{\delta\text{-even}}$ to itself since $\sigma |0\rangle = |0\rangle$. Without loss of generality, we assume an element $\lambda \in (C_2 + 2\mathbb{Z}^n_-)/\sqrt{2}$ by

$$\lambda = \frac{1}{\sqrt{2}} (1, \ldots, 1, -1, \ldots, -1, 0, \ldots, 0) \in \frac{C_2 + 2\mathbb{Z}^n_-}{\sqrt{2}}, \tag{128}$$

where the first $2\ell + 1$ components are 1, the next $2\ell + 1$ components are $-1$, and the others are 0. Setting $\lambda' = \lambda - \sqrt{2} e_i$ for $i = 1, 2, \ldots 2\ell + 1$ and $\lambda' = \lambda + \sqrt{2} e_i$ for $i = 2\ell + 2, \ldots, 4\ell + 2$, we have

$$[J^{1i}_0, V_\lambda(z)] = \frac{i}{2} \epsilon(\mp\sqrt{2} e_i, \lambda) V_{\lambda'}(z), \qquad [J^{1i}_0, V_{\lambda'}(z)] = \frac{i}{2} \epsilon(\pm\sqrt{2} e_i, \lambda') V_\lambda(z),$$
$$[J^{3i}_0, V_\lambda(z)] = \pm\frac{1}{2} V_\lambda(z), \qquad [J^{3i}_0, V_{\lambda'}(z)] = \mp\frac{1}{2} V_{\lambda'}(z), \tag{129}$$

where the upper and lower signs are for $i = 1, 2, \ldots 2\ell + 1$ and $i = 2\ell + 2, \ldots, 4\ell + 2$, respectively. Since we are using the gauge $\epsilon(\sqrt{2} e_i, -\sqrt{2} e_i) = \epsilon(-\sqrt{2} e_i, \sqrt{2} e_i) = -1$, the cocycle condition (22) tells us $\epsilon(-\sqrt{2} e_i, \lambda) = -\epsilon(\sqrt{2} e_i, \lambda - \sqrt{2} e_i)$ and $\epsilon(\sqrt{2} e_i, \lambda) = -\epsilon(-\sqrt{2} e_i, \lambda + \sqrt{2} e_i)$. Thus, we have

$$\sigma^i \begin{pmatrix} V_\lambda(z) \\ V_{\lambda'(z)} \end{pmatrix} (\sigma^i)^{-1} = \exp\left\{ \frac{i\pi}{2\sqrt{2}} \begin{pmatrix} \pm 1 & i\epsilon(\mp\sqrt{2} e_i, \lambda) \\ -i\epsilon(\mp\sqrt{2} e_i, \lambda) & \mp 1 \end{pmatrix} \right\} \begin{pmatrix} V_\lambda(z) \\ V_{\lambda'(z)} \end{pmatrix}$$
$$= \frac{i}{\sqrt{2}} \begin{pmatrix} \pm 1 & i\epsilon(\mp\sqrt{2} e_i, \lambda) \\ -i\epsilon(\mp\sqrt{2} e_i, \lambda) & \mp 1 \end{pmatrix} \begin{pmatrix} V_\lambda(z) \\ V_{\lambda'(z)} \end{pmatrix}. \tag{130}$$

By taking the products with respect to $i$, we obtain

$$\sigma V_\lambda(z)\sigma^{-1} = -2^{-|\lambda|^2}\sum_{\lambda'\in\Delta(\lambda)} i^{n(\lambda,\lambda')}(-1)^{n_R(\lambda,\lambda')}\eta(\lambda,\lambda')V_{\lambda'}(z), \tag{131}$$

where $\Delta(\lambda)$ is the set of vectors obtained by the sign-flip of the components such that $\lambda_j = \pm 1$. We denote by $n(\lambda,\lambda')$ the number of components such that $\lambda_j \neq \lambda'_j$. We define by $\mathcal{I}_L$ and $\mathcal{I}_R$ the set of sign-flipped components from $\lambda$ to $\lambda'$ in $j = 1, 2, \ldots, 2\ell+1$ and $j = 2\ell+2, \ldots, 4\ell+2$, respectively. Here, $n_R(\lambda,\lambda') = |\mathcal{I}_R|$ and

$$\eta(\lambda,\lambda') = \prod_{\mathcal{I}_L}\epsilon(-\sqrt{2}e_i,\lambda)\prod_{\mathcal{I}_R}\epsilon(\sqrt{2}e_i,\lambda). \tag{132}$$

Similarly, we have

$$\sigma V_{-\lambda}(z)\sigma^{-1} = -2^{-|\lambda|^2}\sum_{\lambda'\in\Delta(\lambda)} (-i)^{n(\lambda,\lambda')}(-1)^{n_R(\lambda,\lambda')}\eta(-\lambda,-\lambda')V_{-\lambda'}(z), \tag{133}$$

where

$$\eta(-\lambda,-\lambda') = \prod_{\mathcal{I}_L}\epsilon(\sqrt{2}e_i,-\lambda)\prod_{\mathcal{I}_R}\epsilon(-\sqrt{2}e_i,-\lambda). \tag{134}$$

Now we would like to see $\sigma(V_\lambda(z) + V_{-\lambda}(z))\sigma^{-1}$ by taking the sum of the above two equations. Note that $n(\lambda,\lambda') + n(\lambda,-\lambda') = 4\ell + 2$ and $n_R(\lambda,\lambda') + n_R(\lambda,-\lambda') = 2\ell + 1$. Also, we have $\eta(\lambda,\lambda') = (-1)^{n(\lambda,\lambda')}\eta(\lambda,-\lambda')$ because of the complex conjugation $\epsilon(\lambda,\mu)^* = \epsilon(-\mu,-\lambda) \in \{\pm 1\}$ and (20). Finally, we obtain

$$\sigma(V_\lambda(z) + V_{-\lambda}(z))\sigma^{-1} = -2^{-|\lambda|^2}\sum_{\lambda'\in\Delta_+(\lambda)} i^{n(\lambda,\lambda')}(-1)^{n_R(\lambda,\lambda')}\eta(\lambda,\lambda')(V_{\lambda'}(z) + V_{-\lambda'}(z)). \tag{135}$$

Here, $\Delta_+(\lambda)$ is the subset of $\Delta(\lambda)$ satisfying $n(\lambda,\lambda') \in 2\mathbb{Z}$. The operators appearing in the right-hand side are operators that map $V_{\delta\text{-even}}^+$ to itself, which concludes the proof. $\quad\square$

In the rest of this subsection, we give a proof of Proposition 5.5. For this purpose, we consider a CFT with weight-one operators $P^i(z)$ for $i = 1, 2, \ldots, n$ whose mode expansion is

$$P^i(z) = \sum_{m\in\mathbb{Z}}\frac{\alpha_m^i}{z^{m+1}}, \tag{136}$$

where the oscillators $\alpha_m^i$ satisfy the commutation relations $[\alpha_m^i, \alpha_n^j] = m\delta^{ij}\delta_{m+n,0}$. Then, we can introduce the following proposition, which has been shown in [7].

**Proposition 5.7** (Proposition 6.4 in [7])
Let $\mathcal{T}$ be a conformal field theory with central charge $n$, whose simultaneous eigenvalues of $p^i := \alpha_0^i$ form an integral lattice $\Lambda$ such that $\mathrm{rank}(\Lambda) = n$. Then, the CFT $\mathcal{T}$ is isomorphic to the lattice CFT based on the lattice $\Lambda$.

This proposition ensures that the CFT $\mathcal{T}$, which is not a lattice CFT by definition, can be isomorphic to a lattice CFT, when the simultaneous eigenvalues of $\mathcal{T}$ form an integral lattice. Since the NS-NS partition function of a lattice CFT is invariant under modular $S$ transformation: $\tau \to -1/\tau$ only when the lattice is self-dual, a chiral fermionic CFT $\mathcal{T}$, whose NS-NS partition function is modular $S$ invariant, should be isomorphic to a lattice CFT based on a self-dual lattice.

*Proof of Proposition 5.5.* The proof is the same for the two types of orbifold $(\mathcal{T}/G)_{\pm}$ and we denote them by $\mathcal{T}/G$ below for simplicity. Let $\mathcal{T}$ be a chiral fermionic CFT constructed from a singly-even self-dual code $C$. The shift orbifold $\mathcal{T}/H$ has the Cartan subalgebra $\sqrt{2}J_0^{3i}$ for each direction $i = 1, 2, \ldots, n$, while the reflection orbifold $\mathcal{T}/G$ has the Cartan subalgebra $\sqrt{2}J_0^{1i}$ for each direction $i = 1, 2, \ldots, n$. Both orbifold theories contain $V_{\delta\text{-even}}^+$ because this is fixed by reflection $g$ and shift $h$. Correspondingly, for the shift orbifold $\mathcal{T}/H$, the lattice of simultaneous eigenvalues of $\sqrt{2}J_0^{3i}$ contains $\Lambda_{\delta\text{-even}}$.

Under the action of the triality operator $\sigma$, the Cartan subalgebra $\sqrt{2}J_0^{3i}$ is mapped to $\sqrt{2}J_0^{1i}$. For an eigenstate $|\lambda\rangle_+ \in V_{\delta\text{-even}}^+$ of $\sqrt{2}J_0^{3i}$, the triality operator maps it to an eigenstate $\sigma|\lambda\rangle_+ \in V_{\delta\text{-even}}^+$ of $\sqrt{2}J_0^{1i}$ from Proposition 5.6. Since the reflection orbifold includes the space $V_{\delta\text{-even}}^+$, the lattice of simultaneous eigenvalues of $\sqrt{2}J_0^{1i}$ in the reflection orbifold $\mathcal{T}/G$ contains $\Lambda_{\delta\text{-even}}$ as a sublattice. Note that we have

$$(\Lambda_{\delta\text{-even}})^* = \Lambda_{\delta\text{-even}} \sqcup \Lambda_{\delta\text{-odd}} \sqcup (\Lambda_{\delta\text{-even}} + \tfrac{\delta}{2}) \sqcup (\Lambda_{\delta\text{-odd}} + \tfrac{\delta}{2}). \tag{137}$$

Thus, the lattice of simultaneous eigenvalues of the reflection orbifold $\mathcal{T}/G$, which are a self-dual lattice containing $\Lambda_{\delta\text{-even}}$, is restricted to $\Lambda(C)$ and $\Lambda_\delta^{\text{orb}} := \Lambda_\delta^{\text{orb}+} \cong \Lambda_\delta^{\text{orb}-}$. In the rest of proof, we identify which lattice gives simultaneous eigenvalues in the reflection orbifold $\mathcal{T}/G$.

One can distinguish the two lattices $\Lambda(C)$ and $\Lambda_\delta^{\text{orb}}$ by the number of the weight-one operators $|V_1(\Lambda(C))|$, $|V_1(\Lambda_\delta^{\text{orb}})|$ where we denote by $V_1(\Lambda)$ the space of weight-one operators in the lattice CFT based on $\Lambda$. For the lattice CFT based on $\Lambda(C)$, we have

$$|V_1(\Lambda(C))| = 3n + 16|C_4|, \quad n \in 8\mathbb{Z}, \tag{138}$$

where $|C_4|$ is the number of codewords $c$ with $\mathrm{wt}(c) = 4$. Here, $3n$ comes from the generators of $SU(2)^n$ symmetry and $16|C_4|$ comes from the vertex operators $: \exp\{i\lambda \cdot X(z)\} :$ where $\lambda = ((\pm 1)^4, 0^{n-4})/\sqrt{2}$. On the other hand, the corresponding number for the lattice CFT based on $\Lambda_\delta^{\text{orb}}$ is

$$|V_1(\Lambda_\delta^{\text{orb}})| = n + 8|C_4|, \quad n \in 8\mathbb{Z}_{\geq 3}, \tag{139}$$

since after orbifolding, $J_1^i(z)$ and $J_2^i(z)$ are projected out and only the half of vertex operators $: \exp\{i\lambda \cdot X(z)\} :$ where $\lambda = ((\pm 1)^4, 0^{n-4})/\sqrt{2}$ survive. This shows that $|V_1(\Lambda_\delta^{\text{orb}})|$ is strictly smaller than $|V_1(\Lambda(C))|$ for $n \geq 24$, and we can distinguish the two lattices by their spectra, i.e., the partition functions. We can observe that the NS-NS partition function of the lattice CFT with $\Lambda_\delta^{\text{orb}}$ (117) is the same as that of the reflection orbifold (78). Thus, the simultaneous eigenvalues of the reflection orbifold $\mathcal{T}/G$ form the lattice $\Lambda_\delta^{\text{orb}}$. From Proposition 5.7, we can conclude that, for $n \geq 24$, the reflection orbifold $\mathcal{T}/G$ is isomorphic to the lattice CFT based on $\Lambda_\delta^{\text{orb}}$, which is the shift orbifold $\mathcal{T}/H$.

To complete the proof, we separately consider $n = 8$ and $n = 16$. For $n = 8$, the odd self-dual lattice is unique: $\Lambda(C) = \Lambda_\delta^{\text{orb}} = \mathbb{Z}^8$. Thus, the simultaneous eigenvalues of the reflection orbifold form the lattice $\mathbb{Z}^8$ and the equivalence is trivial. Next, consider $n = 16$ (see Fig. 5 for the result). Compared with $n \geq 24$, the weight-one operators for $\Lambda_\delta^{\text{orb}}$ additionally come from the vertex operators $: \exp\{i\lambda \cdot X(z)\} :$ where $\lambda = (\pm 1, \pm 1, \ldots, \pm 1)/2\sqrt{2}$ such that $\delta \cdot \lambda \in 2\mathbb{Z}$. The number of such operators is $|C_0|$: the total number of doubly-even codewords. Thus, the total number of the weight-one operators in the lattice CFT based on $\Lambda_\delta^{\text{orb}}$ is

$$|V_1(\Lambda_\delta^{\text{orb}})| = n + 8|C_4| + |C_0|, \quad n = 16. \tag{140}$$

When $3n + 16|C_4|$ for $\Lambda(C)$ and $n + 8|C_4| + |C_0|$ for $\Lambda_\delta^{\text{orb}}$ are different, we can straightforwardly show the equivalence between the reflection and shift orbifolds from the same argument as $n \geq 24$. There are five singly-even self-dual codes of length 16. Four of them satisfy $3n + 16|C_4| \neq n + 8|C_4| + |C_0|$. For the last one, $3n + 16|C_4| = n + 8|C_4| + |C_0|$, and $\Lambda(C)$ and $\Lambda_\delta^{\text{orb}}$

cannot be distinguished from the number of weight-one operators. However, we can show that $\Lambda(C) = \Lambda_\delta^{\mathrm{orb}} = (D_8^2)^+$ where $(D_8^2)^+$ is the extremal odd self-dual lattice with minimum norm 2. Thus, the reflection orbifold $\mathcal{T}/G$ is isomorphic to the shift orbifold $\mathcal{T}/H$ for $n = 8, 16$. $\quad\square$

# 6 Shift orbifold for nonbinary codes

In this section, we give the expression of the NS-NS partition function $Z_{(\mathcal{T}/H)_\pm}[0,0]$ and the R-R partition function $Z_{(\mathcal{T}/H)_\pm}[1,1]$ of the shift orbifold theory constructed from a code over $\mathbb{Z}_k$, using the weight enumerator of the code. The other partition functions can be obtained in a similar method or by the modular transformation from $Z_{(\mathcal{T}/H)_\pm}[0,0]$.

Let $C \subset \mathbb{Z}_k^n$ be a Type I code with even $k$ or a self-dual code with odd $k$, $\mathcal{T}$ a CFT constructed from the lattice $\Lambda := \Lambda(C)$, and $H$ a shift $\mathbb{Z}_2$ symmetry generated by $h : X \to X + \pi\delta$ where $\delta \in \Lambda(C)$ will be specified later. From the construction, any vector in $\Lambda(C)$ can be written as

$$\lambda = \frac{1}{\sqrt{k}}(c + km), \quad c \in C, \quad m \in \mathbb{Z}^n. \tag{141}$$

For $\delta \in \Lambda(C)$, we define

$$C_{\delta\text{-even}} = \left\{ c \in C \,\middle|\, \frac{1}{\sqrt{k}} c \cdot \delta \in 2\mathbb{Z} \right\}, \qquad C_{\delta\text{-odd}} = \left\{ c \in C \,\middle|\, \frac{1}{\sqrt{k}} c \cdot \delta \in 2\mathbb{Z}+1 \right\}, \tag{142}$$

$$\Gamma_{\delta\text{-even}} = \left\{ \sqrt{k}m \in \sqrt{k}\mathbb{Z}^n \,\middle|\, \sqrt{k}m \cdot \delta \in 2\mathbb{Z} \right\}, \quad \Gamma_{\delta\text{-odd}} = \left\{ \sqrt{k}m \in \sqrt{k}\mathbb{Z}^n \,\middle|\, \sqrt{k}m \cdot \delta \in 2\mathbb{Z}+1 \right\}, \tag{143}$$

which satisfy $C = C_{\delta\text{-even}} \sqcup C_{\delta\text{-odd}}$ and $\sqrt{k}\mathbb{Z}^n = \Gamma_{\delta\text{-even}} \sqcup \Gamma_{\delta\text{-odd}}$. Then, $\Lambda_{\delta\text{-even/odd}}$ can be written as

$$\Lambda_{\delta\text{-even}} = \left( \frac{1}{\sqrt{k}} C_{\delta\text{-even}} + \Gamma_{\delta\text{-even}} \right) \sqcup \left( \frac{1}{\sqrt{k}} C_{\delta\text{-odd}} + \Gamma_{\delta\text{-odd}} \right),$$
$$\Lambda_{\delta\text{-odd}} = \left( \frac{1}{\sqrt{k}} C_{\delta\text{-even}} + \Gamma_{\delta\text{-odd}} \right) \sqcup \left( \frac{1}{\sqrt{k}} C_{\delta\text{-odd}} + \Gamma_{\delta\text{-even}} \right). \tag{144}$$

When we write

$$\Gamma'_{\delta\text{-even}} = \Gamma_{\delta\text{-even}} + \frac{1}{2}\delta, \qquad \Gamma'_{\delta\text{-odd}} = \Gamma_{\delta\text{-odd}} + \frac{1}{2}\delta, \tag{145}$$

the momentum lattices of the orbifold theories are

$$\Lambda_\delta^{\mathrm{orb}+} = \left( \frac{1}{\sqrt{k}} C_{\delta\text{-even}} + \left(\Gamma_{\delta\text{-even}} \sqcup \Gamma'_{\delta\text{-even}}\right) \right) \sqcup \left( \frac{1}{\sqrt{k}} C_{\delta\text{-odd}} + \left(\Gamma_{\delta\text{-odd}} \sqcup \Gamma'_{\delta\text{-odd}}\right) \right), \tag{146}$$

$$\Lambda_\delta^{\mathrm{orb}-} = \left( \frac{1}{\sqrt{k}} C_{\delta\text{-even}} + \left(\Gamma_{\delta\text{-even}} \sqcup \Gamma'_{\delta\text{-odd}}\right) \right) \sqcup \left( \frac{1}{\sqrt{k}} C_{\delta\text{-odd}} + \left(\Gamma_{\delta\text{-odd}} \sqcup \Gamma'_{\delta\text{-even}}\right) \right). \tag{147}$$

Throughout this section, we do not care about the ambiguity of the fermion parity $(-1)^F$, i.e. the choice of the characteristic vector $\chi$, since it only causes the sign in the R-R partition function $Z_{(\mathcal{T}/H)_\pm}[1,1]$ for our results.

## 6.1 $k$: even

For a more detailed analysis, it is convenient to choose $\delta$ that does not depend on $C$. The following proposition for codes is useful for this purpose.

**Proposition 6.1**
Every self-dual code $C \subset \mathbb{Z}_{2m}^n$ contains an all-$m$ vector.

*Proof.* For any $c \in C$,

$$c \cdot (m, \ldots, m) \in 2m\mathbb{Z} \quad \Leftrightarrow \quad \sum_{i=1}^{n} c_i \in 2\mathbb{Z} \quad \Leftrightarrow \quad \sum_{i=1}^{n} c_i^2 = c^2 \in 2\mathbb{Z} \tag{148}$$

holds and $c^2 \in 2\mathbb{Z}$ is always satisfied from the self-orthogonality. Thus, the all-$m$ vector is in $C^\perp = C$. $\qquad\square$

In later discussions for even $k = 2m$, we consider $\delta = \frac{1}{\sqrt{2m}}(m, \ldots, m) \in \Lambda(C)$.

**$k \in 4\mathbb{Z} + 2$**

When $k \in 4\mathbb{Z} + 2$, we can take $\delta = \frac{1}{\sqrt{k}}(\frac{k}{2}, \ldots, \frac{k}{2})$ only if $n \in 8\mathbb{Z}$ because of the condition $\delta^2 = \frac{1}{4}kn \in 4\mathbb{Z}$. For other conditions, $\delta$ is not a characteristic vector since $\delta \cdot \lambda \not\equiv \lambda^2 \mod 2$ for $\lambda = \sqrt{k}(1, 0, \ldots, 0) \in \Lambda(C)$ and $\frac{\delta}{2} \notin \Lambda(C)$ is always satisfied from $\frac{\delta}{2} \notin \frac{1}{\sqrt{k}}\mathbb{Z}^n$.

The NS-NS partition function of the orbifold theory can be written as

$$Z_{(\mathcal{T}/H)_\pm}[0,0] = \frac{1}{\eta(\tau)^n} \sum_{\lambda \in \Lambda(C)} \left( \frac{1 + (-1)^{\lambda \cdot \delta}}{2} q^{\frac{1}{2}\lambda^2} + \frac{1 \pm (-1)^{\lambda \cdot \delta}}{2} q^{\frac{1}{2}(\lambda + \frac{1}{2}\delta)^2} \right) \tag{149}$$

$$= \frac{1}{\eta(\tau)^n} \frac{1}{2} \left( W_C(\{g_{a,k}^{0,0}\}) + W_C(\{g_{a,k}^{1,0}\}) + W_C(\{g_{a,k}^{0,1}\}) \pm W_C(\{g_{a,k}^{1,1}\}) \right), \tag{150}$$

where

$$g_{a,k}^{\alpha,\beta}(\tau) = \sum_{m \in \mathbb{Z}} (-1)^{\frac{1}{2}\alpha(km+a)} q^{\frac{k}{2}\left(m + \frac{a}{k} + \beta\frac{1}{4}\right)^2}. \tag{151}$$

From (43) and (50), the R-R partition function can be expressed as

$$Z_{(\mathcal{T}/H)_+}[1,1]$$

$$= \frac{1}{\eta(\tau)^n} \sum_{\xi \in S(\Lambda)} \left( \frac{1 \pm (-1)^{\xi \cdot \delta}}{2} (-1)^{\xi \cdot \chi^{\mathrm{orb}+}} q^{\frac{1}{2}\xi^2} + \frac{1 \pm (-1)^{\xi \cdot \delta}}{2} (-1)^{(\xi + \frac{\delta}{2}) \cdot \chi^{\mathrm{orb}+}} q^{\frac{1}{2}(\xi + \frac{\delta}{2})^2} \right) \tag{152}$$

$$= \frac{1}{\eta(\tau)^n} \sum_{\xi \in S(\Lambda)} \frac{1 \pm (-1)^{\xi \cdot \delta}}{2} (-1)^{\xi \cdot \chi} \left( q^{\frac{1}{2}\xi^2} \pm q^{\frac{1}{2}(\xi + \frac{\delta}{2})^2} \right) \tag{153}$$

$$= \frac{1}{\eta(\tau)^n} \frac{1}{2} \left( \widetilde{W}_C(\{g_{a,k}^{0,0}\}) \pm \widetilde{W}_C(\{g_{a,k}^{1,0}\}) \pm \widetilde{W}_C(\{g_{a,k}^{0,1}\}) + \widetilde{W}_C(\{g_{a,k}^{1,1}\}) \right), \tag{154}$$

where double signs are all $+$ when $n \in 16\mathbb{Z}$ and $-$ when $n \in 16\mathbb{Z} + 8$.

Similarly,

$$Z_{(\mathcal{T}/H)_-}[1,1] = \frac{1}{\eta(\tau)^n} \frac{1}{2} \left( \widetilde{W}_C(\{g_{a,k}^{0,0}\}) \mp \widetilde{W}_C(\{g_{a,k}^{1,0}\}) \mp \widetilde{W}_C(\{g_{a,k}^{0,1}\}) - \widetilde{W}_C(\{g_{a,k}^{1,1}\}) \right), \tag{155}$$

where double signs are all $-$ when $n \in 16\mathbb{Z}$ and $+$ when $n \in 16\mathbb{Z} + 8$.

From simple calculations, $g_{a,k}^{\alpha,\beta}$ has properties such as

$$g_{a,k}^{0,0} = g_{-a,k}^{0,0}, \quad g_{a,k}^{1,0} = (-1)^a g_{-a,k}^{1,0}, \quad g_{a,k}^{0,1} = g_{-a-\frac{k}{2},k}^{0,1}, \quad g_{a,k}^{1,1} = \mathrm{i}^{\frac{k}{2}}(-1)^a g_{-a-\frac{k}{2},k}^{1,1}. \tag{156}$$

In particular, in the binary case ($k = 2$), the relations become

$$g_{0,2}^{1,0} = (\theta_3 \theta_4)^{\frac{1}{2}}, \quad g_{1,2}^{1,0} = 0, \quad g_{0,2}^{0,1} = g_{1,2}^{0,1} = \left( \frac{\theta_2 \theta_3}{2} \right)^{\frac{1}{2}}, \quad g_{0,2}^{1,1} = \mathrm{i} g_{1,2}^{1,1} = \left( \frac{\theta_2 \theta_4}{2} \right)^{\frac{1}{2}}. \tag{157}$$

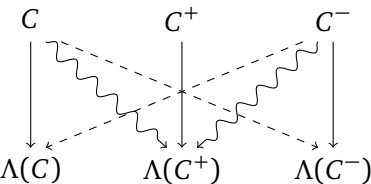

Figure 3: The relation between codes $C$, $C^{\pm}$ over $\mathbb{Z}_{4l}$ and their Construction A lattices. The arrows mean $C \to \Lambda(C)$, $C \rightsquigarrow \Lambda(C)^{\mathrm{orb}+}_{\delta}$, and $C \dashrightarrow \Lambda(C)^{\mathrm{orb}-}_{\delta}$ where $\delta = \frac{1}{\sqrt{4l}}(2l, \ldots, 2l)$. Note that we do not define the orbifold theory by $\delta$ from $C^+$ since $(l, \ldots, l) \subset C^+$.

Thus, the NS-NS partition function of the orbifold theory for a binary code is

$$Z_{(\mathcal{T}/H)_{\pm}}[0,0] = \frac{1}{\eta(\tau)^n} \frac{1}{2}\left( \Theta_{\Lambda(C)} + \left(\theta_3 \theta_4\right)^{\frac{n}{2}} + |C|\left(\frac{\theta_2 \theta_3}{2}\right)^{\frac{n}{2}} \pm \left(|C_0| - |C_2|\right)\left(\frac{\theta_2 \theta_4}{2}\right)^{\frac{n}{2}} \right)$$

$$= \frac{1}{2}\left( Z_{\mathcal{T}}[0,0] + \frac{(\theta_3 \theta_4)^{\frac{n}{2}} + (\theta_2 \theta_3)^{\frac{n}{2}}}{\eta(\tau)^n} \right), \tag{158}$$

which is consistent with Proposition 5.3. For the R-R partition function, the second term is 0 since $\vec{0} \notin S(C)$ and the third and fourth terms are also 0 since the contributions from $C_1$ and $C_3$ cancel each other out as shown using $\mathrm{wt}(s) = n/2 = 0 \pmod 4$, thus

$$Z_{(\mathcal{T}/H)_{\pm}}[1,1] = \frac{1}{2} Z_{\mathcal{T}}[1,1]. \tag{159}$$

### $k \in 4\mathbb{Z}$

When $k = 4l$, $l \in \mathbb{Z}$, $\Gamma_{\delta\text{-odd}}$ is empty and $\Gamma_{\delta\text{-even}} = 2\sqrt{l}\mathbb{Z}^n =: \Gamma$.

Let $C \subset \mathbb{Z}_{4l}^n$ be a Type I code. We can take $\delta = \frac{1}{\sqrt{4l}}(2l, \ldots, 2l)$ only if $ln \in 4\mathbb{Z}$ and $(l, \ldots, l) \notin (C \sqcup S(C))$ because of the gauging conditions for $\delta$. We assume these in the following. In this case, since $\frac{\delta}{2} = \frac{1}{\sqrt{4l}}(l, \ldots, l) \in \frac{1}{\sqrt{4l}}\mathbb{Z}^n$, the effect of the shift $\delta/2$ can be included in the code. In fact, from (146) and (147), we obtain

$$\Lambda^{\mathrm{orb}+}_{\delta} = \frac{1}{\sqrt{p}} C_{\delta\text{-even}} + \left(\Gamma \sqcup \Gamma'\right) = \Lambda(C^+),$$

$$\Lambda^{\mathrm{orb}-}_{\delta} = \left(\frac{1}{\sqrt{p}} C_{\delta\text{-even}} + \Gamma\right) \sqcup \left(\frac{1}{\sqrt{p}} C_{\delta\text{-odd}} + \Gamma'\right) = \Lambda(C^-), \tag{160}$$

where

$$C^+ = C_{\delta\text{-even}} \sqcup (C_{\delta\text{-even}} + (l, \ldots, l)),$$

$$C^- = C_{\delta\text{-even}} \sqcup (C_{\delta\text{-odd}} + (l, \ldots, l)). \tag{161}$$

Thus, the orbifold theories can be constructed from other self-dual codes. Note that we can show $C^{\pm}$ are Type I directly or from the odd self-duality of $\Lambda(C^{\pm})$.

We want to express the weight enumerator polynomial of $C^{\pm}$ by that of $C$.

For any subset $D \subset \mathbb{Z}_{4l}^n$ s.t. $c^2 \in 2\mathbb{Z}$ for all $c \in D$, $D$ can be divided into two subsets

$$D_{\delta\text{-even}} = \left\{ c \in D \;\middle|\; \sum_{i=1}^n c_i \in 4\mathbb{Z} \right\}, \qquad D_{\delta\text{-odd}} = \left\{ c \in D \;\middle|\; \sum_{i=1}^n c_i \in 4\mathbb{Z} + 2 \right\}, \tag{162}$$

and then the corresponding weight enumerator polynomials are

$$W_{D_{\delta\text{-even/odd}}}(\{x_a\}) = \frac{1}{2} \left( W_D(\{x_a\}) \pm W_D(\{i^a x_a\}) \right). \tag{163}$$

Using this, the complete weight enumerator of $C^\pm$ can be written as

$$W_{C^\pm}(\{x_a\}) = W_{C_{\delta\text{-even}}}(\{x_a\}) + W_{C_{\delta\text{-even/odd}}}(\{x_{a+l}\}) \tag{164}$$

$$= \frac{1}{2} \left( W_C(\{x_a\}) + W_C(\{i^a x_a\}) + W_C(\{x_{a+l}\}) \pm W_C(\{i^{a+l} x_{a+l}\}) \right). \tag{165}$$

The NS-NS partition function $Z_{(\mathcal{T}/H)_\pm}[0,0]$ is given by substituting the theta functions as (90a).

To calculate the R-R partition function, we need to identify $C_i^\pm$, $i = 0, 1, 2, 3$ for $C^\pm$. The subsets $C_0^\pm, C_2^\pm$ of $C^\pm$ are defined by $c^2 \in 8l\mathbb{Z}$ or $8l\mathbb{Z} + 4l$, thus

$$C_0^+ = \begin{cases} (C_0)_{\delta\text{-even}} \sqcup ((C_0)_{\delta\text{-even}} + (l, \ldots, l)) & (ln \in 8\mathbb{Z}), \\ (C_0)_{\delta\text{-even}} \sqcup ((C_2)_{\delta\text{-even}} + (l, \ldots, l)) & (ln \in 8\mathbb{Z} + 4), \end{cases} \tag{166}$$

$$C_2^+ = \begin{cases} (C_2)_{\delta\text{-even}} \sqcup ((C_2)_{\delta\text{-even}} + (l, \ldots, l)) & (ln \in 8\mathbb{Z}), \\ (C_2)_{\delta\text{-even}} \sqcup ((C_0)_{\delta\text{-even}} + (l, \ldots, l)) & (ln \in 8\mathbb{Z} + 4), \end{cases} \tag{167}$$

$$C_0^- = \begin{cases} (C_0)_{\delta\text{-even}} \sqcup ((C_2)_{\delta\text{-odd}} + (l, \ldots, l)) & (ln \in 8\mathbb{Z}), \\ (C_0)_{\delta\text{-even}} \sqcup ((C_0)_{\delta\text{-odd}} + (l, \ldots, l)) & (ln \in 8\mathbb{Z} + 4), \end{cases} \tag{168}$$

$$C_2^- = \begin{cases} (C_2)_{\delta\text{-even}} \sqcup ((C_0)_{\delta\text{-odd}} + (l, \ldots, l)) & (ln \in 8\mathbb{Z}), \\ (C_2)_{\delta\text{-even}} \sqcup ((C_2)_{\delta\text{-odd}} + (l, \ldots, l)) & (ln \in 8\mathbb{Z} + 4). \end{cases} \tag{169}$$

When we define $C_1, C_3$ and $C_1^\pm, C_3^\pm$ by $s \in S(C) \cap S(C^\pm)$,

$$C_1^+ = \begin{cases} (C_1)_{\delta\text{-even}} \sqcup ((C_1)_{\delta\text{-even}} + (l, \ldots, l)) & (ln \in 8\mathbb{Z}), \\ (C_1)_{\delta\text{-odd}} \sqcup ((C_3)_{\delta\text{-odd}} + (l, \ldots, l)) & (ln \in 8\mathbb{Z} + 4), \end{cases} \tag{170}$$

$$C_3^+ = \begin{cases} (C_3)_{\delta\text{-even}} \sqcup ((C_3)_{\delta\text{-even}} + (l, \ldots, l)) & (ln \in 8\mathbb{Z}), \\ (C_3)_{\delta\text{-odd}} \sqcup ((C_1)_{\delta\text{-odd}} + (l, \ldots, l)) & (ln \in 8\mathbb{Z} + 4), \end{cases} \tag{171}$$

$$C_1^- = \begin{cases} (C_1)_{\delta\text{-odd}} \sqcup ((C_3)_{\delta\text{-even}} + (l, \ldots, l)) & (ln \in 8\mathbb{Z}), \\ (C_1)_{\delta\text{-even}} \sqcup ((C_1)_{\delta\text{-odd}} + (l, \ldots, l)) & (ln \in 8\mathbb{Z} + 4), \end{cases} \tag{172}$$

$$C_3^- = \begin{cases} (C_3)_{\delta\text{-odd}} \sqcup ((C_1)_{\delta\text{-even}} + (l, \ldots, l)) & (ln \in 8\mathbb{Z}), \\ (C_3)_{\delta\text{-even}} \sqcup ((C_3)_{\delta\text{-odd}} + (l, \ldots, l)) & (ln \in 8\mathbb{Z} + 4). \end{cases} \tag{173}$$

Therefore, the weight enumerator polynomial $\widetilde{W}$ defined in (87) can be written as

$$\widetilde{W}_{C^+}(\{x_a\}) = \frac{1}{2} \left( \widetilde{W}_C(\{x_a\}) \pm \widetilde{W}_C(\{i^a x_a\}) \pm \widetilde{W}_C(\{x_{a+l}\}) + \widetilde{W}_C(\{i^{a+l} x_{a+l}\}) \right), \tag{174}$$

$$\widetilde{W}_{C^-}(\{x_a\}) = \frac{1}{2} \left( \widetilde{W}_C(\{x_a\}) \mp \widetilde{W}_C(\{i^a x_a\}) \mp \widetilde{W}_C(\{x_{a+l}\}) - \widetilde{W}_C(\{i^{a+l} x_{a+l}\}) \right), \tag{175}$$

where double signs correspond to the cases $t \in 8\mathbb{Z}$ (above) and $t \in 8\mathbb{Z}+4$ (below) respectively. The R-R partition function $Z_{(\mathcal{T}/H)_\pm}[1,1]$ is also given by substituting the theta functions as (90d).

## 6.2  $k$: odd

Let $C \subset \mathbb{Z}_k^n$ be a self-dual code with odd $k$.

As already mentioned, from (41), the orbifold theories by $\delta$ and $\delta + 2\lambda_e$, $\lambda_e \in \Lambda_{\delta\text{-even}}$ are the same. Thus, for any $\delta \in \Lambda(C) \subset \frac{1}{\sqrt{k}}\mathbb{Z}^n$ which satisfies the anomaly condition $\delta^2 \in 4\mathbb{Z}$, it is equivalent to consider $k\delta \in \sqrt{k}\mathbb{Z}^n$. Moreover, for $u = \sqrt{k}\delta \in \mathbb{Z}^n$, vectors

$$
\begin{cases}
\sqrt{k}(0^{i-1}, 1, 0^{n-i}), & \text{if } u_i \in 2\mathbb{Z}, \\
\sqrt{k}(0^{i-1}, 2, 0^{n-i}), & \text{if } u_i \in 2\mathbb{Z}+1,
\end{cases}
\tag{176}
$$

for $i \in \{1, \dots, n\}$ are in $\Lambda_{\delta\text{-even}}$ and then we can always get a vector consisting of 0 and $\pm\sqrt{k}$ by adding and subtracting even multiples of these vectors. Therefore, we can consider only $\delta \in \{0, \pm\sqrt{k}\}^n$ without loss of generality.

On the other hand, any $\chi \in \{\pm\sqrt{k}\}^n$ is a characteristic vector of any $\Lambda(C)$.

For simplicity, we assume $\delta \in \{0, +\sqrt{k}\}^n$ where the number of $\sqrt{k}$ is $t \in 4\mathbb{Z}$ (to satisfy $\delta^2 \in 4\mathbb{Z}$) and choose $\chi = \sqrt{k}(1, \dots, 1)$. Let us define

$$
\begin{aligned}
\mathrm{wt}_a^{(1)}(c) &= |\{i \in \{1, \dots, n\} \mid c_i = a, \text{ and } \delta_i = \sqrt{k}\}|, \\
\mathrm{wt}_a^{(0)}(c) &= |\{i \in \{1, \dots, n\} \mid c_i = a, \text{ and } \delta_i = 0\}|,
\end{aligned}
\tag{177}
$$

for $a \in \mathbb{Z}_k$, which satisfy $\mathrm{wt}_a(c) = \mathrm{wt}_a^{(1)}(c) + \mathrm{wt}_a^{(0)}(c)$, and

$$
W_C(\{x_a\}, \{y_a\}) = \sum_{c \in C} \prod_{a \in \mathbb{Z}_k} x_a^{\mathrm{wt}_a^{(1)}(c)} y_a^{\mathrm{wt}_a^{(0)}(c)}.
\tag{178}
$$

From (41), the NS-NS partition function of the orbifold theory is

$$
Z_{(\mathcal{T}/H)_\pm}[0, 0]
\tag{179}
$$
$$
= \frac{1}{\eta(\tau)^n} \frac{1}{2} \left( W_C(\{f_{a,k}^{0,0}\}, \{f_{a,k}^{0,0}\}) + W_C(\{f_{a,k}^{1,0}\}, \{f_{a,k}^{0,0}\}) + W_C(\{f_{a,k}^{0,1}\}, \{f_{a,k}^{0,0}\}) \pm W_C(\{f_{a,k}^{1,1}\}, \{f_{a,k}^{0,0}\}) \right).
$$

From (49), the R-R partition function for $\Lambda_\delta^{\mathrm{orb}+}$ is

$$
Z_{(\mathcal{T}/H)_+}[1, 1]
$$
$$
= \frac{1}{\eta(\tau)^n} \sum_{\lambda \in \Lambda} \left( \frac{1 \pm (-1)^{\lambda \cdot \delta}}{2} (-1)^{(\lambda + \frac{\chi}{2}) \cdot \chi} q^{\frac{1}{2}(\lambda + \frac{\chi}{2})^2} \pm \frac{1 \pm (-1)^{\lambda \cdot \delta}}{2} (-1)^{(\lambda + \frac{\chi + \delta}{2}) \cdot \chi} q^{\frac{1}{2}(\lambda + \frac{\chi + \delta}{2})^2} \right)
\tag{180}
$$
$$
= \frac{1}{\eta(\tau)^n} \frac{1}{2} \left( W_C(\{f_{a,k}^{1,1}\}, \{f_{a,k}^{1,1}\}) \pm W_C(\{f_{a,k}^{0,1}\}, \{f_{a,k}^{1,1}\}) \pm W_C(\{f_{a,k}^{1,0}\}, \{f_{a,k}^{1,1}\}) + W_C(\{f_{a,k}^{0,0}\}, \{f_{a,k}^{1,1}\}) \right),
$$

where double signs are all $+$ when $t \in 8\mathbb{Z}$ and $-$ when $t \in 8\mathbb{Z}+4$. Technically, to define the fermion parity, we chose $\chi$ when $t \in 8\mathbb{Z}$ and $\chi + 2\lambda_o$ where $\lambda_o \in \Lambda_{\delta\text{-odd}}$ s.t. $\lambda_o^2 \in 2\mathbb{Z}$ when $t \in 8\mathbb{Z}+4$.

Similarly, from (51), we obtain the R-R partition function for $\Lambda_\delta^{\mathrm{orb}-}$

$$
Z_{(\mathcal{T}/H)_-}[1, 1]
\tag{181}
$$
$$
= \frac{1}{\eta(\tau)^n} \frac{1}{2} \left( W_C(\{f_{a,k}^{1,1}\}, \{f_{a,k}^{1,1}\}) \mp W_C(\{f_{a,k}^{0,1}\}, \{f_{a,k}^{1,1}\}) \mp W_C(\{f_{a,k}^{1,0}\}, \{f_{a,k}^{1,1}\}) - W_C(\{f_{a,k}^{0,0}\}, \{f_{a,k}^{1,1}\}) \right),
$$

where double signs are all $-$ when $t \in 8\mathbb{Z}$ and $+$ when $t \in 8\mathbb{Z}+4$.

**Interpretation by codes**   Let $C \subset \mathbb{Z}_k^n$ be a Type I code with even $k$ or a self-dual code with odd $k$. We define a code over $\mathbb{Z}_{4k}$ with the same length $n$ by

$$\widehat{C} = \{\hat{c} \in \mathbb{Z}_{4k}^n \mid \hat{c} \equiv 2c \mod 2k, \ c \in C\}, \tag{182}$$

which is Type I and satisfies $\Lambda(C) = \Lambda(\widehat{C})$. After orbifolding by $\delta = \frac{1}{\sqrt{k}}(c + km)$, $c \in C$, $m \in \mathbb{Z}^n$, which is not a characteristic vector and satisfies $\frac{\delta}{2} \notin \Lambda(C)$, $\delta^2 \in 4\mathbb{Z}$, the theory can be expressed as

$$
\begin{aligned}
\Lambda(C)_\delta^{\mathrm{orb}+} &= \Lambda(\widehat{C}^+), & \widehat{C}^+ &= \widehat{C}_{\delta\text{-even}} \sqcup \left(\widehat{C}_{\delta\text{-even}} + c + k\tilde{m}\right), \\
\Lambda(C)_\delta^{\mathrm{orb}-} &= \Lambda(\widehat{C}^-), & \widehat{C}^- &= \widehat{C}_{\delta\text{-even}} \sqcup \left(\widehat{C}_{\delta\text{-odd}} + c + k\tilde{m}\right),
\end{aligned}
\tag{183}
$$

where $\tilde{m} \in \{0, 1, 2, 3\}^n$ s.t. $\tilde{m} \equiv m \mod 4$. We can confirm the self-duality of $\widehat{C}^\pm$ explicitly. Note that $(c + k\tilde{m})^2 \in 4k\mathbb{Z}$ is guaranteed from $\delta^2 \in 4\mathbb{Z}$.

# 7   Applications

This section is devoted to the applications of our code construction of chiral fermionic CFTs. We first demonstrate the shift orbifolds of the lattice CFT based on $\mathbb{Z}^8$ using various types of code including both binary and nonbinary ones. Next, we construct the reflection and shift orbifold theories with central charge 16 from binary codes, which reproduces the classification of lattices and CFTs. Finally, we give the method searching for supersymmetric CFTs and provide several fermionic CFTs with $c \geq 24$ of our interest.

## 7.1   Examples of shift orbifold with $\mathbb{Z}^8$

$\boldsymbol{k = 2.}$   There is only one odd self-dual lattice in dimension $n = 8$: $\mathbb{Z}^8$. A singly-even self-dual code $C$ over $\mathbb{Z}_2$ is also unique, whose generator matrix is

$$
\begin{bmatrix}
1 & 1 & 0 & 0 & 0 & 0 & 0 & 0 \\
0 & 0 & 1 & 1 & 0 & 0 & 0 & 0 \\
0 & 0 & 0 & 0 & 1 & 1 & 0 & 0 \\
0 & 0 & 0 & 0 & 0 & 0 & 1 & 1
\end{bmatrix}.
\tag{184}
$$

From the odd self-duality, the lattices $\Lambda(C)$ and $\Lambda(C)_\delta^{\mathrm{orb}\pm}$ must be $\mathbb{Z}^8$ up to rotations for any $\delta \in \Lambda(C)$ satisfying the gauging condition. In particular, the case $\delta = \frac{1}{\sqrt{2}}(1, \ldots, 1)$ is illustrated in Figure 4. The weight enumerators of $C$ are

$$W_C(x_0, x_1) = (x_0^2 + x_1^2)^4, \qquad \widetilde{W}_C(x_0, x_1) = 0, \tag{185}$$

and from (90) the NS-NS and R-R partition functions of the CFT $\mathcal{T}$ constructed from $\Lambda(C)$ are

$$
\begin{aligned}
Z_{\mathcal{T}}[0, 0] &= \frac{1}{\eta(\tau)^8} W_C(\theta_3(2\tau), \theta_2(2\tau)) = \frac{\theta_3(\tau)^8}{\eta(\tau)^8}, \\
Z_{\mathcal{T}}[1, 1] &= \frac{1}{\eta(\tau)^8} \widetilde{W}_C(\theta_3(2\tau), \theta_2(2\tau)) = 0.
\end{aligned}
\tag{186}
$$

From (158), (159), and the identity $\theta_3(\tau)^4 = \theta_2(\tau)^4 + \theta_4(\tau)^4$, those of the shift orbifold theory $(\mathcal{T}/H)_\pm$ are

$$Z_{(\mathcal{T}/H)_\pm}[0, 0] = \frac{\theta_3^8 + (\theta_3\theta_4)^4 + (\theta_2\theta_3)^4}{2\eta(\tau)^8} = \frac{\theta_3(\tau)^8}{\eta(\tau)^8}, \qquad Z_{(\mathcal{T}/H)_\pm}[1, 1] = 0, \tag{187}$$

which are indeed the same as the original theory $\mathcal{T}$.

**$k = 3$.** A similar argument can be made for codes over other $\mathbb{Z}_k$. For $k = 3$, a self-dual code $C'$ over $\mathbb{Z}_3$ is also unique, whose generator matrix is

$$
\begin{bmatrix}
1 & 0 & 0 & 0 & 1 & 1 & 0 & 0 \\
0 & 1 & 0 & 0 & 1 & 2 & 0 & 0 \\
0 & 0 & 1 & 0 & 0 & 0 & 1 & 1 \\
0 & 0 & 0 & 1 & 0 & 0 & 1 & 2
\end{bmatrix}. \tag{188}
$$

If we choose $\delta = \sqrt{3}(1, 1, 1, 1, 0, 0, 0, 0)$, the modified weight enumerator of $C'$ is

$$
W_{C'}(\{x_a\}, \{y_a\}) = (x_0^2 y_0^2 + (x_1 + x_2) y_0 (x_2 y_1 + x_1 y_2) + x_0 (y_1 + y_2)(x_1 y_1 + x_2 y_2))^2, \tag{189}
$$

and by substituting $f_{a,3}^{\alpha,\beta}$ according to (179), (180), and (181), we obtain the same result as (187).

**$k = 4$.** For $k = 4$, we consider a self-dual code $C''$ over $\mathbb{Z}_4$ generated by

$$
\begin{bmatrix}
1 & 0 & 0 & 0 & 0 & 1 & 1 & 3 \\
0 & 1 & 0 & 0 & 1 & 3 & 0 & 3 \\
0 & 0 & 1 & 0 & 1 & 0 & 1 & 1 \\
0 & 0 & 0 & 1 & 3 & 3 & 1 & 0
\end{bmatrix}. \tag{190}
$$

The weight enumerator is

$$
\begin{aligned}
W_{C''}(\{x_a\}) = {}& x_0^8 + x_1^4 x_2^4 + x_1^7 x_3 + 6 x_1^2 x_2^4 x_3^2 + 7 x_1^5 x_3^3 + 3 x_0^3 x_2 (x_1 + x_3)^4 \\
& + 6 x_0^2 x_2^2 (x_1 + x_3)^4 + 3 x_0 x_2^3 (x_1 + x_3)^4 + x_2^4 (x_2^4 + x_3^4) \\
& + x_1^3 (4 x_2^4 x_3 + 7 x_3^5) + x_1 (4 x_2^4 x_3^3 + x_3^7) + x_0^4 (14 x_2^4 + (x_1 + x_3)^4).
\end{aligned} \tag{191}
$$

By substituting $x_a = \Theta_{a,2}/\eta$, we arrive at (186). One can take the shift orbifold by $\delta = (1, 1, \ldots, 1)$ since this code satisfies $n \in 4\mathbb{Z}$ and $(1, 1, \ldots, 1) \notin (C'' \sqcup S(C''))$. The shift orbifold theories $(\mathcal{T}/H)_\pm$ are lattice CFTs constructed from new codes $(C'')^\pm$, which are generated by the following matrices, respectively:

$$
G^+ = \begin{bmatrix}
0 & 1 & 0 & 0 & 1 & 3 & 0 & 3 \\
0 & 0 & 1 & 0 & 1 & 0 & 1 & 1 \\
0 & 0 & 0 & 1 & 3 & 3 & 1 & 0 \\
3 & 0 & 0 & 3 & 1 & 2 & 0 & 3
\end{bmatrix}, \qquad
G^- = \begin{bmatrix}
0 & 1 & 0 & 0 & 1 & 3 & 0 & 3 \\
0 & 0 & 1 & 0 & 1 & 0 & 1 & 1 \\
0 & 0 & 0 & 1 & 3 & 3 & 1 & 0 \\
2 & 3 & 2 & 0 & 1 & 3 & 0 & 1 \\
2 & 0 & 2 & 3 & 3 & 1 & 1 & 2
\end{bmatrix}. \tag{192}
$$

Note that after shift orbifolding, the Construction A lattices are $\mathbb{Z}^8$ again: $\Lambda((C'')^\pm) = \mathbb{Z}^8$ since an 8-dimensional odd self-dual lattice is unique.

## 7.2 Classification at $c = 16$

There are 6 odd self-dual lattices in dimension $n = 16$:

$$
\mathbb{Z}^{16}, \qquad E_8 \times \mathbb{Z}^8, \qquad D_{12}^+ \times \mathbb{Z}^4, \qquad (E_7^2)^+ \times \mathbb{Z}^2, \qquad A_{15}^+ \times \mathbb{Z}, \qquad (D_8^2)^+. \tag{193}
$$

Since norm 1 vectors exist only in $\mathbb{Z}$, the lattices can be identified by the number of them $(32, 16, 8, 4, 2 \text{ and } 0)$. For simplicity, it is denoted by $N_1$ as $N_1(E_8 \times \mathbb{Z}^8) = 16$.

For a lattice constructed from a code $C$ over $\mathbb{Z}_2$, norm 1 vectors are generated from only $c \in C$ with $\text{wt}(c) = 2$. For example, if $(1, 1, 0, \ldots, 0) \in C$, then $\frac{1}{\sqrt{2}}(\pm 1, \pm 1, 0, \ldots, 0) \in \Lambda(C)$.

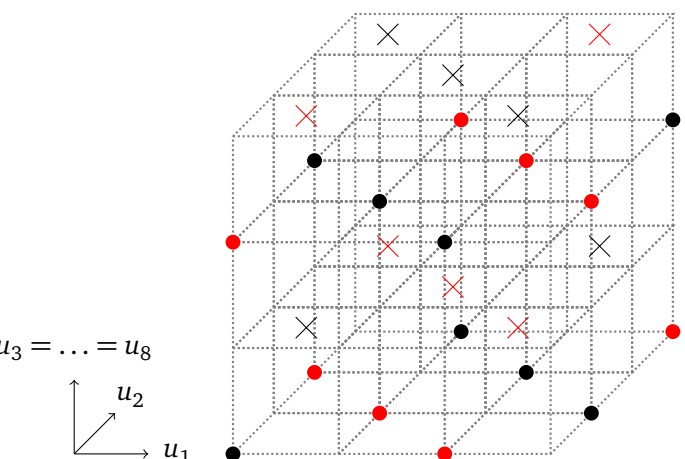

$$u_3 = \ldots = u_8$$

$u_2$

$u_1$

Figure 4: The 3-dimensional slice of $\mathbb{R}^8$ with $u_3 = u_4 = \ldots = u_8$, where $u_i$ $(i = 1, \ldots, 8)$ are the coordinates. Black circle: $\Lambda(C)_{\delta\text{-even}}$; Red circle: $\Lambda(C)_{\delta\text{-odd}}$; Black cross: $\Lambda(C)_{\delta\text{-even}} + \frac{\delta}{2}$; Red cross: $\Lambda(C)_{\delta\text{-odd}} + \frac{\delta}{2}$.

Let $C_{(i)} \subset \mathbb{Z}_2^{16}$, $i = 1, \ldots, 5$ be the single-even self-dual codes with length $n = 16$ (the index of the codes conforms to the order of [54]). The number of weight 2 codewords in $C_{(i)}$ is $4, 2, 8, 1$ and $0$ respectively and $N_1(\Lambda(C_{(i)}))$ is four times that. For the orbifold lattice $\Lambda(C_{(i)})^{\text{orb}\pm}_\delta$ by $\delta = \frac{1}{\sqrt{2}}(1, \ldots, 1)$, from the discussion in section 5, the numbers of norm 1 vectors in $\Lambda(C_{(i)})_{\delta\text{-even}}$ and $\Lambda(C_{(i)})_{\delta\text{-odd}}$ are the same and that in $\Lambda(C_{(i)}) + \frac{\delta}{2}$ is 0. Therefore, $N_1(\Lambda(C_{(i)})^{\text{orb}\pm}_\delta) = \frac{1}{2}N_1(\Lambda(C_{(i)}))$.

The relation between codes and lattices are summarized in Figure 5. As for lattices to CFTs, the straight construction is obvious from their names and $\mathbb{Z}$ corresponding to two Majorana-Weyl fermions $2\psi$. The theory $\overline{(E_8)_2} \times 1\psi$ is determined by the fact that the number of $\psi$ is halved after orbifolding, as discussed near (80). It is worth noting that this theory cannot be directly constructed from the lattice.

The NS-NS partition functions can be obtained from (90a) using the weight enumerators of the codes and the relation (158) between the original theory and the orbifold theory. For example,

$$Z_{\overline{(E_8)_2} \times 1\psi}[0,0] = \frac{\theta_3(\tau)^{16}}{\eta(\tau)^{16}} - 31\frac{\theta_3(\tau)^4}{\eta(\tau)^4} = q^{-\frac{2}{3}} + q^{-\frac{1}{6}} + 248q^{\frac{1}{3}} + 4124q^{\frac{5}{6}} + O(q^{\frac{4}{3}}), \tag{194}$$

$$Z_{\overline{(D_8)_1^2}}[0,0] = \frac{(\theta_3(\tau)\theta_4(\tau))^8 + (\theta_2(\tau)\theta_3(\tau))^8}{\eta(\tau)^{16}} = q^{-\frac{2}{3}} + 240q^{\frac{1}{3}} + 4096q^{\frac{5}{6}} + O(q^{\frac{4}{3}}). \tag{195}$$

Note that $Z_{\overline{(D_8)_1^2}}[0,0]$ is fixed solely by (158) since the theory $\overline{(D_8)_1^2}$ returns to itself under orbifolding. The R-R partition functions are all 0.

## 7.3 Search for supersymmetry

In [56,57], three conditions that strongly suggest the existence of the $\mathcal{N} = 1$ supersymmetry are proposed. In this section, we discuss whether the orbifold theory satisfies these conditions. In particular, we consider the shift orbifold with $\delta = \frac{1}{\sqrt{2}}(1, \ldots, 1)$ for lattices constructed from binary codes and the reflection orbifold for general self-dual lattices. The result from the shift orbifold is contained within that from the reflection orbifold since $\mathcal{T}/H \cong \mathcal{T}/G$ in Proposition 5.5; however, the considerations in both orbifolds provide different implications regarding the structure of the theory.

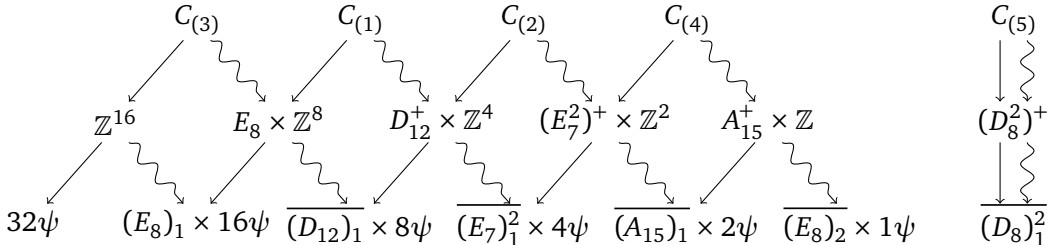

Figure 5: All singly-even self-dual codes over $\mathbb{Z}_2$, odd self-dual lattices, and chiral fermionic CFTs at $n = 16$. The arrows in the first row mean $C_{(i)} \to \Lambda(C_{(i)})$ and $C_{(i)} \rightsquigarrow \Lambda(C_{(i)})^{\text{orb}\pm}_\delta$ for $\delta = \frac{1}{\sqrt{2}}(1, \ldots, 1)$, and that in the second row mean $\to$: the straight construction and $\rightsquigarrow$: the reflection orbifold. See [55] and [10] for the names of lattices and CFTs, respectively.

**Shift orbifold** When $k = 2$, $n \in 8\mathbb{Z}$ and $\delta = \frac{1}{\sqrt{2}}(1, \ldots, 1)$, from the fact that $\text{wt}(s) = n/2 = 0$ (mod 4), $S(C)_{\delta\text{-even}} = S(C)$ and $S(C)_{\delta\text{-odd}}$ is empty. Thus, from (50) and (52),

$$S(\Lambda^{\text{orb}+}_\delta) = \begin{cases} \frac{1}{\sqrt{2}}S(C) + \left(\Gamma_{\delta\text{-even}} \sqcup \Gamma'_{\delta\text{-even}}\right) & (n \in 16\mathbb{Z}), \\ \frac{1}{\sqrt{2}}S(C) + \left(\Gamma_{\delta\text{-odd}} \sqcup \Gamma'_{\delta\text{-odd}}\right) & (n \in 16\mathbb{Z} + 8), \end{cases} \tag{196}$$

$$S(\Lambda^{\text{orb}-}_\delta) = \begin{cases} \frac{1}{\sqrt{2}}S(C) + \left(\Gamma_{\delta\text{-odd}} \sqcup \Gamma'_{\delta\text{-even}}\right) & (n \in 16\mathbb{Z}), \\ \frac{1}{\sqrt{2}}S(C) + \left(\Gamma_{\delta\text{-even}} \sqcup \Gamma'_{\delta\text{-odd}}\right) & (n \in 16\mathbb{Z} + 8). \end{cases} \tag{197}$$

In the following, in discussions that are valid for either $\Lambda^{\text{orb}+}_\delta$ or $\Lambda^{\text{orb}-}_\delta$, we will simply write $\Lambda^{\text{orb}}_\delta$.

The first condition is that the NS sector contains a spin-$\frac{3}{2}$ Virasoro primary operator. For a CFT constructed from an odd self-dual lattice $\Lambda$, this is equivalent to the existence of $\lambda \in \Lambda$ s.t. $\lambda^2 = 3$. When $\Lambda(C)$ contains such a vector, from $\Lambda(C) \subset \frac{1}{\sqrt{2}}\mathbb{Z}^n$, it should be

$$\frac{1}{\sqrt{2}}(1, 1, 2, 0, \ldots, 0), \quad \text{or} \quad \frac{1}{\sqrt{2}}(1, 1, 1, 1, 1, -1, 0, \ldots, 0), \tag{198}$$

up to permutation and signs. From $\sqrt{2}\mathbb{Z}^n \subset \Lambda(C)$, vectors with different signs for arbitrary elements are also in $\Lambda(C)$ and among them there is always a vector in $\Lambda(C)_{\delta\text{-even}}$. (We chose (198) as such an example.) Since $\Lambda(C)_{\delta\text{-even}} \subset \Lambda(C)_{\delta\text{-orb}}$, we can conclude that if the original theory contains a spin-$\frac{3}{2}$ primary operator, then the orbifold theory also does.

The second condition is that any primary operator in the R sector satisfies $h \geq \frac{n}{24}$ where $h$ is the conformal weight. For the CFT from the lattice $\Lambda$, this is equivalent to any $\xi \in S(\Lambda)$ satisfying $\xi^2 \geq \frac{n}{12}$. For $S(\Lambda^{\text{orb}}_\delta)$, $\xi \in \frac{1}{\sqrt{2}}S(C) + \Gamma'_{\delta\text{-even/odd}}$ always satisfies it because the norm is greater than or equal to $(\frac{1}{2\sqrt{2}})^2 n = \frac{n}{8}$ for $\Gamma'_{\delta\text{-even}}$ and $(\frac{1}{2\sqrt{2}})^2(n-1) + (\frac{3}{2\sqrt{2}})^2 = \frac{n}{8} + 1$ for $\Gamma'_{\delta\text{-odd}}$. Since the other sector $\frac{1}{\sqrt{2}}S(C) + \Gamma_{\delta\text{-even/odd}}$ is in the shadow of the original lattice $S(\Lambda(C))$, we can conclude that if the original theory satisfies $h \geq \frac{n}{24}$, then the orbifold theory also does.

The third condition is that the R-R partition function is constant, which can already be confirmed from (159). More precisely, in the original theory, the contribution from $\frac{1}{\sqrt{2}}S(C) + \Gamma_{\delta\text{-even}}$ and $\frac{1}{\sqrt{2}}S(C) + \Gamma_{\delta\text{-odd}}$ are equal from (102) and thus both are half of $Z_{\mathcal{T}}[1, 1]$. In addition, the contributions from $\frac{1}{\sqrt{2}}C_1 + \Gamma'_{\delta\text{-even(odd)}}$ and $\frac{1}{\sqrt{2}}C_3 + \Gamma'_{\delta\text{-even(odd)}}$ cancel each other out from (110).

Thus, if the original theory has the $\mathcal{N} = 1$ supersymmetry, then the orbifold theory also satisfies all "SUSY conditions", which strongly suggests the existence of the $\mathcal{N} = 1$ supersymmetry. Note that for nonbinary cases, the SUSY conditions are not necessarily preserved by the shift orbifolding.

**Reflection orbifold**  Let $\Lambda \subset \mathbb{R}^n$ be an odd self-dual lattice. First, we assume that the lattice CFT constructed from $\Lambda$ satisfies the SUSY conditions and prove its reflection orbifold theory also does.

For the first condition, the lattice $\Lambda$ contains a vector $\lambda \in \Lambda$ s.t. $\lambda^2 = 3$, thus the reflection orbifold theory has the state $|\lambda\rangle + |-\lambda\rangle$ with the same spin.

For the second condition, the twisted R sector always satisfies the condition since the conformal weights of the ground states $|\Omega_j\rangle$ is $h = \frac{n}{16}$ from the discussion in section 3.3. The untwisted R sector is contained in the original theory and therefore also satisfies the condition.

The third condition is obvious from (79).

Next, we examine the converse direction: whether the original theory satisfies the SUSY conditions given that the reflection orbifold theory does.

For the first condition, even if the orbifold theory includes a spin-3/2 operator in the NS sector, it may come from the twisted sector. Since the weight of the twisted ground states is $h = \frac{n}{16}$, when $n \geq 32$, the spin-3/2 operator must be in the untwisted sector. In the cases of $n = 8, 16$ and 24, the complete classification of self-dual lattices is known [58], and it can be directly verified that all such lattices contain vectors of norm 3. Consequently, the original theory satisfies the condition.

For the second condition, the R sector of the orbifold theory includes all states of the form $|\chi\rangle \pm |-\chi\rangle$, $\chi \in S(\Lambda)$ (+ for $(\mathcal{T}/G)_+$ and − for $(\mathcal{T}/G)_-$). Therefore, all states in the R sector of the original theory, spanned by $|\chi\rangle$ and its descendants, satisfy the condition.

The third condition is also obvious from (79).

Thus, we conclude that the SUSY conditions for the original theory and those for the reflection orbifold theory are equivalent.

## 7.4  Chiral fermionic CFTs with $c = 24$

In this subsection, we construct chiral fermionic CFTs with central charge 24 from singly-even self-dual codes over $\mathbb{Z}_2$ and their orbifolds. As an example, we identify binary codes of length 24 that yield the "Beauty and the Beast" SCFT [13] and the Baby Monster CFT [14] with a Majorana-Weyl fermion.

### 7.4.1  "Beauty and the Beast" SCFT

Let $C \subset \mathbb{Z}_2^{24}$ be a singly-even self-dual code generated by ( [54])

$$\begin{bmatrix} 1 & 0 & 0 & 0 & 0 & 0 & 0 & 0 & 0 & 0 & 0 & 0 & 0 & 1 & 1 & 1 & 1 & 1 & 0 & 0 & 0 & 0 & 1 & 1 \\ 0 & 1 & 0 & 0 & 0 & 0 & 0 & 0 & 0 & 0 & 0 & 0 & 0 & 1 & 0 & 1 & 1 & 1 & 1 & 1 & 1 & 0 & 0 & 0 \\ 0 & 0 & 1 & 0 & 0 & 0 & 0 & 0 & 0 & 0 & 1 & 0 & 0 & 0 & 0 & 0 & 1 & 0 & 1 & 0 & 0 & 1 & 1 & 0 \\ 0 & 0 & 0 & 1 & 0 & 0 & 0 & 0 & 0 & 0 & 1 & 0 & 0 & 0 & 1 & 1 & 1 & 1 & 1 & 1 & 1 & 1 & 0 & 0 \\ 0 & 0 & 0 & 0 & 1 & 0 & 0 & 0 & 0 & 0 & 1 & 0 & 0 & 1 & 1 & 1 & 1 & 0 & 0 & 1 & 0 & 1 & 1 & 1 \\ 0 & 0 & 0 & 0 & 0 & 1 & 0 & 0 & 0 & 0 & 0 & 0 & 0 & 1 & 0 & 0 & 0 & 1 & 1 & 0 & 1 & 1 & 0 & 0 \\ 0 & 0 & 0 & 0 & 0 & 0 & 1 & 0 & 0 & 0 & 1 & 0 & 0 & 0 & 0 & 1 & 1 & 1 & 0 & 1 & 0 & 1 & 1 & 0 \\ 0 & 0 & 0 & 0 & 0 & 0 & 0 & 1 & 0 & 0 & 1 & 0 & 0 & 1 & 0 & 1 & 0 & 0 & 0 & 1 & 1 & 0 & 1 & 1 \\ 0 & 0 & 0 & 0 & 0 & 0 & 0 & 0 & 1 & 0 & 0 & 0 & 0 & 0 & 1 & 1 & 0 & 1 & 0 & 0 & 0 & 1 & 1 & 0 \\ 0 & 0 & 0 & 0 & 0 & 0 & 0 & 0 & 0 & 1 & 0 & 0 & 0 & 0 & 1 & 0 & 0 & 0 & 1 & 1 & 0 & 1 & 0 & 1 \\ 0 & 0 & 0 & 0 & 0 & 0 & 0 & 0 & 0 & 0 & 0 & 1 & 0 & 1 & 1 & 1 & 1 & 0 & 1 & 0 & 0 & 0 & 0 & 0 \\ 0 & 0 & 0 & 0 & 0 & 0 & 0 & 0 & 0 & 0 & 0 & 0 & 1 & 1 & 1 & 1 & 0 & 1 & 1 & 1 & 1 & 0 & 1 & 1 \end{bmatrix}, \qquad (199)$$

which is the only code with the minimum weight $d(C) = 6$ at $n = 24$ up to permutations.

The lattice $\Lambda(C)$ can be identified as the sixth lattice in Table 17.1a [22] since the neighbors $\Lambda(C_0 \sqcup C_1)$ and $\Lambda(C_0 \sqcup C_3)$ are $D_4^6$ and $A_1^{24}$, which can be confirmed from the Coxeter number

| Code | Lattice | CFT | # spin-$\frac{3}{2}$ | min $E_R$ | $Z_{\mathcal{T}}[1,1]$ |
|---|---|---|---|---|---|
| | | $\mathcal{T}(\Lambda_{A_1^{24}})$ | 4096 | 0 | 96 |
| | $\Lambda_{A_1^{24}}$ | | | | |
| $C$ | | $\mathcal{T}(O_{24})$ | 4096 | 0 | 48 |
| | $O_{24}$ | | | | |
| | | B&B | 4096 | 0 | 24 |

Figure 6: The "Beauty and the Beast" CFT constructed from the code $C$. $O_{24}$ is the odd Leech lattice and $\Lambda_{A_1^{24}}$ is the sixth lattice in Table 17.1a [22], which is associated with the root system $A_1^{24}$. Here, "B&B" stands for "Beauty and the Beast" $\mathcal{N} = 1$ supersymmetric CFT [13]. We also show the number of spin-3/2 operators in the NS sector, the minimum conformal weight in the R sector, and the R-R partition function for each theory.

(see Table 16.1 [22]) and the fact that there is no odd self-dual lattice s.t. its even neighbors are $A_5^4 D_4$ and $A_1^{24}$.

When we consider the shift orbifold theory from $\Lambda(C)$ by $\delta = \frac{1}{\sqrt{2}}(1,\ldots,1)$, $\Lambda_\delta^{\mathrm{orb}\pm}$ is the odd Leech lattice $O_{24}$, which is the only lattice with the minimum norm 3 at $n = 24$. This can be shown from the fact that $d(C) = 6$ and vectors of norm 2 such as $\sqrt{2}(1,0,\ldots,0)$ are in $\Lambda_{\delta\text{-odd}} \not\subset \Lambda_\delta^{\mathrm{orb}\pm}$. Note that this lattice cannot be directly generated by Construction A from codes over $\mathbb{Z}_2$. In addition, the reflection orbifold theory from the odd Leech lattice is known to have the $\mathcal{N} = 1$ supersymmetry [9] and is called "Beauty and the Beast" [13]. These relations are summarized in Figure 6.

Since the weight enumerators of the code $C$ are

$$W_C(x_0, x_1) = x_0^{24} + 64\, x_0^{18}\, x_1^6 + 375\, x_0^{16}\, x_1^8 + 960\, x_0^{14}\, x_1^{10} + 1296\, x_0^{12}\, x_1^{12}$$
$$+ 960\, x_0^{10}\, x_1^{14} + 375\, x_0^8\, x_1^{16} + 64\, x_0^6\, x_1^{18} + x_1^{24},$$
$$\widetilde{W}_C(x_0, x_1) = 6\, x_0^{20}\, x_1^4 - 24\, x_0^{16}\, x_1^8 + 36\, x_0^{12}\, x_1^{12} - 24\, x_0^8\, x_1^{16} + 6\, x_0^4\, x_1^{20}, \tag{200}$$

the partition functions of the theory $\mathcal{T}(\Lambda_{A_1^{24}})$ are

$$Z_{\mathcal{T}(\Lambda_{A_1^{24}})}[0,0] = \frac{1}{\eta(\tau)^{24}} W_C(\theta_3(2\tau), \theta_2(2\tau)) = q^{-1} + 72 + 4096 q^{\frac{1}{2}} + 98580 q + O(q^{\frac{3}{2}}),$$
$$Z_{\mathcal{T}(\Lambda_{A_1^{24}})}[1,1] = \frac{1}{\eta(\tau)^{24}} \widetilde{W}_C(\theta_3(2\tau), \theta_2(2\tau)) = 96. \tag{201}$$

From (158) and (159), the partition functions of $\mathcal{T}(O_{24})$ and "B&B" must be

$$Z_{\mathcal{T}(O_{24})}[0,0] = q^{-1} + 24 + 4096 q^{\frac{1}{2}} + 98580 q + O(q^{\frac{3}{2}}), \qquad Z_{\mathcal{T}(O_{24})}[1,1] = 48, \tag{202}$$

and

$$Z_{\text{"B\&B"}}[0,0] = q^{-1} + 4096 q^{\frac{1}{2}} + 98580 q + O(q^{\frac{3}{2}}), \qquad Z_{\text{"B\&B"}}[1,1] = 24, \tag{203}$$

which are consistent with the known values (see Appendix C in [59] for "B&B"). In Figure 6, the values corresponding to the SUSY conditions are shown: the number of spin-$\frac{3}{2}$ primary operators in the NS sector, the minimum conformal weight in the R sector, and the partition function $Z_{\mathcal{T}}[1,1]$. As discussed in the previous section, since $\mathcal{T}(\Lambda_{A_1^{24}})$ satisfies the SUSY conditions, $\mathcal{T}(O_{24})$ and "B&B" do as well.

### 7.4.2 Baby Monster CFT

Let us consider a singly-even self-dual code $C$ generated by ( [54])

$$
\begin{bmatrix}
1 & 0 & 0 & 0 & 0 & 0 & 0 & 0 & 0 & 0 & 0 & 0 & 0 & 1 & 0 & 1 & 0 & 1 & 1 & 1 & 0 & 1 & 1 & 0 \\
0 & 1 & 0 & 0 & 0 & 0 & 0 & 0 & 0 & 0 & 0 & 0 & 0 & 1 & 0 & 1 & 1 & 1 & 0 & 0 & 1 & 1 & 0 & 1 \\
0 & 0 & 1 & 0 & 1 & 0 & 0 & 0 & 0 & 0 & 0 & 0 & 0 & 0 & 0 & 0 & 0 & 0 & 0 & 0 & 0 & 0 & 0 & 0 \\
0 & 0 & 0 & 1 & 0 & 0 & 0 & 0 & 0 & 0 & 0 & 0 & 0 & 1 & 1 & 1 & 1 & 0 & 1 & 1 & 0 & 1 & 0 \\
0 & 0 & 0 & 0 & 0 & 1 & 0 & 0 & 0 & 0 & 0 & 0 & 0 & 1 & 0 & 0 & 0 & 1 & 1 & 0 & 1 & 1 & 0 & 0 \\
0 & 0 & 0 & 0 & 0 & 0 & 1 & 0 & 0 & 0 & 0 & 0 & 0 & 0 & 1 & 1 & 0 & 1 & 1 & 1 & 0 & 0 & 0 & 0 \\
0 & 0 & 0 & 0 & 0 & 0 & 0 & 1 & 0 & 0 & 0 & 0 & 0 & 1 & 1 & 1 & 1 & 0 & 0 & 0 & 1 & 0 & 0 & 0 \\
0 & 0 & 0 & 0 & 0 & 0 & 0 & 0 & 1 & 0 & 0 & 0 & 0 & 0 & 0 & 1 & 1 & 1 & 0 & 0 & 0 & 1 & 1 & 0 \\
0 & 0 & 0 & 0 & 0 & 0 & 0 & 0 & 0 & 1 & 0 & 0 & 0 & 1 & 1 & 1 & 1 & 0 & 1 & 1 & 0 & 0 & 0 & 1 \\
0 & 0 & 0 & 0 & 0 & 0 & 0 & 0 & 0 & 0 & 1 & 0 & 0 & 0 & 1 & 0 & 1 & 0 & 0 & 1 & 0 & 0 & 1 & 1 \\
0 & 0 & 0 & 0 & 0 & 0 & 0 & 0 & 0 & 0 & 0 & 1 & 0 & 1 & 0 & 1 & 0 & 0 & 0 & 1 & 0 & 1 & 0 & 1 \\
0 & 0 & 0 & 0 & 0 & 0 & 0 & 0 & 0 & 0 & 0 & 0 & 1 & 1 & 0 & 1 & 1 & 1 & 1 & 1 & 1 & 0 & 1 & 1 \\
\end{bmatrix} .
\tag{204}
$$

Note that for this code $C = C_0 \sqcup C_2$, both associated codes $C_0 \sqcup C_1$ and $C_0 \sqcup C_3$ turn out to be the extended Golay code, which is the Type II self-dual code with minimum weight 8. The weight enumerators of this code $C$ are

$$
\begin{aligned}
W_C(x_0, x_1) = {} & x_0^{24} + x_0^{22} x_1^2 + 77 x_0^{18} x_1^6 + 407 x_0^{16} x_1^8 + 946 x_0^{14} x_1^{10} + 1232 x_0^{12} x_1^{12} \\
& + 946 x_0^{10} x_1^{14} + 407 x_0^8 x_1^{16} + 77 x_0^6 x_1^{18} + x_0^2 x_1^{22} + x_1^{24} ,
\end{aligned}
\tag{205}
$$

and $\widetilde{W}_C(x_0, x_1) = 0$. The NS-NS partition function of the theory $\mathcal{T}$ is

$$
\begin{aligned}
Z_{\mathcal{T}}[0,0] &= \frac{1}{\eta(\tau)^{24}} W_C(\theta_3(2\tau), \theta_2(2\tau)) \\
&= q^{-1} + 4 q^{-\frac{1}{2}} + 72 + 5200 q^{\frac{1}{2}} + 106772 q + O(q^{\frac{3}{2}}).
\end{aligned}
\tag{206}
$$

The R-R partition function is vanishing due to $\widetilde{W}_C(x_0, x_1) = 0$. The Construction A lattice $\Lambda(C)$ can be decomposed into $\mathbb{Z}^2 \times \Lambda_{22}$. The lattice theta function of $\Lambda_{22}$ is

$$
\Theta_{\Lambda_{22}}(\tau) = 1 + 44 q + 4928 q^{\frac{3}{2}} + 85404 q^2 + 788480 q^{\frac{5}{2}} + 4900896 q^3 + O(q^{\frac{13}{4}}),
\tag{207}
$$

where this lattice can be identified with the last row in Table 16.7 [22]. The shift orbifold $\mathcal{T}/H$ and reflection orbifold $\mathcal{T}/G$ has the NS-NS partition function

$$
Z_{\mathcal{T}/G}[0,0] = Z_{\mathcal{T}/H}[0,0] = q^{-1} + 2 q^{-\frac{1}{2}} + 24 + 4648 q^{\frac{1}{2}} + 102676 q + O(q^{\frac{3}{2}}).
\tag{208}
$$

The odd self-dual lattice after orbifolding is $\mathbb{Z} \times O_{23}$ where $O_{23}$ is the shorter Leech lattice whose lattice theta function is

$$
\Theta_{O_{23}}(\tau) = 1 + 4600 q^{\frac{3}{2}} + 93150 q^2 + 953856 q^{\frac{5}{2}} + 6476800 q^3 + O(q^{\frac{13}{4}}).
\tag{209}
$$

After taking both orbifolds, we obtain the NS-NS partition function

$$
Z_{\mathcal{T}/H/G}[0,0] = q^{-1} + q^{-\frac{1}{2}} + 4372 q^{\frac{1}{2}} + 100628 q + O(q^{\frac{3}{2}}).
\tag{210}
$$

This exactly agrees with the partition function of the Baby $\times \psi$ fermionic CFT [59], where Baby denotes the Baby Monster CFT with $c = 47/2$ and $\psi$ denotes the Majorana-Weyl fermion. The Baby Monster CFT was constructed using the Monster CFT in [14]. The partition function of the Baby Monster CFT is

$$
Z_{\text{Baby}}[0,0] = q^{-\frac{47}{48}} + 4371 q^{\frac{25}{48}} + 96256 q^{\frac{49}{48}} + 1143745 q^{\frac{73}{48}} + O(q^{\frac{97}{48}}).
\tag{211}
$$

Each coefficient of the above can be decomposed into dimensions of irreducible representations of the Baby Monster group since the first few of them are $1, 4371, 96255, 1139374, \ldots$. The global symmetry of this CFT has been shown to be the direct product of the Baby Monster group and the cyclic group of order 2 [60].

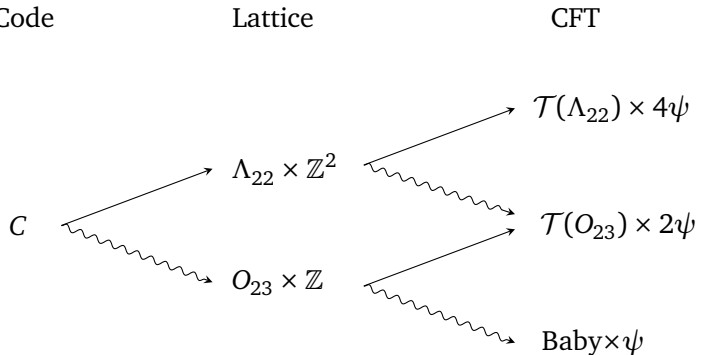

Figure 7: The Baby Monster CFT constructed from the code $C$. $O_{23}$ is the shorter Leech lattice and $\Lambda_{22}$ is the 22-dimensional odd self-dual lattice with 44 norm 2 vectors. Here, "Baby$\times \psi$" stands for the Baby Monster CFT [14] with a Majorana-Weyl fermion.

## 7.5 Chiral fermionic CFTs with $c \geq 32$

We give examples of chiral fermionic CFTs with central charge $c \geq 32$ constructed from binary codes. Our starting point is a list of generator matrices of singly-even self-dual codes [54]. By taking the reflection and shift orbifolds, we construct chiral fermionic CFTs with large energy gaps and show their torus partition functions.

### 7.5.1 $c = 32$

Let us consider a singly-even self-dual code $C$ generated by ( [54])

$$
\begin{bmatrix}
1 & 1 & 0 & 0 & 0 & 0 & 0 & 0 & 0 & 0 & 0 & 0 & 0 & 0 & 0 & 0 & 0 & 0 & 0 & 0 & 0 & 0 & 0 & 0 & 0 & 0 & 0 & 0 & 0 & 0 & 0 & 0 \\
0 & 0 & 1 & 0 & 0 & 0 & 0 & 0 & 0 & 1 & 0 & 0 & 0 & 1 & 0 & 0 & 0 & 1 & 0 & 0 & 0 & 0 & 1 & 1 & 0 & 1 & 1 & 1 & 0 & 0 & 0 & 1 \\
0 & 0 & 0 & 1 & 0 & 0 & 0 & 0 & 0 & 1 & 0 & 0 & 0 & 1 & 0 & 0 & 0 & 1 & 0 & 0 & 0 & 0 & 1 & 1 & 0 & 1 & 1 & 1 & 1 & 1 & 1 & 0 \\
0 & 0 & 0 & 0 & 1 & 0 & 0 & 0 & 0 & 0 & 0 & 0 & 0 & 1 & 0 & 0 & 0 & 1 & 0 & 0 & 0 & 1 & 0 & 1 & 1 & 1 & 1 & 0 & 1 & 1 & 1 & 1 \\
0 & 0 & 0 & 0 & 0 & 1 & 0 & 0 & 0 & 0 & 0 & 0 & 0 & 1 & 0 & 0 & 0 & 1 & 0 & 0 & 0 & 1 & 0 & 1 & 1 & 1 & 1 & 0 & 1 & 0 & 0 & 1 & 1 \\
0 & 0 & 0 & 0 & 0 & 0 & 1 & 0 & 0 & 1 & 0 & 0 & 0 & 0 & 0 & 0 & 0 & 0 & 0 & 0 & 0 & 1 & 1 & 0 & 0 & 1 & 1 & 0 & 0 & 1 & 0 & 1 \\
0 & 0 & 0 & 0 & 0 & 0 & 0 & 1 & 0 & 1 & 0 & 0 & 0 & 0 & 0 & 0 & 0 & 0 & 0 & 0 & 0 & 1 & 1 & 0 & 1 & 1 & 1 & 0 & 1 & 1 & 0 & 1 & 0 \\
0 & 0 & 0 & 0 & 0 & 0 & 0 & 0 & 1 & 1 & 0 & 0 & 0 & 0 & 0 & 0 & 0 & 0 & 0 & 0 & 0 & 1 & 1 & 0 & 0 & 1 & 1 & 0 & 0 \\
0 & 0 & 0 & 0 & 0 & 0 & 0 & 0 & 0 & 0 & 1 & 0 & 0 & 1 & 0 & 0 & 0 & 0 & 0 & 0 & 1 & 0 & 1 & 0 & 0 & 1 & 0 & 1 & 0 & 1 & 1 & 0 \\
0 & 0 & 0 & 0 & 0 & 0 & 0 & 0 & 0 & 0 & 0 & 1 & 0 & 1 & 0 & 0 & 0 & 0 & 0 & 0 & 1 & 0 & 0 & 1 & 1 & 0 & 1 & 0 & 1 & 0 & 0 & 1 \\
0 & 0 & 0 & 0 & 0 & 0 & 0 & 0 & 0 & 0 & 0 & 0 & 1 & 1 & 0 & 0 & 0 & 0 & 0 & 0 & 0 & 1 & 1 & 1 & 1 & 0 & 0 & 1 & 1 & 0 & 0 \\
0 & 0 & 0 & 0 & 0 & 0 & 0 & 0 & 0 & 0 & 0 & 0 & 0 & 0 & 1 & 0 & 0 & 1 & 0 & 0 & 1 & 1 & 1 & 1 & 0 & 1 & 0 & 1 & 0 & 1 & 0 \\
0 & 0 & 0 & 0 & 0 & 0 & 0 & 0 & 0 & 0 & 0 & 0 & 0 & 0 & 0 & 1 & 0 & 1 & 0 & 0 & 1 & 1 & 1 & 1 & 0 & 1 & 0 & 1 & 1 & 0 & 0 & 1 \\
0 & 0 & 0 & 0 & 0 & 0 & 0 & 0 & 0 & 0 & 0 & 0 & 0 & 0 & 0 & 0 & 1 & 1 & 0 & 0 & 1 & 1 & 0 & 0 & 1 & 1 & 0 & 0 & 0 & 0 & 0 \\
0 & 0 & 0 & 0 & 0 & 0 & 0 & 0 & 0 & 0 & 0 & 0 & 0 & 0 & 0 & 0 & 0 & 1 & 0 & 0 & 1 & 1 & 1 & 0 & 1 & 1 & 0 & 0 & 1 & 1 & 0 \\
0 & 0 & 0 & 0 & 0 & 0 & 0 & 0 & 0 & 0 & 0 & 0 & 0 & 0 & 0 & 0 & 0 & 0 & 1 & 1 & 0 & 0 & 0 & 1 & 0 & 1 & 0 & 0 & 1 & 0 & 1
\end{bmatrix}
$$

The weight enumerators of this code are

$$
\begin{aligned}
W_C(x_0, x_1) &= x_0^{32} + x_0^{30} x_1^2 + 19 x_0^{26} x_1^6 + 412 x_0^{24} x_1^8 + 2241 x_0^{22} x_1^{10} + 7040 x_0^{20} x_1^{12} \\
&\quad + 14123 x_0^{18} x_1^{14} + 17862 x_0^{16} x_1^{16} + 14123 x_0^{14} x_1^{18} + 7040 x_0^{12} x_1^{20} \\
&\quad + 2241 x_0^{10} x_1^{22} + 412 x_0^8 x_1^{24} + 19 x_0^6 x_1^{26} + x_0^2 x_1^{30} + x_1^{32},
\end{aligned}
$$

$$
\widetilde{W_C}(x_0, x_1) = 0.
$$

This code contains only a codeword with $\text{wt}(c) = 2$ and no codewords with $\text{wt}(c) = 4$. The NS-NS partition function of the theory $\mathcal{T}$ is

$$
\begin{aligned}
Z_{\mathcal{T}}[0,0] &= \frac{1}{\eta(\tau)^{32}} W_C(\theta_3(2\tau), \theta_2(2\tau)) \\
&= q^{-\frac{4}{3}} + 4q^{-\frac{5}{6}} + 96 q^{-\frac{1}{3}} + 1584 q^{\frac{1}{6}} + 110064 q^{\frac{2}{3}} + O(q^{\frac{7}{6}}),
\end{aligned}
$$

and the R-R partition function is vanishing. After an orbifold by the reflection symmetry $G$ or the shift symmetry $H$, the theory $\mathcal{T}/G \cong \mathcal{T}/H$ has the partition function

$$Z_{\mathcal{T}/G}[0,0] = q^{-\frac{4}{3}} + 2q^{-\frac{5}{6}} + 32q^{-\frac{1}{3}} + 792q^{\frac{1}{6}} + 88048q^{\frac{2}{3}} + O(q^{\frac{7}{6}}). \tag{215}$$

Further, we can take the orbifold and obtain the theory $\mathcal{T}/H/G$ whose partition function is

$$Z_{\mathcal{T}/H/G}[0,0] = q^{-\frac{4}{3}} + q^{-\frac{5}{6}} + 396q^{\frac{1}{6}} + 77040q^{\frac{2}{3}} + O(q^{\frac{7}{6}}). \tag{216}$$

Since this theory contains an excitation of weight 1/2, we can decompose this theory into a Majorana-Weyl fermion and the rest part $\mathcal{T}_R$ with central charge $c = 63/2$ [30].

Dividing (216) by the contribution from a Majorana-Weyl fermion, the latter part has the NS-NS partition function

$$Z_{\mathcal{T}_R}[0,0] = q^{-\frac{21}{16}} + 395q^{\frac{3}{16}} + 76644q^{\frac{11}{16}} + 2099673q^{\frac{19}{16}} + O(q^{\frac{27}{16}}). \tag{217}$$

This shows that any non-trivial operator in this theory has the conformal weight $h \geq 3/2$. Due to the absence of spin one operators, the theory cannot have continuous symmetry.

### 7.5.2 $c = 40$

Let us consider a singly-even self-dual code of length 40 generated by

$$\left[\begin{smallmatrix}
1 & 0 & 0 & 0 & 0 & 0 & 0 & 0 & 0 & 0 & 0 & 0 & 0 & 0 & 0 & 0 & 0 & 0 & 0 & 0 & 1 & 0 & 1 & 0 & 1 & 0 & 1 & 0 & 0 & 1 & 0 & 0 & 1 & 0 & 0 & 0 & 1 & 0 & 0 & 0 \\
0 & 1 & 0 & 0 & 0 & 0 & 0 & 0 & 0 & 0 & 0 & 0 & 0 & 0 & 0 & 0 & 0 & 0 & 0 & 0 & 1 & 1 & 1 & 0 & 1 & 0 & 1 & 1 & 1 & 0 & 1 & 0 & 1 & 0 & 1 & 0 & 1 & 1 & 0 & 1 \\
0 & 0 & 1 & 0 & 0 & 0 & 0 & 0 & 0 & 0 & 0 & 0 & 0 & 0 & 0 & 0 & 0 & 0 & 0 & 0 & 1 & 0 & 1 & 1 & 0 & 0 & 1 & 0 & 0 & 1 & 1 & 0 & 1 & 1 & 1 & 0 & 1 & 0 & 0 & 1 \\
0 & 0 & 0 & 1 & 0 & 0 & 0 & 0 & 0 & 0 & 0 & 0 & 0 & 0 & 0 & 0 & 0 & 0 & 0 & 0 & 1 & 0 & 0 & 0 & 0 & 1 & 1 & 0 & 0 & 1 & 1 & 1 & 1 & 0 & 1 & 1 & 0 & 1 & 1 & 0 \\
0 & 0 & 0 & 0 & 1 & 0 & 0 & 0 & 0 & 0 & 0 & 0 & 0 & 0 & 0 & 0 & 0 & 0 & 0 & 0 & 0 & 1 & 1 & 1 & 1 & 0 & 1 & 0 & 1 & 0 & 1 & 0 & 0 & 1 & 0 & 1 & 0 & 0 & 0 & 1 \\
0 & 0 & 0 & 0 & 0 & 1 & 0 & 0 & 0 & 0 & 0 & 0 & 0 & 0 & 0 & 0 & 0 & 0 & 0 & 0 & 1 & 0 & 1 & 1 & 1 & 1 & 1 & 0 & 1 & 1 & 0 & 0 & 1 & 1 & 1 & 1 & 0 & 0 \\
0 & 0 & 0 & 0 & 0 & 0 & 1 & 0 & 0 & 0 & 0 & 0 & 0 & 0 & 0 & 0 & 0 & 0 & 0 & 0 & 1 & 0 & 1 & 1 & 0 & 1 & 1 & 1 & 1 & 0 & 1 & 1 & 1 & 0 & 1 & 0 & 1 & 0 & 0 & 1 & 1 \\
0 & 0 & 0 & 0 & 0 & 0 & 0 & 1 & 0 & 0 & 0 & 0 & 0 & 0 & 0 & 0 & 0 & 0 & 0 & 0 & 1 & 1 & 0 & 1 & 0 & 1 & 0 & 1 & 0 & 1 & 0 & 0 & 0 & 1 & 1 & 1 & 0 & 1 & 0 & 1 \\
0 & 0 & 0 & 0 & 0 & 0 & 0 & 0 & 1 & 0 & 0 & 0 & 0 & 0 & 0 & 0 & 0 & 0 & 0 & 0 & 1 & 0 & 1 & 0 & 0 & 1 & 1 & 0 & 1 & 0 & 1 & 0 & 0 & 0 & 1 & 0 & 0 & 1 \\
\end{smallmatrix}\right], \tag{218}$$

which is the top matrix of $\mathbb{F}_2$, $n = 40$, $d(C) = 8$ without shadows with $\mathrm{wt}(s) = 4$ in [54]. The weight enumerators of this code are

$$\begin{aligned}
W_C(x_0, x_1) &= x_0^{40} + 125\,x_0^{32}x_1^8 + 1664\,x_0^{30}x_1^{10} + 10720\,x_0^{28}x_1^{12} + 44160\,x_0^{26}x_1^{14} \\
&\quad + 119810\,x_0^{24}x_1^{16} + 216320\,x_0^{22}x_1^{18} + 262976\,x_0^{20}x_1^{20} + 216320\,x_0^{18}x_1^{22} \\
&\quad + 119810\,x_0^{16}x_1^{24} + 44160\,x_0^{14}x_1^{26} + 10720\,x_0^{12}x_1^{28} + 1664\,x_0^{10}x_1^{30} \\
&\quad + 125\,x_0^8 x_1^{32} + x_1^{40},
\end{aligned}$$

and $\widetilde{W}_C(x_0, x_1) = 0$. The NS-NS partition function of the theory $\mathcal{T}$ is

$$Z_{\mathcal{T}}[0,0] = q^{-\frac{5}{3}} + 120q^{-\frac{2}{3}} + 39180q^{\frac{1}{3}} + 1703936q^{\frac{5}{6}} + O(q^{\frac{4}{3}}), \tag{219}$$

and the R-R partition function is zero. After orbifolding, the theory $\mathcal{T}/G \cong \mathcal{T}/H$ has the NS-NS partition function

$$Z_{\mathcal{T}/G}[0,0] = q^{-\frac{5}{3}} + 40q^{-\frac{2}{3}} + 19980q^{\frac{1}{3}} + 1376256q^{\frac{5}{6}} + O(q^{\frac{4}{3}}). \tag{220}$$

Finally, the theory $\mathcal{T}/H/G$ has the NS-NS partition function

$$Z_{\mathcal{T}/H/G}[0,0] = q^{-\frac{5}{3}} + 10380q^{\frac{1}{3}} + 1212416q^{\frac{5}{6}} + O(q^{\frac{4}{3}}). \tag{221}$$

The orbifold reduces the number of excitations with relatively small conformal weights. While the original theory $\mathcal{T}$ and the orbifold $\mathcal{T}/G \cong \mathcal{T}/H$ have operators with $h = 1$, the theory $\mathcal{T}/H/G$ does not have such operators, and the spectral gap becomes $\Delta = 2$.

# 8 Discussion

We have considered chiral fermionic CFTs based on lattices and investigate their orbifolds by two $\mathbb{Z}_2$ symmetries: the reflection symmetry $G$ and the shift symmetry $H$. We have shown the equivalence between the reflection and shift orbifolds using a triality structure, when a lattice is constructed from a binary error-correcting code. We have also systematically computed the partition functions of the orbifold theories for binary and nonbinary codes. Finally, we have provided applications to the code construction of supersymmetric CFTs and chiral fermionic CFTs with no continuous symmetries.

As encountered in $c = 16$ chiral fermionic CFTs in Fig. 5, there is a coincidence between the original theory and their orbifolds, i.e., the self-duality under $\mathbb{Z}_2$-gauging. It is known that a theory that is self-dual under gauging admits a non-invertible duality defect [61]. The duality defects in bosonic lattice CFTs based on $E_8$ lattice and $D_n$ lattice have been constructed in [62, 63]. See also [64] for fermionic theories. It would be interesting to investigate the duality defects in the chiral fermionic CFTs constructed from binary codes.

In section 5, we established the triality in chiral fermionic CFTs from binary codes by explicitly giving the triality operator. The triality also appears in the case of bosonic theories [6]. When constructing the Monster CFT from the extended Golay code through the Leech lattice, the triality together with the automorphism of the Leech lattice (an extension of Conway's group) generates the whole global symmetry of the theory $\mathcal{T}/H/G$, the Monster group. A possible interesting direction is to extend this story to our construction. From the automorphism of lattices through codes, we may give an alternative proof of the discrete global symmetry of the "Beauty and the Beast" SCFT and the Baby Monster CFT, and further identify the global symmetry of the orbifold theories with large central charges such as we have constructed in section 7.5.

As discussed in section 7.3, we proposed the conditions that a CFT constructed from a code must satisfy when it has $\mathcal{N} = 1$ superconformal symmetry [8]. In [65], we examined the modular transformation and the spectral flow of the elliptic genus, i.e., the R-R partition function graded by the U(1) current, of a CFT with $\mathcal{N} = 2$ superconformal symmetry, and organized the conditions under which the CFT constructed from a code satisfies these properties. Although these conditions strongly suggest the existence of supersymmetry, they are merely necessary conditions and do not guarantee the existence of the supercurrent with appropriate OPEs. Deriving sufficient conditions for supersymmetry in CFTs constructed from lattices and the orbifold theories discussed in this paper remains a goal for future work.

## Acknowledgments

We are grateful to Y. Matsuo for their valuable discussions.

**Funding information**   The work of K. K. and S. Y. was supported by FoPM, WINGS Program, the University of Tokyo. The work of K. K. was supported by JSPS KAKENHI Grant-in-Aid for JSPS fellows Grant No. 23KJ0436.

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
