# Peer review of "Orbifolds of chiral fermionic CFTs and their duality"

_SciPost Physics, doi:SciPost Phys. 18, 166 (2025)_

## Round 1 · Referee Report · Anonymous (Referee 1) · 2025-2-3

Strengths

1- Carefully written manuscript

2- Connections with recent classification on c=16 chiral fermionic CFTs

3- Opens a way to construct new chiral CFTs with moderately large values of c, thanks to the code construction

Weaknesses

1- Many technical details that could have been moved to an appendix

Report

This paper considers chiral fermionic CFTs constructed from lattices and error-correcting codes, and various operations on them. The main result is the equivalence of two orbifolds, by reflection and by shift. It also discusses the relation to explicit examples of chiral fermionic CFTs code constructions over Z_2 and general Z_k. This leads to a reproduction of the CFTs from the recent classification at c=16.

Here is some general skepticism: There is a growing literature on explicit constructions of chiral CFTs from lattices/codes, often heavy with details and without a clear physical motivation. In this paper, however, the authors manage to convey an appealing motivation for this work. Apart from the result on the equivalence of orbifold constructions, a great merit of the paper is section 7 which presents an array of interesting examples relating to both known and new CFTs.

The paper is well-written and the main results are summarised in the introduction. The results are interesting and the paper deserves to be published. The presentation could have benefited from moving some material into an appendix, but I accept the authors' choice to keep all information in the main text. I recommend publishing as submitted.

Requested changes

Typo: page 10: fermion party -> fermion parity

Recommendation

Publish (meets expectations and criteria for this Journal)

---

## Round 1 · Referee Report · Anonymous (Referee 2) · 2025-2-27

Report

The manuscript under review studies chiral fermionic theories with a $\mathbb{Z}_2$ symmetry in addition to the conventional fermion parity symmetry. The authors systematically analyze various $\mathbb{Z}_2$ gauging.

The authors expand the known correspondence between error-correcting codes and CFTs into the fermionic cases. They also provides an equivalence between different orbifolds, pointing to a deeper symmetry structure in code-based CFTs.

A key result is to demonstrate a duality between reflection and shift orbifolds for theories built from binary codes, utilizing a triality symmetry between the the $SU(2)$ current algebras, which is nobel and original.

They also present the explicit computation of orbifold partition functions from code invariants, which is original and highly practical. This approach allows for an efficient classification of fermionic CFTs derived from code-based lattices.

The manuscript is well-written and presents interesting and new results. Thus, it will easily pass the criteria to be published in this journal, once a few minor questions are properly addressed:

(1) The duality between $T/G$ and $T/H$ is established under the assumption of a binary code construction, where triality relies on the presence of three $SU(2)$ currents from weight-2 codewords. The reviewer wonders whether such an argument can be generalized to non-binary code CFTs.

(2) In Section 7.5, the authors constructed two theories with a spectral gap greater than 1. The reviewer is curious whether these theories are associated with certain sporadic finite groups.

Recommendation

Ask for minor revision

  • validity: -
  • significance: -
  • originality: -
  • clarity: -
  • formatting: -
  • grammar: -

Author:  Kohki Kawabata  on 2025-04-17  [id 5385]

(in reply to Report 2 on 2025-02-27)

Thank you very much for reviewing our manuscript. We address the questions raised by the referee as follows:

(1) In the binary construction, the SU(2) current is always present in the spectrum due to the structure of Construction A. However, in the non-binary case, the presence of a current depends on the specific code used in constructing the CFT. Therefore, the duality that holds in the binary setting cannot be straightforwardly generalized to the non-binary case.

(2) When the spectral gap exceeds 1, the theory does not exhibit any continuous symmetry but is expected to possess some discrete symmetry. At present, we have not yet identified these discrete symmetries, but it would be interesting to investigate their possible relation to the symmetries of the underlying code.

---

## Round 2 · Referee Report · Anonymous (Referee 2) · 2025-4-21

Report

The authors have successfully addressed the concerns and suggestion. These revisions have improved the clarity of the manuscript. The reviewer is thus pleased to recommend the revised manuscript for publication.

Recommendation

Publish (meets expectations and criteria for this Journal)

---

## Round 2 · List of Changes

• On page 6, we clarified that topological operations generate three independent theories. ("These operations generate three independent Hilbert spaces up to stacking invertible phases:")
  • typo on page 10 raised by the referee (fermion party -> fermion parity)

---

## Editorial Decision

published